



# From model to radar variables: a new forward polarimetric radar operator for COSMO

Daniel Wolfensberger[1] and Alexis Berne[1]

[1]LTE, Ecole polytechnique fédérale de Lausanne (EPFL), Lausanne, Switzerland

*Correspondence to:* EPFL ENAC SSIE-GE , GR C2 564, Station 2 . CH-1015 Lausanne, Switzerland, E-mail:
alexis.berne@epfl.ch

**Abstract.** In this work, a new forward polarimetric radar operator for the COSMO numerical weather prediction
(NWP) model is proposed. This operator is able to simulate measurements of radar reflectivity at horizontal polar-
ization, differential reflectivity as well as specific differential phase shift and Doppler variables for ground based or
spaceborne radar scans from atmospheric conditions simulated by COSMO. The operator includes a new Doppler
scheme, which allows to estimate the full Doppler spectrum, as well a melting scheme which allows to represent the
very specific polarimetric signature of melting hydrometeors. In addition, the operator is adapted to both the oper-
ational one-moment microphysical scheme of COSMO and its more advanced two-moment scheme. The parameters
of the relationships between the microphysical and scattering properties of the various hydrometeors are derived
either from the literature or, in the case of graupel and aggregates, from observations collected in Switzerland. The
operator is evaluated by comparing the simulated fields of radar observables with observations from the Swiss opera-
tional radar network, from a high resolution X-band research radar and from the dual-frequency precipitation radar
of the Global Precipitation Measurement satellite (GPM-DPR). This evaluation shows that the operator is able to
simulate an accurate Doppler spectrum and accurate radial velocities as well as realistic distributions of polarimetric
variables in the liquid phase. In the solid phase, the simulated reflectivities agree relatively well with radar obser-
vations, but the simulated differential reflectivity and specific differential phase shift upon propagation tend to be
underestimated. This radar operator makes it possible to compare directly radar observations from various sources
with COSMO simulations and as such is a valuable tool to evaluate and test the microphysical parameterizations of
the model.

## 1 Introduction

Weather radars deliver areal measurements of precipitation at a high temporal and spatial resolution. Most recent
operational weather radar systems have dual-polarization and Doppler capabilities (called polarimetric in the follow-
ing), which provide not only information about the intensity of precipitation, but also about the type of precipitation
(e.g. phase, homogeneity and shape of hydrometeors). Additionally, the Doppler capability of weather radars allows
to monitor the radial velocity of hydrometeors. In view of their capacities, weather radars offer great opportunities





for validation of and assimilation in numerical weather prediction (NWP) models. This is unfortunately far from being a trivial task since radar observables which are derived from the backscattered power and phase from precipitation cannot be simply put into relation with the state of the atmosphere as simulated by the model. There is thus the need for a conversion tool, able to simulate synthetic radar observations from simulated model variables: a

so-called *forward radar operator*.

Over the past few years, several forward radar operators have been developed. One of the first efforts was made by Pfeifer et al. (2008) who designed a polarimetric operator for the COSMO model, able to simulate horizontal reflectivity $Z_\mathrm{H}$, differential reflectivity $Z_\mathrm{DR}$ and linear depolarization ratio LDR observations. The operator relies on the T-matrix method (Mishchenko et al., 1996) to estimate scattering properties of individual hydrometeors.

Assumptions about shape, density and canting angles, which cannot be obtained from the NWP model were obtained from a sensitivity study. A limitation of this operator is that it does not perform any integration over the antenna power density pattern and thus neglects the beam broadening effect which can be quite significant at longer distances from the radar (Ryzhkov, 2007).

Cheong et al. (2008) developed a three-dimensional stochastic radar simulator able to simulate raw time series of

weather radar data. Doppler characteristics are retrieved by moving discrete scatterers with the three-dimensional model wind field, which allows to produce sample-to-sample time series data, instead of theoretical moments as with conventional radar simulators. Thanks to this, the radar simulator is able to generate the full Doppler spectrum, at the expense, however, of a high computation cost and without taking attenuation into account.

Jung et al. (2008) developed a polarimetric radar operator able to simulate $Z_\mathrm{H}$, $Z_\mathrm{DR}$ as well as the specific

differential phase on propagation $K_\mathrm{dp}$ and adapted it for two different microphysical schemes: one single-moment scheme and one two-moment scheme. The authors also proposed a method to simulate the effect of the melting layer with a weather model that does not explicitly simulate wet hydrometeors. They used this operator to simulate realistic polarimetric radar signatures of a supercell storm from simulations obtained with the Advanced Regional Prediction System (ARPS; Xue et al. (2000)). The validation of the operator was however limited to idealized cases

at S-band only.

Ryzhkov et al. (2011) developed an advanced forward radar operator for a research cloud model with spectral microphysics able to simulate $Z_\mathrm{H}$, $Z_\mathrm{DR}$, LDR and $K_\mathrm{dp}$. Scattering amplitudes of smaller particles are estimated with the Rayleigh approximation whereas the T-matrix method is used for larger hydrometeors. Note, however, that this cloud model is computationally expensive and is not used for operational weather prediction.

Augros et al. (2016) elaborated a polarimetric forward radar operator for the French non-hydrostatic mesoscale research NWP model Meso-NH (Lafore et al., 1998) based on the forward conventional radar operator of Caumont et al. (2006) which simulates all operational polarimetric radar observables: $Z_\mathrm{H}$, $Z_\mathrm{DR}$, the differential phase shift upon propagation $\phi_\mathrm{DP}$, the copolar correlation coefficient $\rho_\mathrm{hv}$ and $K_\mathrm{dp}$. The operator uses the T-matrix method for rain, snow and graupel particles and Mie scattering for pristine ice particles. Beam-broadening is taken into





account by approximating the integration over the antenna normalized power density pattern with a Gauss-Hermite quadrature scheme.

Finally, Zeng et al. (2016) developed a forward radar operator for the COSMO model. The operator is designed for operational purposes (assimilation and validation) with an emphasis on performance and modularity. It simulates
Doppler velocity with fall speed and reflectivity weighting as well as attenuated horizontal reflectivity, with different levels of approximation that can be specified. Note that the operator is currently not able to simulate polarimetric variables.

Most available radar operators are primarily designed to simulate operational PPI (plane position indicator) scans from operational weather radars at S, C or X bands. In research however, other types of radar data are available
which can also be relevant in the evaluation of a NWP model, especially for the simulated vertical structure of precipitation. Some examples of radar data used for research include satellite swaths at higher frequencies, such as measurements of the GPM-DPR satellite at Ku and Ka band (Iguchi et al., 2003) as well as power weighted distributions of scatterer radial velocities (Doppler spectra), commonly recorded by many research radars.

The purpose of this work is to design a state of the art forward polarimetric radar operator for the COSMO
NWP model taking into account the physical aspects of beam propagation and scattering as accurately as possible, while ensuring a reasonable computation time on a standard desktop computer. The radar operator also needs to be versatile and able to simulate a variety of radar variables at many frequencies and for different microphysical schemes, in order to be used in the future as a model evaluation tool with operational and research weather radar data. As such, this radar operator includes a number of innovative features: (1) the ability to simulate the full
Doppler spectrum at a very low computational cost, (2) the ability to simulate observations from both ground and spaceborne radars (3) a probabilistic parameterization of the properties of solid hydrometeors derived from a large dataset of observations in Switzerland, (4) the inclusion of cloud hydrometeors (which contribution becomes important at higher frequencies). Besides, the radar operator has been thoroughly evaluated using a large selection of radar data at different frequencies and corresponding to various synoptic conditions.

The article is structured as follows. In Section 2, a description of the COSMO NWP model as well as the radar data used for the evaluation of the operator is given in Section 3, the different steps of the polarimetric radar operator are extensively described and its assumptions are discussed in details. Section 4 focuses on the qualitative and quantitative evaluation of the simulated radar observables using real radar observations from both operational and research ground weather radars, as well as GPM satellite data. Finally Section 5 summarizes the main results
and opens perspectives for possible applications of the operator.



## 2 Description of the data

### 2.1 COSMO model

The COSMO Model is a mesoscale limited area model initially developed as the Lokal Modell (LM) at the Deutscher Wetterdienst (DWD). It is now operated and developed by several weather services in Europe (Switzerland, Italy, Germany, Poland, Romania and Russia). Besides its operational applications, it is also used for scientific purposes in weather dynamics, microphysics and prediction and for regional climate simulations. The COSMO Model is a non-hydrostatic model based on the fully compressible primitive equations integrated using a split-explicit third-order Runge-Kutta scheme (Wicker and Skamarock, 2002). The spatial discretization is based on a fifth-order upstream advection scheme on an Arakawa C-grid with Lorenz vertical staggering. Height-based Gal-Chen coordinates are used in the vertical (Gal-Chen and Somerville, 1975). The model uses a rotated coordinate system where the pole is displaced to ensure approximatively horizontal resolution over the model domain. Sub-grid scale processes are taken into account with parameterizations.

In COSMO, grid-scale clouds and precipitation are parameterized operationally with a one-moment scheme similar to Rudledge and Hobbs (1983) and Lin et al. (1983), with five hydrometeor categories: rain, snow, graupel, ice crystals and cloud droplets. Snow is assumed to be in the form of rimed aggregates of ice-crystals that have become large enough to have an appreciable fall velocity. Cloud ice is assumed to be in the form of small hexagonal plates that are suspended in the air and have no appreciable fall velocity. The particle size distributions (PSD) are assumed to be exponential for all hydrometeors, except for rain where a gamma PSD is assumed. A more advanced two-moment scheme with a sixth hydrometeor category, hail, was developed for COSMO by Seifert and Beheng (2006). As this scheme significantly increases the overall computation time it is currently not used operationally.

In COSMO, for both microphysical schemes, mass-diameter relations as well as velocity-diameter relations for the precipitating hydrometeors are assumed to be power-laws, except for rain in the two-moments scheme, where a slightly more refined formula by Rogers et al. (1993) is used. Additionally, in contrast with the one-moment scheme, ice crystals are considered to have an appreciable terminal velocity in the two-moment scheme.

For both microphysical schemes, all PSDs can be expressed as particular cases of generalized gamma PSDs.

$$N(D) = N_0 D^\mu \exp\left(-\Lambda \cdot D^\nu\right) \quad \mathrm{m^{-3}mm^{-1}} \tag{1}$$

where $N_0$ is the *intercept* parameter in units of $\mathrm{mm^{-\mu}m^{-3}}$, $\mu$ is the dimensionless *shape* parameter, $\Lambda$ is the *slope* parameter in units of $\mathrm{mm^{-\nu}}$ and $\nu$ is the dimensionless *family* parameter.

In the one-moment scheme, which is used operationally, the only free parameter of the PSDs is $\Lambda$ which can be obtained from the prognostic mass concentrations. $N_0$ is either assumed to be constant during the simulation, or in the case of snow, to be temperature dependent. $\mu$ is equal to zero (exponential PSDs) for all hydrometeors, except for rain where it is set to 0.5 by default and $\nu$ is always equal to one.





In the two-moment scheme, both $\Lambda$ and $N_0$ are prognostic parameters, and can be obtained from the prognostic moment of order zero (number concentration) and from the mass concentration. $\mu$ and $\nu$ are defined *a-priori*.

Table 1 gives the values of the PSD parameters $\mu$, $N_0$, and $\nu$ as well as the *mass-diameter* power-law parameters $a$ and $b$ and the *terminal velocity-diameter* power-law parameters $\alpha$ and $\beta$ for all hydrometeor types and the two microphysical schemes.

| | Rain | Snow | Graupel | Hail | Ice crystals |
|---|---|---|---|---|---|
| $N_0$ | 2529/f | [1]/free | 4000/free | $\emptyset$/free | -/free |
| $\mu$ | 0.5/2 | 0/1.2 | 0/5.37 | $\emptyset$/5 | -/ |
| $\nu$ | 1/1 | 1/1.1 | 1/1.06 | -/1 | $\emptyset$/ |
| $a$ | 5.23e-7/5.24e-7 | 3.80e-8/3.80e-8 | 8.50e-8/8.50e-8 | $\emptyset$/3.39e-7 | 1.3e-7/1.17e-7 |
| $b$ | 3.00/3.00 | 2.00/2.00 | 3.10/3.10 | $\emptyset$/3.00 | 3.00/3.31 |
| $\alpha$ | 4.11/- | 0.871/0.871 | 0.945/1.258 | $\emptyset$/3.362 | -/0.966 |
| $\beta$ | 0.50/- | 0.25/0.20 | 0.89/0.85 | $\emptyset$/0.50 | -/1.20 |

**Table 1.** Parameters of the hydrometeor PSDs and power-laws for the one-moment and two-moment parameterizations (separated by a slash sign). $\emptyset$ indicates that the hydrometeor is not simulated in this scheme, a dash indicates that this parameter is not defined in this parameterization, and "free" indicates a prognostic parameter. Note that the value of $\mu$ for rain can be specified in the COSMO user set-up, 0.5 being the default value. The parameters $a$ and $b$ correspond to the power-law: $m(D) = aD^b$, with $m$ is in kg and $D$ in mm. The parameters $\alpha$ and $\beta$ correspond to the power-law: $v_t(D) = \alpha D^\beta$, with $v_t$ being the terminal fall velocity in m s$^{-1}$, and $D$ is the diameter in mm.

Non-precipitating quantities (cloud droplets and cloud ice) do not have a spectral representation in the one-moment scheme of COSMO, but are instead treated as bulk, with the total number of particles being a function of the air temperature.

In the operational setup, for the parameterization of atmospheric turbulence the COSMO model uses a prognostic turbulent kinetic energy (TKE) closure at level 2.5 similar to Mellor and Yamada (1982). The main difference is the use of variables that are conserved under moist adiabatic processes: total cloud water and liquid water potential temperature. Additionally, a so-called "circulation term" is included which describes the transfer of nonturbulent subgrid kinetic energy from larger-scale circulation toward TKE. The reader is refered to Baldauf et al. (2011) and the model documentation (Doms et al., 2011) for a more in-depth description of the various COSMO sub-grid parameterizations.

## 2.2 Radar data

For the evaluation of polarimetric variables, the final product from the Swiss operational radar network was used. The Swiss network consists of five polarimetric C-band radars, performing PPI scans at 20 different elevation angles

---

[1] for snow, a relation of $N_0$ with the temperature is used (Field et al., 2005)





|  | MXPol | Swiss radar network |
|---|---|---|
| Location | Payerne: 46.813°N, 6.943°E, 495 m a.s.l | Albis: 47.284°N, 8.512°E, 891 m a.s.l |
|  |  | La Dôle: 46.425°N, 6.099°E, 1680 m a.s.l |
|  |  | Monte Lema: 46.040°N, 8.833°E, 1604 m a.s.l |
| Frequency $f$ | 9.41 GHz (X-band) | 5.6 GHz (C-band) |
| Pulse width $\tau$ | 0.5 $\mu$s | 0.577 $\mu$s |
| PRF | 1666 Hz | 500 to 1500 Hz (depends on elevation) |
| FFT length | 128 | - |
| 3dB beamwidth | 1.45° | 1° |
| Sensitivity (SNR = 10dB) | 11 dBZ at 10 km | 0 dBZ at 10 km |

**Table 2.** Specifications of the ground radars used in the evaluation of the radar operator

(Germann et al., 2006). The final quality-checked measurements are corrected for ground clutter, calibrated and aggregated at a resolution of 500 m. In this work, $Z_H$ was used as provided, $Z_{DR}$ was corrected with a daily radar-dependent calibration constant provided by MeteoSwiss and $K_{dp}$ was estimated from $\Psi_{DP}$ using the Kalman filter ensemble method of Schneebeli et al. (2013). Note that two of the operational radars were installed only quite

recently (2014 and 2016) and were thus not used in this study (see Figure 1).

For the evaluation of simulated Doppler variables (mean radial velocity and Doppler spectrum at vertical incidence), observations from a mobile X-band radar (MXPol) deployed in Payerne in Western Switzerland in Spring 2014 were used. The radar was deployed in the context of the PARADISO measurement campaign (Figueras i Ventura et al., 2015). The PARADISO dataset provides a great opportunity to evaluate the simulated radial velocities,

as Payerne is the location from which the radiosoundings, which are assimilated every three hours in the model, are launched.

An overview of the specifications of all radars used in this study is given in Table 2. The location of the three Swiss operational radars used in the evaluation of the radar operator (Section 4.3) and their maximum considered range (100 km) are shown in Figure 1.

Besides ground radar data, measurements from the dual-frequency precipitation radar (DPR, Furukawa et al. (2016)), on-board the core satellite of the Global Precipitation Measurement mission (GPM, Iguchi et al. (2003)) were used to validate the simulation of spaceborne radar swaths. The GPM-DPR radar operates at both Ku (13.6 GHz) and Ka (35.6 GHz) bands. At Ku-band, the satellite swath covers approximately 245 km in width, with an horizontal resolution approximatively 5 km and a 250 m vertical (radial) resolution. At Ka-band, the satellite swath

is more narrow, covering only 125 km in width.



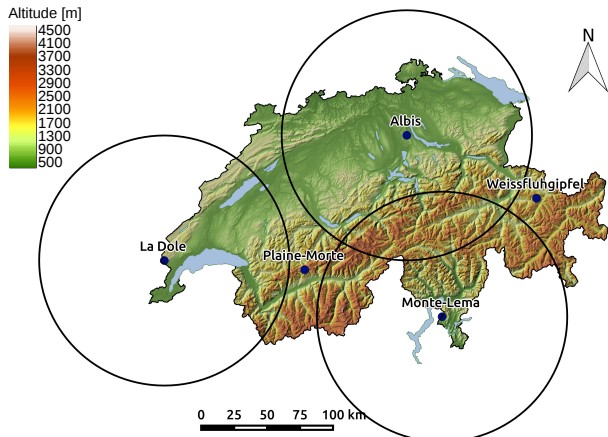

**Figure 1.** Location of the five Swiss operational radars. The black circles indicate the maximum range of radar data (100 km) used for the evaluation of the radar operator (Section 4.3). Since they were installed only quite recently, no data from the Weissfluhgipfel and Plaine Morte radars was used in this study.

## 2.3 Parsivel data

In order to compare the COSMO drop size distribution parameterizations with real observations, data from three Parsivel-1 optical disdrometers were used. These instruments were deployed at short distance from each other, near the Payerne MeteoSwiss station. Like the X-band radar presented above, these instruments were deployed in the context of the PARADISO measurement campaign. The measured drop size distributions were corrected with measurements from a 2-dimensional video disdrometer (2DVD) using the method of Raupach and Berne (2015). For more details regarding these instruments, see Raupach and Berne (2015). All disdrometers were located within the same COSMO grid cell, so the measured DSDs were simply averaged before comparing them with the COSMO parameterizations.

## 2.4 Precipitation events

A list and short description of all five events used for the evaluation of the radar operator with data from the operational C-band radars (Section 4.3) and all six events from the PARADISO campaign used for the evaluation of the radar operator with data from MXPol (Section 4.2) and from Parsivel data (Section 4.4) is given in Table 3.

For the comparison of simulated GPM swaths with real observations, the 100 overpasses with the largest precipitation fluxes recorded between March 2014 and the end of 2016 were selected. Overall, this selection is a balanced mix between widespread low-intensity precipitation and local strong convective storms.



| Event | Description | Used for |
|---|---|---|
| 1 February 2013 | Heavy snowfall event with strong westerly geostrophic winds. | A |
| 22 March 2014 | Stationary front with widespread stratiform liquid precipitation over Switzerland. | B |
| 8 April 2014 | After the crossing of a cold front, presence of mostly liquid widespread stratiform precipitation over Switzerland. | A/B |
| 1th May 2014 | Occlusion over Switzerland with mild temperatures and widespread stratiform precipitation | B |
| 7 May 2014 | Wake of a cold front with scattered stratiform precipitation | B |
| 11 May 2014 | Wake of a cold front with strong scattered stratiform and occasionally convective precipitation | B |
| 14 May 2014 | Occlusion over Switzerland with mild temperatures and widespread stratiform precipitation | B |
| 8 November 2014 | The first two weeks of november 2014 were characterized by very heavy rainfall over the Southern Alps with strong Foehn winds, due to the presence of a very strong low pressure system over the Mediterranean (Xandra). | A |
| 9 January 2015 | Crossing of a warm front over Switzerland with widespread stratiform precipitation and snowfall over the Swiss Alps. | C |
| 26 January 2015 | Snowfall event over the Swiss Alps with very similar characteristics to the 9 January 2015 event | C |
| 23 February 2015 | Crossing of a cold front over Switzerland with some widespread and medium-intensity snowfall | C |
| 13 August 2015 | Strong summer convection triggered by the presence of very warm and wet subtropical air over Switzerland. | A |
| 7 June 2016 | Presence of warm and moist air over Western Europe with a succession of thunderstorms. | A |

**Table 3.** List of all events used for the comparison of simulated radar observables with real ground radar observations. The last column indicates the context of the comparison. A indicates the comparison with the operational C-band radars (Section 4.3), B indicates the comparison with the X-band radar (Section 4.2) and the Parsivel data (Section 4.4 in Payerne and C indicates the evaluation of ice crystals with the X-band radar in the Swiss Alps in Davos (Section 4.6).



## 3 Description of the polarimetric radar operator

The radar operator simulates observations of $Z_\mathrm{H}$, $Z_\mathrm{DR}$, $K_\mathrm{dp}$, average Doppler (radial) velocity and of the full Doppler spectrum based on COSMO simulations and user-specified radar characteristics, such as its position, its frequency, the 3 dB antenna beamwidth $\Delta_\mathrm{3dB}$, the pulse duration $\tau$ and the pulse repetition frequency (PRF). Figure 2 summarizes the main steps of this procedure, which will be more extensively detailed in the further section.

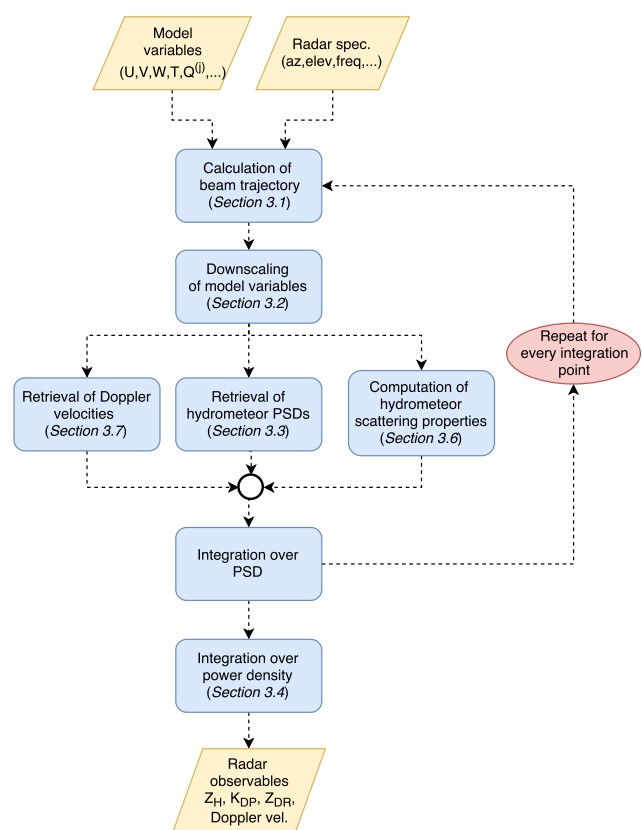

**Figure 2.** Forward operator workflow

### 3.1 Propagation of the radar beam

Microwaves in the atmosphere propagate along curved lines at speeds $v < c$ as the permittivity of the atmosphere $\epsilon$ is larger than $\epsilon_0$, the permittivity of vacuum. In the case of large atmospheric permittivity gradients the beam can even be refracted back to the surface, which can cause distant ground objects to appear on the radar scan. Obviously in order to simulate the propagation of the radar beam, the effect of atmospheric refraction needs to be taken into account. In the radar operator, computing the distance at the ground $s$, and the height above ground $h$ for every radial distance $r$ (see Figure 3), can be done in two ways.





*Equivalent Earth Model*

The Equivalent Earth Model is a simple yet often used model, in which the atmospheric refractive index $n = \sqrt{\epsilon}$ is assumed to be a horizontally homogeneous linear function of height $\frac{dn}{dh} = cst$. This approximation is simple and often used in practice, as it does not require any knowledge about the current state of the atmosphere, and is quite

accurate as long as the assumed vertical profile of $n$ is valid in the first kilometers of the atmosphere.

*Atmospheric refraction model (Zeng et al., 2014)*

In case of non-standard temperature profiles, such as a temperature inversion, the profile of $n$ can vary significantly from the one assumed by the Equivalent Earth Model, which can lead to strong underestimation of the beam refraction. Fortunately Zeng et al. (2014) proposed a more generic and accurate model that is based on the vertical

profile of atmospheric refractivity derived from the model data. This vertical profile can be approximated from the temperature $T$, the partial pressure of water vapour $P_w$ and the total pressure $P$ (Doviak and Zrnić, 2006). The height at a given range can then be estimated by solving a second order ordinary differential equation derived from Snell's law for spherically stratified layers. Again, this model assumes horizontal homogeneity of the atmospheric refractivity.

The choice of the refraction model (Earth equivalent or atmospheric refraction) is left to the user of the radar operator, noting that the computation cost for the latter is slightly larger. The whole evaluation of the radar operator presented in Section 4 was performed with the more advanced model of Zeng et al. (2014).

## 3.2   Downscaling of model variables

Once the distance at the ground $s$, and the height above ground $h$, are obtained from the refraction model, it is

easy to retrieve the lat/lon/height coordinates $(\psi^{\mathrm{WGS}}, \lambda^{\mathrm{WGS}}, h)$ of the corresponding radar gate, knowing the beam elevation $\theta_0$ and azimuth $\phi_0$ angles, as well as the position of the radar.

Once the coordinates of all radar gates have been defined, the model variables must be downscaled to the location of the radar gates. The advantage of downscaling model variables before estimating radar observables, instead of doing the opposite, is twofold. At first, it is much more computationally efficient, because computing radar observables

requires numerical integration over a particle size distribution at every bin, which is costly. Secondly, model variables are much less correlated than the final radar observables, which tend to be strongly correlated, with the exception of the radial velocity. This was tested by computing the Spearman rank correlations for a representative subsets of model simulations. For model variables, the correlations are generally low ($\pm 0.2$), except between temperature, snow and graupel concentration where they are around 0.7. For radar observables, however, the correlations are very

high (almost 1 between $K_{\mathrm{dp}}$ and $Z_{\mathrm{H}}$ and around 0.9 between $Z_{\mathrm{DR}}$ and $Z_{\mathrm{H}}$. Since multidimensional downscaling is difficult and expensive, it is thus preferable to independently downscale the less correlated model variables.

Technical details about the trilinear downscaling procedure are given in Appendix A.



### 3.3 Retrieval of particle size distributions

In the one-moment scheme, for a given hydrometeor $j$, the COSMO specific mass concentration $Q_M^{(j)}$ in kg·m$^{-3}$ is proportional to a specific moment of the particle size distributions (PSD), since the COSMO parameterizations assumes simple power-laws for the mass-diameter relations: $m^{(j)}(D) = a^{(j)} D^{b^{(j)}}$. Because all COSMO PSDs belong

to the class of generalized gamma PSDs, $Q_M$ can be expressed as:

$$Q_M^{(j)} = a^{(j)} \int_{D_{\min}^{(j)}}^{D_{\max}^{(j)}} D^{b^{(j)}} \cdot \overbrace{N_0^{(j)} D^{\mu^{(j)}} \exp\left(-\Lambda^{(j)} D^{\nu^{(j)}}\right)}^{N^{(j)}(D)} \, dD \tag{2}$$

As in the COSMO microphysical parameterization (see Doms et al. (2011)), the PSDs are assumed to be only weakly truncated and the integration bounds $[D_{\min}^{(j)}, D_{\max}^{(j)}]$ are replaced by $[0, \infty)$, in order to get an analytical solution and avoid the cost of numerical root finding. Note that this truncation hypothesis is done only for the

retrieval of $\Lambda$ and not when computing the radar observables (Section 3.6.3 and Appendix C). For the one-moment scheme, by integrating the Equation 2, one gets the following expression for the free parameter $\Lambda^{(j)}$.

$$\Lambda_{1\text{mom}}^{(j)} = \left( \frac{N_0^{(j)} a^{(j)} \Gamma\left( \frac{b^{(j)} + \mu^{(j)} + 1}{\nu^{(j)}} \right)}{\nu^{(j)} Q_M^{(j)}} \right)^{\frac{\nu^{(j)}}{b^{(j)} + \mu^{(j)} + 1}} \tag{3}$$

For the two-moment scheme, the method is similar, except that both mass and number concentrations are needed to retrieve $\Lambda$ and $N_0$. The corresponding mathematical formulation is given in Appendix B.

Equation 3 allows to retrieve the PSD parameters for all hydrometeors in Table 1[2] at every radar gate using the model variable $Q_M^{(j)}$, and, for the two-moments scheme, $Q_N^{(j)}$ as well. Knowing the PSDs ($N^{(j)}(D)$) makes it possible to perform the integration of polarimetric variables over ensemble of hydrometeors as will be described in the next steps of the operator.

The contribution of ice crystals and cloud droplets to the overall radar signature has often been neglected in other

radar simulators (e.g., Augros et al., 2016; Jung et al., 2008). In our radar operator, cloud droplets are neglected because the radar operator is designed for common precipitation radar frequencies (2.7 up to 35 GHz), for which the contribution of cloud droplets is very small (Fabry, 2015). However at higher frequencies and in weak precipitation, the contribution of ice crystals can be significant, especially for $Z_{\text{DR}}$, as these crystals can be quite oblate (Battaglia et al., 2001). Therefore, ice crystals are considered explicitly, even though they do not have a spectral representation

in the one-moment scheme of COSMO. Instead, a realistic PSD is retrieved with the double-moment normalization method of Lee et al. (2004). This formulation of the PSD requires to know two moments of the PSD as well as an appropriate normalized PSD function. Field et al. (2005) proposes best-fit relations between the moments of ice

---

[2]except for the ice crystals in the one-moment scheme, where COSMO does not consider any spectral representation





crystals PSDs as well as fits of generating functions for different pair of moments. Taking advantage of these results, the PSDs of ice crystals in the operator can be retrieved by estimating the second moment from the third moment (the COSMO mass concentration) and by using these two moments with the corresponding generating function proposed by Field et al. (2005).

## 3.4  Integration over the antenna pattern

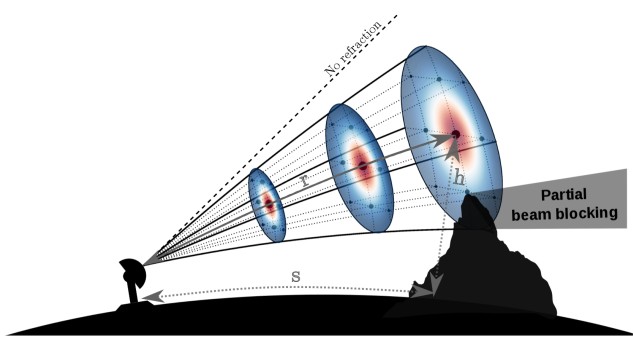

**Figure 3.** Beam broadening increases the sampling volume with increasing range and is caused by the fact that the normalized power density pattern of the antenna (shown in red/blue tones) is not completely concentrated on the beam axis. The blue dots correspond to the integration points used in the quadrature scheme (in this case with $J, K = 3$ for illustration purposes) and their size depends on their corresponding weights. The effect of atmospheric refraction on the propagation of the radar beam is also illustrated: $r$ is the radial distance (radar range), $s$ is the ground distance and $h$ the distance above ground of a given radar gate, which need to be estimated accurately.

Part of the transmitted power is directed away from the axis of the antenna main beam, which will increase the size of the radar sampling volume with range, an effect known as beam-broadening. Depending on the antenna beamwidth this effect can be quite significant and needs to be accounted for by integrating the radar observables at every gate over the antenna power density pattern. Equation 4 formulates the antenna integration for an arbitrary
radar observable $y$ and a normalized power density pattern of the antenna represented by $f^2$, as in Doviak and Zrnić (2006).

$$I\left[y\right]\left(r_o,\theta_o,\phi_o\right) = \int_{r_o-\Delta_r/2}^{r_o+\Delta_r/2} \int_{\theta_o-\pi/2}^{\theta_o+\pi/2} \int_{\phi_o-\pi}^{\phi_o+\pi} y(r,\theta,\phi)f^4(\theta_0-\theta,\phi_0-\phi)|W(r_0-r)|^2 \cos\theta\, dr\, d\theta\, d\phi \quad (4)$$

In our operator, similarly to Caumont et al. (2006) and Zeng et al. (2016), we set $W(r_0-r) = 1$ if $r \in \left[r_0 - \frac{c\tau}{4}, r_0 + \frac{c\tau}{4}\right]$ and $W(r_0 - r) = 0$ otherwise. Indeed since the model resolution (1-2 km) is about one order of magnitude larger
than the typical gate length of a modern radar (80-250 m), effects related to the finite receiver bandwidth can be





neglected. Integration over $r$ can still be done *a posteriori* by using a higher radial resolution and aggregating the simulated radar observables afterwards.

Another often used simplification is to neglect side lobes in the power density pattern and to approximate $f^2$ by a circularly symmetric Gaussian. These simplifications reduce the integration to Equation 5.

$$5 \quad I[y](r_o, \theta_o, \phi_o) = \int\limits_{\theta_o-\pi/2}^{\theta_o+\pi/2} \int\limits_{\phi_o-\pi}^{\phi_o+\pi} y(r_0, \theta, \phi)\exp\left(-8\ln 2\frac{(\theta_0-\theta)^2}{\Delta_{3\mathrm{dB}}} - 8\ln 2\frac{(\phi_0-\phi)^2}{\Delta_{3\mathrm{dB}}}\right)\cos\theta d\theta d\phi \quad (5)$$

This integration can be accurately approximated with a Gauss-Hermite quadrature (Caumont et al., 2006):

$$I[y](r_o, \theta_o, \phi_o) \approx \sum_{j=1,k=1}^{J,K} w'_j w'_k y(r_0, \theta_0 + z'_j, \phi_0 + z'_k)\cos(\theta_0 + z'_k) \quad (6)$$

where $w' = \sigma w, z' = \sigma z$ with $\sigma = \frac{\Delta_{3\mathrm{dB}}}{2\sqrt{2\log 2}}$, where $\Delta_{3\mathrm{dB}}$ is the 3 dB beamwidth of the antenna in degrees, and $w$ and $z$ are respectively the weights and the roots of the Hermite polynomial of order $J$ (for azimuthal integration) or $K$ (for elevational integration). For the integration in the radar operator, default values of $J = 5$ and $K = 7$ are used according to Zeng et al. (2016). The quadrature points thus correspond to separate sub-beams with different azimuth and elevation angles that are resolved independently. A schematic example of this quadrature scheme is shown in Figure 3 for $J, K = 3$.

Another advantage of using a quadrature scheme is that is makes it easy to consider partial beam-blocking (grayed out area in Figure 3). Note that in our operator, the blocked sub-beams are simply lost (i.e. are not considered in the integration) and no modelling of ground echoes is performed. However, as was done in the evaluation of the operator (Section 4), these beams can easily be identified and removed when comparing simulated radar observables with real measurements.

The choice of this simple Gaussian quadrature was validated by comparison with an exhaustive integration scheme during three precipitation events (two stratiform and one convective). The exhaustive integration consists in the decomposition of a real antenna pattern (obtained from lab measurements) into a regular grid of $200 \times 200$ sub-beams. Such an integration is obviously extremely computationally expensive and can not be considered as a reasonable choice of quadrature in practice. Four other quadrature schemes were tested, (1) a sparse Gauss-Hermite quadrature scheme (Smolyak, 1963), (2) a custom hybrid Gauss-Hermite/Legendre quadrature scheme based on the decomposition of the real antenna diagram in radial direction with a sum of Gaussians (3) a Gauss-Legendre quadrature scheme weighted by the real antenna pattern and (4) a recursive Gauss-Lobatto scheme (Gautschi, 2006) based on the real antenna pattern. All schemes were tested in terms of bias and root mean square error (RMSE) in horizontal reflectivity $Z_{\mathrm{H}}$ and differential reflectivity $Z_{\mathrm{DR}}$ as a function of beam elevation (from 0 to 90°), taking the exhaustive integration scheme as a reference. Figure 4 shows an example for one of the two stratiform events. It was observed that the simple Gauss-Hermite scheme was the one which performed the best on average (lowest bias and RMSE for



both $Z_\mathrm{H}$ and $Z_\mathrm{DR}$), with schemes (1) and (3) performing almost systematically worse. Schemes (2) and (4) tend to perform slightly better at low elevation angles in particular situations where strong vertical gradients are present, generated for instance by a melting layer or by strong convection. This is due to the fact that in these situations, the contribution of the side lobes can become quite important, for example when the main beam is located in the solid precipitation above the melting layer but the first side lobe shoots through the melting layer or the rain underneath. However, considering that these schemes are more computationally expensive and tend to perform worse at elevations $> 3°$, it was decided to keep the simple Gauss-Hermite scheme, which seems to offer the best trade-off. As an improvement to the operator, it could however be possible to use an adaptive scheme that depends on the specific state of the atmosphere and the beam elevation.

## 3.5 Derivation of polarimetric variables

All radar observables for a simultaneous transmitting radar can be defined in terms of the backscattering covariance matrix $C^b$ and the forward scattering vector $S^f$. For a given hydrometeor of type $(j)$ and diameter $D$.

$$C^{b,(j)}(D) = \begin{bmatrix} |s_{hh}^{b,(j)}|^2 & s_{vv}^{b,(j)}\left(s_{hh}^{b,(j)}\right)^* \\ s_{hh}^{b,(j)}\left(s_{vv}^{b,(j)}\right)^* & |s_{vv}^{b,(j)}|^2 \end{bmatrix} \in \mathcal{R}^{2\times2} \tag{7}$$

and

$$S^{f,(j)}(D) = \begin{bmatrix} s_{hh}^{f,(j)} \\ s_{vv}^{f,(j)} \end{bmatrix} \in \mathcal{C}^{2\times1} \tag{8}$$

where the superscripts $b$ and $f$ indicate backward, respectively forward scattering directions and $s$ are elements of the scattering matrix that relates the scattered electric field to the incident electric field for a given particle of diameter $D$.

The radar backscattering cross sections $\sigma^b$ are easily obtained from $C^b$:

$$\sigma_h^{b,(j)}(D) = 4\pi C_{1,1}^{b,(j)}(D)$$
$$\sigma_v^{b,(j)}(D) = 4\pi C_{2,2}^{b,(j)}(D) \tag{9}$$

All polarimetric variables at the radar gate polar coordinates $(r_o, \theta_o, \phi_o)$ are function of $C^b$ and $S^f$ and can be otained by first integrating these scattering properties over the particle size distributions, summing them over all hydrometeor types and finally integrating them over the antenna power density. The exhaustive mathematical formulation of all simulated radar observables is given in Appendix C. Additionally, real radar observations of $Z_\mathrm{H}$ and $Z_\mathrm{DR}$ are affected by attenuation, which needs to be accounted for to simulate realistic radar measurements. The

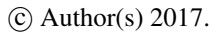



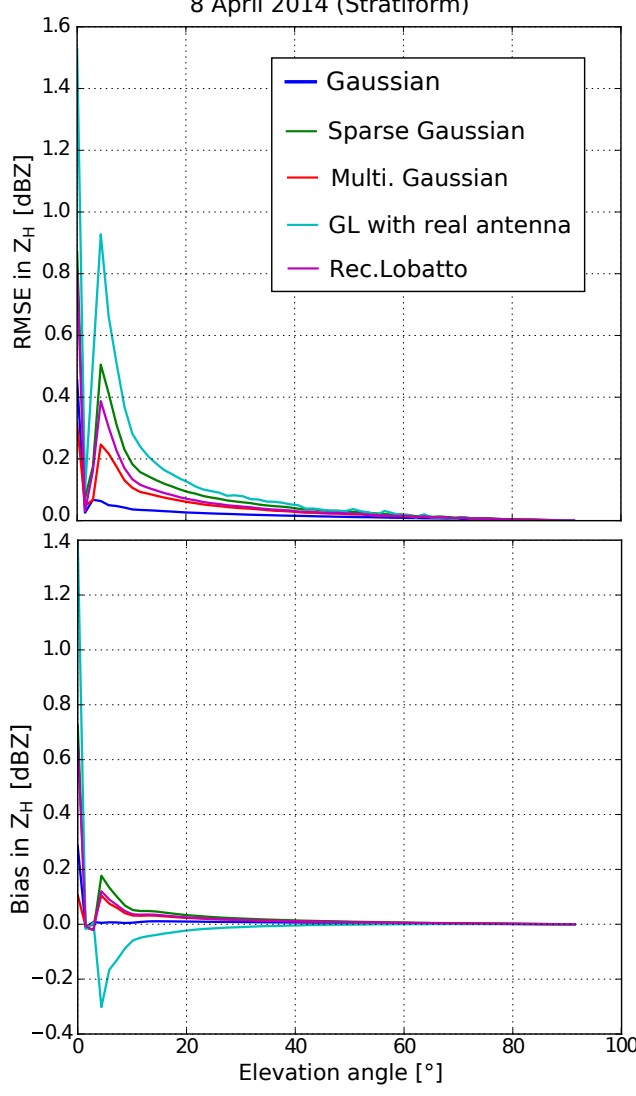

**Figure 4.** Bias and RMSE in terms of $Z_{\mathrm{H}}$ during one day of stratiform of precipitation (around 120 RHI scans), for the five possible quadrature schemes. The exhaustive quadrature scheme is used as a reference. The other two events show similar results.

specific differential phase shift on propagation $K_{\mathrm{dp}}$ also needs to be modified in order to account for the specific phase shift on backscattering (see Appendix C).

### 3.6   Scattering properties of individual hydrometeors

Estimation of $C^{b,(j)}$ and $S^{f,(j)}$ for individual hydrometeors is performed with the transition-matrix (T-matrix) method. The T-matrix method is an efficient and exact generalization of Mie scattering by randomly oriented non-





spherical particles (Mishchenko et al., 1996). Since the shape of raindrops is widely accepted to be well approximated by spheroids (e.g., Andsager et al., 1999; Beard and Chuang, 1987; Thurai et al., 2007), the T-matrix method provides a well suited method for the computation of the scattering properties of rain. This method was also used for the solid hydrometeors (snow, graupel and hail), at the expense of some adjustments, that will be described later

on.

The T-matrix method requires knowledge about the permittivity, the shape as well as the orientation of particles. Since particles are assumed to be spheroids, the aspect-ratio $a_r$, defined in the context of this work as the ratio between the smallest dimension and the largest dimension of a particle, is sufficient to characterize their shapes. The orientation $o$ is defined as the angle formed between the horizontal and the major axis (canting angle $\in$ [-90,90])

and can be characterized with the Euler angle $\beta$ (pitch).

In order to make the overall computation time reasonable, the scattering properties for the individual hydrometeors are pre-computed for various common radar frequencies and stored in three-dimensional lookup tables: *diameter*, *elevation* and *temperature* for dry hydrometeors and *diameter*, *elevation* and *wet fraction* for wet hydrometeors (Section 3.7). On run time, these scattering properties are then simply queried from the lookup tables, for a given

elevation angle and temperature/wet fraction.

### 3.6.1   Aspect-ratios and orientations

*Rain*

For liquid precipitation (raindrops), the aspect-ratio model of Thurai et al. (2007) is used and the drop orientation are assumed to be normally distributed with a zero mean and a standard deviation of 7° according to Bringi and

Chandrasekar (2001).

*Snow and graupel*

For solid precipitation, estimation of these parameters is a much more arduous task, since solid particles have a very wide variability in shape. Few aspect-ratio models have been reported in the literature and even less is known about the orientations of solid hydrometeors.

In terms of aspect-ratio, Straka et al. (2000) report values ranging between 0.6 and 0.8 for dry aggregates and between 0.6 and 0.9 for graupels while Garrett et al. (2015) reports a median aspect-ratio of 0.6 for aggregates and a strong mode in graupel aspect-ratios around 0.9.

In terms of orientation distributions, both Ryzhkov et al. (2011) and Augros et al. (2016) consider a Gaussian distribution with zero mean and a standard deviation of 40° for aggregates and graupels in their simulations.

Given the large uncertainty associated with the geometry of solid hydrometeors, a parameterization of aspect-ratios and orientations for graupel and aggregates was derived using using observations from a multi-angle snowflake camera (MASC). A detailed description of the MASC can be found in Garrett et al. (2012). MASC observations recorded during one year in the Eastern Swiss Alps were classified with the method of Praz et al. (2017), giving a total of around 30'000 particles for both hydrometeor types. The particles were grouped into 50 diameter classes





and inside every class a probability distribution was fitted for the aspect-ratio and the orientations. For sake of numerical stability, the fit was done on the inverse of the aspect-ratio (large dimension over small dimension). In accordance with the microphysical parameterization of the model, the considered reference for the diameter of solid hydrometeors is their maximum dimension.

The inverse of aspect ratio, $1/a_r$, is assumed to follow a gamma distribution, whereas the canting angle $o$ is assumed to be normally distributed with zero mean, and the parameters of these distributions depend on the considered diameter bin $\lfloor D \rfloor$.

$$o \sim \mathcal{N}\left(0, \sigma_o^{\lfloor D \rfloor}\right) \tag{10}$$

$$1/a_r \sim \Gamma\left(\Lambda_{a_r}^{\lfloor D \rfloor}, l, M^d\right) = \frac{(a_r^{\lfloor D \rfloor} - l)^{\Lambda_{a_r}^{\lfloor D \rfloor} - 1} \exp\left(-\frac{a_r^{\lfloor D \rfloor} - 1}{M^{\lfloor D \rfloor}}\right)}{(M^{\lfloor D \rfloor})^{\Lambda_{a_r}^{\lfloor D \rfloor}} \Gamma(\Lambda_{a_r}^{\lfloor D \rfloor})}, \quad \text{where } l = 1 \tag{11}$$

where $\Lambda_{a_r}$ and $M$ are the *shape* and *scale* parameters of the gamma aspect-ratio probability density function and $\sigma_o$ is the *standard deviation* of the Gaussian canting angle distribution. The superscript $\lfloor D \rfloor$ indicates that these parameters depend on the considered diameter bin $d$. Note that the gamma distribution is rescaled with a constant factor $l = 1$, to account for the fact that the smallest possible aspect-ratio is 1 and not 0. The relationship of all parameters $\Lambda_{a_r}$, $M$, and $\sigma_o$ to the diameter bins $\lfloor D \rfloor$, was fitted with a power law, which allows to estimate

them for any arbitrary maximum diameter $D$. This also allows integration over the canting angle and aspect-ratio distributions for all particle sizes.

    Figure 5 shows the fitted densities for every diameter and every value of inverse aspect-ratio and canting angle. Overlaid are the empirical quantiles (dashed lines) and the quantiles of the fitted distributions (solid lines). Generally the match is quite good. The fitted models are able to take into account the increase in aspect-ratio spread and

decrease in canting angle spread with particle size, which are the two dominant trends that can be identified in the observations.

    Figure 6 shows the effect of using this MASC-based parameterization instead of the values from the litterature (Ryzhkov et al., 2011) on the resulting polarimetric variables. Whereas only a small increase is observed for the horizontal reflectivity $Z_H$, the difference is quite important for $Z_{DR}$ and $K_{dp}$, especially for graupel. The MASC

parameterization tends to produce a stronger polarimetric signature. It is interesting to notice that $Z_{DR}$ tends to decrease with the concentration, which is rather counter-intuitive as $Z_{DR}$ is thought to be independent of concentration effects. This can be explained by the fact that, in COSMO, the density of snowflakes decreases with their size (they become less compact) and therefore the permittivity computed with the mixture model decreases as well. When the concentration increases, the proportion of larger (and more oblate) snowflakes increases but given their smaller

permittivity, the overall trend is a slight decrease in $Z_{DR}$. This trend hence reflects an assumption in COSMO, not necessarily the reality.





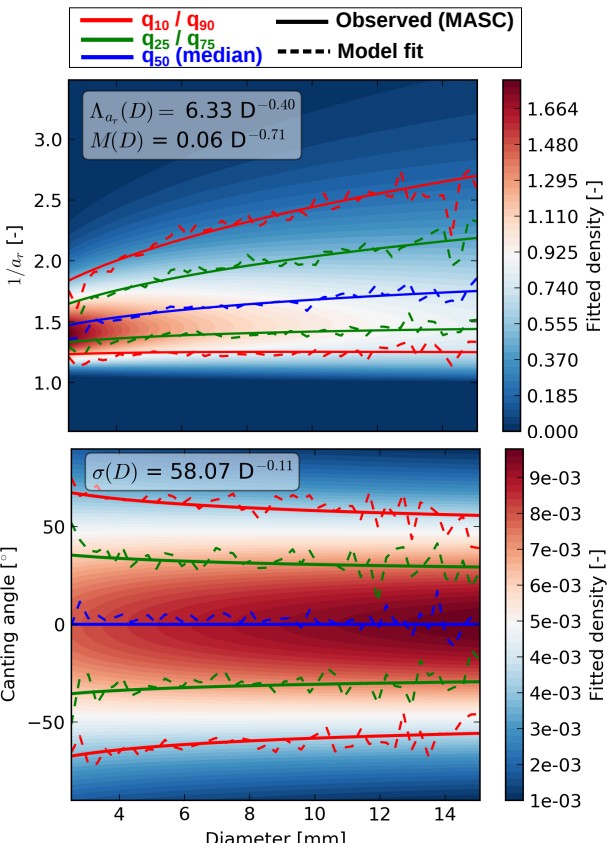

**Figure 5.** Fitted probability density functions for the inverse of the aspect-ratio (top) and the canting angle (bottom). The power-laws relating the particle density function parameters to the diameter are displayed in the grey boxes on the top-left. Note that the fit was performed on the inverse of the aspect-ratio (major axis over minor axis).

Note that even if this increase in the polarimetric signature of aggregates and graupel seems particularly drastic, comparisons with real radar measurements indicate that the operator is still underestimating the polarimetric variables in snow (Section 4.3).

*Hail*

5    A similar analysis could not be performed for hail, as no MASC observations of hail were available. Hence, the canting angle distribution is assumed to be Gaussian with zero mean and a standard deviation of $40°$, whereas the aspect-ratio model is taken from (Ryzhkov et al., 2011).

$$a_r^{\text{hail}} = \begin{cases} 1 - 0.02D, & \text{if } D < 10 \text{ mm} \\ 0.8, & \text{if } D \geq 10 \text{ mm} \end{cases} \tag{12}$$





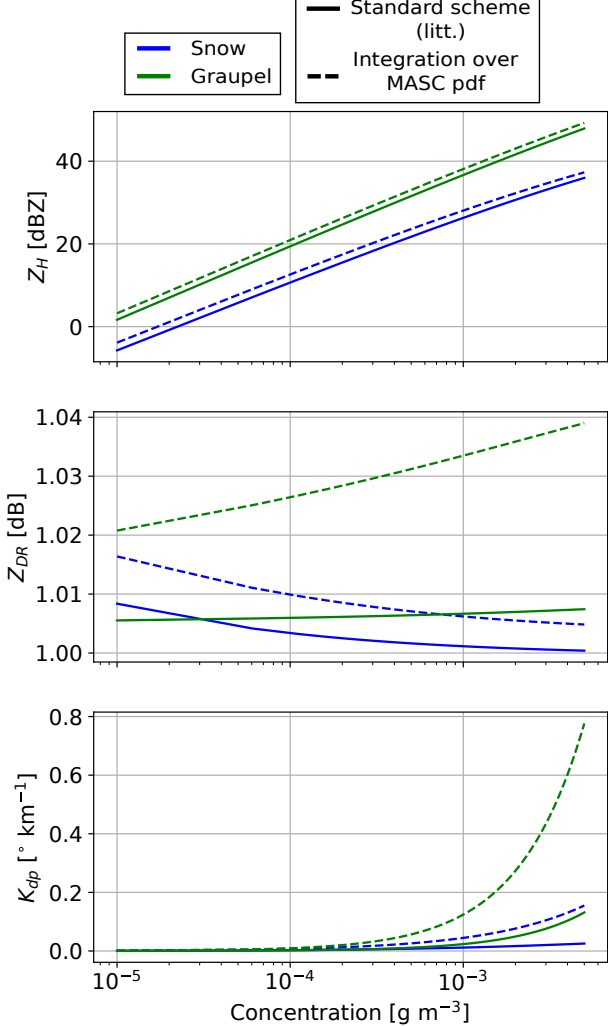

**Figure 6.** Polarimetric variables as a function of the mass concentration for snow and graupel when using canting angle and aspect-ratio parameterizations from the litterature (Ryzhkov et al., 2011) (solid line) and when using the parameterization based on MASC data (dashed line).

*Ice crystals*

For ice-crystals, the aspect-ratio model is taken from Auer and Veal (1970) for hexagonal columns, whereas the canting angle distribution is assumed to be Gaussian with zero mean and a standard deviation of 5°, which corresponds to the upper range of the canting angle standard deviations observed by Noel and Sassen (2005) in cirrus and midlevel clouds.





### 3.6.2 Permittivities

*Rain*

For the permittivity of rain $\epsilon^r$, the well known model of Liebe et al. (1991) for the permittivity of water at microwave frequencies is used. Note that recently, a new model for water permittivity has been proposed by Turner et al. (2016), which appears to provide a better agreement with field observations at high frequencies. However, for common precipitation radar frequencies ($< 30$ GHz) and temperatures ($> -20°$) both models agree very well.

*Snow, graupel, hail and ice crystals*

Dry solid hydrometeors consist of a mixture of air and solid ice. The permittivity of such mixtures can be estimated with the Maxwell-Garnett mixing formula (e.g., Ryzhkov et al., 2011).

$$\epsilon^{(j)} = \frac{1 + 2\frac{\rho_{(j)}}{\rho_i}\frac{\epsilon_i - 1}{\epsilon_i + 2}}{1 - \frac{\rho^{(j)}}{\rho_i}\frac{\epsilon_i - 1}{\epsilon_i + 2}} \tag{13}$$

where $\rho_i$ and $\rho^{(j)}$ are respectively the densities of ice and of the given hydrometeor (snow, graupel or hail) and $\epsilon_i$ is the permittivity of ice, which can be estimated with Hufford (1991)'s formula.

The densities $\rho^{(j)}$ can be easily obtained from the COSMO mass-diameter relations $\rho^{(j)} = \frac{a^{(j)}D^{(b)}}{\pi/6D^3}$ and the density of ice is assumed to be constant $\rho_i = 916$ kg m$^{-3}$.

### 3.6.3 Integration of scattering properties

The matrices $C^{b,(j)}(D)$ (Equation 7) and $S^{f,(j)}(D)$ (Equation 8) are obtained by integration over distributions of canting angles and, for snow and graupel, aspect-ratios. For $C^{b,(j)}$ this gives for snow and graupel:

$$C^{b,(j)}(D) = \frac{1}{2\pi} \int_0^{2\pi} \int_0^{\pi} \int_0^1 c^{b,(j)}(D, a_r, \alpha, \beta) \cdot sin(\beta) \; p(\beta)p(a_r) \; d\alpha \; d\beta \; da_r \tag{14}$$

And for rain and hail, where $a_r$ is constant for a given diameter:

$$C^{b,(j)}(D) = \frac{1}{2\pi} \int_0^{2\pi} \int_0^{\pi} c^{b,(j)}(D, \alpha, \beta) \; d\alpha \; d\beta \tag{15}$$

where $c^{b,(j)}(D, \alpha, \beta)$ are the scattering properties for a fixed diameter and $\alpha$ and $\beta$ (the canting angle) are the yaw (azimuthal orientation) and pitch Euler angles. The only difference between $o$ and $\beta$ is that $\beta \in [0, \pi]$ and $o \in [-\pi/2, \pi/2]$. $p(\beta)$ and $p(a_r)$ are the probabilities of $\beta$ and $a_r$ for a given diameter $D$ as obtained from Equations 11 and 10. Note that the final scattering properties are averaged over all azimuthal angles $\alpha$, which are all considered to be equiprobable. The $sin(\beta)$ in the equation is the *surface element* which arises from the fact that the integration over $\alpha$ and $\beta$ is a surface integration in spherical coordinates. The procedure for $S^f$ is exactly the same.





Since the computation of the T-matrix for a large number of canting angles and aspect-ratios can be quite expensive, two different quadrature schemes were used, one Gauss-Hermite scheme for the integration over the Gaussian distributions of canting angles, and one recursive Gauss-Lobatto scheme (Gander and Gautschi, 2000) for the integration over aspect-ratios.

### 3.6.4 Taking into account the radar sensitivity

The received power at the radar antenna decreases with the square of the range, which will lead to a decrease of signal-to-noise ratio (SNR) with the distance. To take into account this effect, all simulated radar variables at a range $r_0$ are censored if:

$$Z_{\mathrm{H}}(r_0) < S + G + SNR_{\mathrm{thr}} + 20 \cdot \log_{10}(r) \tag{16}$$

where $G$ is the overall radar gain in dBm, $S$ is the radar sensitivity in dBm, $Z_{\mathrm{H}}$ is the horizontal reflectivity in dBZ and $r$ is the range in km. $SNR_{\mathrm{thr}}$ corresponds to the desired signal-to-noise threshold in dB (typically 8 dB in the following).

### 3.7 Simulation of the melting layer effect

Stratiform rain situations are generally associated with the presence of a melting layer (ML), characterized by a strong signature in polarimetric radar variables (e.g., Szyrmer and Zawadzki, 1999; Fabry and Zawadzki, 1995; Matrosov, 2008; Wolfensberger et al., 2016). In order to simulate realistic radar observables, this effect needs to be taken into account by the radar operator. Unfortunately COSMO does not operationally simulate wet hydrometeors, even though a non-operational parameterization was developed by Frick and Wernli (2012). Jung et al. (2008) proposed a method to retrieve the concentration of wet snow aggregates by considering co-existence of rain and dry hydrometeors as an indicator of melting. A certain fraction of rain and dry snow is then converted to wet snow which shows intermediate properties between rain and dry snow, depending on the fraction of water within (wet fraction). As a first try to simulate the melting layer we have implemented Jung et al. (2008)'s method and adapted it to also consider wet graupel. However, two issues with this method have been observed. First of all the co-existence of liquid water and wet hydrometeors causes a secondary mode in the Doppler spectrum within the melting layer, due to the different terminal velocities, a mode that was never observed in the corresponding radar measurements. Secondly, the splitting of the total mass into separate hydrometeor classes (rain and wet hydrometeors) causes an unrealistic decrease in reflectivity just underneath the melting layer. It was thus decided to use an alternative parameterization in which only wet aggregates and wet graupel exist within the melting layer. At the bottom of the melting layer, where the wet fraction is usually almost equal to unity, these particle behave almost like rain and at the top of the melting layer, where the wet fraction is usually very small, these particles behave like their dry counterparts. Note that in contrary to Frick and Wernli (2012) which explicitly consider separate prognostic variables for the meltwater





on snowflakes, our scheme is purely diagnostic and is meant to be used in post-processing, when the COSMO model has been run without a parameterization for melting snow.

### 3.7.1 Mass concentrations of wet hydrometeors

The fraction of wet hydrometeor mass is obtained by converting the total mass of rain and dry hydrometeors within the melting layer into melting aggregates and melting graupel.

$$Q^{ms} = Q^s + \left( Q^r \frac{Q^s}{Q^s + Q^g} \right) \tag{17}$$

$$Q^{mg} = Q^g + \left( Q^r \frac{Q^g}{Q^s + Q^g} \right) \tag{18}$$

where the superscripts $s$, $g$ and $r$ indicate dry snow, dry graupel and rain, and $ms$ and $mg$ indicate wet snow and graupel. Note that the mass of rainwater is added to the mass of wet hydrometeors proportionally to their relative fractions.

The wet fraction within melting hydrometeors can be estimated by the fraction of mass coming from rainwater over the total mass:

$$f_{wet}^{ms} = \frac{Q^r Q^s}{Q^s \left( Q^s + Q^g \right) + Q^r Q^s} \tag{19}$$

$$f_{wet}^{mg} = \frac{Q^r Q^g}{Q^g \left( Q^s + Q^g \right) + Q^r Q^g} \tag{20}$$

### 3.7.2 Diameter dependent properties

*Mass*

For the mass of wet hydrometeors, the quadratic relation proposed by Jung et al. (2008) is used:

$$m^m(D) = (f_{wet}^m)^2 m^r(D) + \left[ 1 - (f_{wet}^m)^2 \right] m^d(D) \tag{21}$$

where the superscript $d$ indicates the corresponding dry hydrometeor and the superscript $m$ indicates the melting hydrometeor.

*Terminal velocity*

For the terminal velocity $v_t^m$ of melting hydrometeors, the equation is computed from the terminal velocities of rain and dry hydrometeors, using a best-fit obtained from wind tunnel observations by Mitra et al. (1990).

$$v_t^m(D) = \phi v_t^r(D) + (1 - \phi) v_t^d(D) \tag{22}$$





where $\phi = 0.246 f_{\text{wet}}^m + (1 - 0.246)\left(f_{\text{wet}}^m\right)^7$

This relation is also used by Frick and Wernli (2012) and Szyrmer and Zawadzki (1999).

*Canting angle distributions*

For the canting angle distributions, a linear shift of $\sigma_{\text{cant}}$ (the standard deviation of the Gaussian distribution of
canting angle) with $f_{\text{wet}}^m$ is considered:

$$\sigma_{\text{cant}}^m(D) = f_{\text{wet}}^m \sigma_{\text{cant}}^r(D) + (1 - f_{\text{wet}}^m)\sigma_{\text{cant}}^d(D) \tag{23}$$

*Aspect-ratio*

For a given diameter, the distribution of aspect-ratio for melting hydrometeors is the renormalized sum of the
gamma distribution of dry aspect-ratios obtained from the MASC observations (Equation 11) and the aspect-ratio
distribution of rain, linearly weighted by the melting fraction $f_{\text{wet}}^m$. Since for rain the aspect-ratio is considered
constant for a given diameter, the distribution would be a Dirac. Instead, in order to perform the weighted sum, the
distributions of aspect-ratios in rain are represented by a very narrow Gaussian distribution ($\sigma_{\text{a-r}}^r = 0.001$) centered
around the corresponding aspect-ratio.

*Permittivity*

To estimate the permittivity of these mixtures, the Maxwell-Garnett mixture model is used again. Melting hy-
drometeors are a mixture of three components: water, ice, and air. The volume fractions $V_f$ can again be estimated
with the mass-diameter model:

$$V_f^{\text{water}} = f_{\text{wet}}^m \frac{\rho^{\text{total}}}{\rho^{\text{water}}} \tag{24}$$

$$V_f^{\text{ice}} = \frac{\rho^{\text{total}} - V_f^{\text{water}}\rho^{\text{water}}}{\rho^{\text{ice}}} \tag{25}$$

$$V_f^{\text{air}} = 1 - V_f^{\text{water}} - V_f^{\text{ice}} \tag{26}$$

$$\tag{27}$$

Unfortunately the estimated permittivity will depend on whether water is treated as the matrix and snow as the
inclusions or the opposite, giving two different possible outcomes. To overcome this issue, a formulation proposed by
Meneghini and Liao (1996) is used, where the final permittivity is a weighted sum of both permittivities and where
the weights are function of the wet fraction. This method is also used by Ryzhkov et al. (2011).

### 3.7.3   Particle size distribution for melting hydrometeors

Once the mass concentrations and the wet fractions are known, it is possible to retrieve a particle size distribution
for melting hydrometeors. Szyrmer and Zawadzki (1999) suggest to match the flux of rainwater at every diameter:



$$N^r(D)v_t^r(D) = N^m(D)v_t^m(D) \quad \implies \quad N^m(D) = N^r(D)\frac{v_t^r(D)}{v_t^m(D)} \tag{28}$$

where $v_t$ is the hydrometeor terminal velocity.

In our model, this PSD is adjusted by multiplying it with a mass conservation factor $\kappa$ to ensure that the integral of the PSD weighted by the particle mass matches the concentrations of wet hydrometeors $Q^m$. Hence $N^{m,\mathrm{corr}}(D) = \kappa N^m(D)$ with:

$$\kappa = \frac{Q^m}{\int_{D_{\min}}^{D_{\max}} m(D)N^r(D)dD} \tag{29}$$

where $m^m(D)$ is the mass of a melting particle of diameter $D$ (Equation 21).

### 3.7.4 Integration scheme

Due to the sharp transition it causes in the simulated polarimetric variables, the melting layer effect causes major difficulties when integrating radar variables over the antenna power density. Indeed, the Gauss-Hermite quadrature scheme is appropriate only for continuous functions and will work well with a small number of quadrature points only for a relatively smooth function. Using a small number of quadrature points in the case of a melting layer was found to create unrealistic artifacts with the presence of several shifted melting layers of decreasing intensities. Globally increasing the number of quadrature points by a significant amount is not a viable solution since the computation time will increase linearly. Instead, the best compromise was found by increasing the number of quadrature points only at the edges of the melting layer, where the transitions are the strongest. In practice this is done by using ten times more quadrature points in the vertical than normally, but taking into account only the 10% of quadrature points with the highest weights for the computation of radar variables, except near the melting layer edges where all points are used.

### 3.8 Retrieval of Doppler velocities

#### 3.8.1 Average radial velocity

As illustrated in Figure 7, the average radial velocity $v_{\mathrm{rad}}$ is the power-weighted sum of the projections of $U$ (eastward wind component), $V$ (northward wind component), $W$ (vertical wind component), and $v_t$, the hydrometeor terminal velocity, onto the axis of the radar beam defined by elevation $\theta_0$ and azimuth $\phi_0$.

Estimating $v_{\mathrm{rad}}$ requires to know the terminal velocity of precipitating hydrometeors. In this work, we use the power-law relations prescribed by COSMO's microphysical parameterizations with parameters as given in Table 1.

It can be shown (e.g., Bringi and Chandrasekar, 2001) that, in the hypothesis of radial homogeneity inside a radar resolution volume, the average radial velocity at a given radar gate characterized by coordinates $r$ (range),




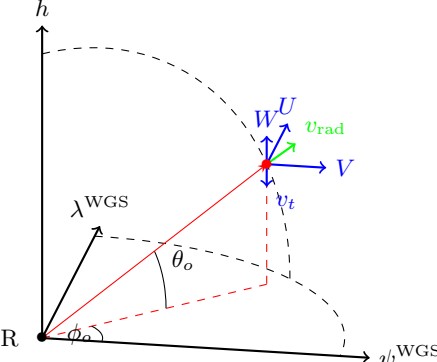

**Figure 7.** Trigononometric expression of the radial velocity as the power-weighted sum of the projection into the beam axis of the 3-dimensional wind field $(U, V, W)$ and the hydrometeor terminal velocity $v_t$.

$$v_{\mathrm{rad}}(r, \phi_o, \theta_o) = \frac{I\left[\sum_{j=1}^{H} \int_{D_{\min}^{(j)}}^{D_{\max}^{(j)}} v_{\mathrm{rad}}^{(j)}(D, r, \phi_o, \theta_o) \sigma_h^{(j)}(D) N^{(j)}(D)\, dD\right]}{\underbrace{I\left[\sum_{j=1}^{H} \int_{D_{\min}^{(j)}}^{D_{\max}^{(j)}} \sigma_h^{(j)}(D) N^{(j)}(D)\, dD\right]}_{\eta(r, \phi, \theta)}}$$

where

$$v_{\mathrm{rad}}^{(j)}(D, r, \phi, \theta) = [U(r, \phi, \theta) \sin\phi + V(r, \phi, \theta) \cos\phi] \cos\theta + \left[W(r, \phi, \theta) - v_t^{(j)}(D)\right] \sin\theta$$

$$(30)$$

$\phi$ (azimuth) and $\theta$ (elevation) is given by Equation 30, where $\sigma_h^{b,(j)}(D)$ is the backscattering radar cross-section at horizontal polarizations for an hydrometeor of type $j$ and diameter $D$,

where $I$ is the quadrature antenna integration operator defined in Equation 6.

### 3.9 Doppler spectrum

5 In this section we propose a simple scheme able to compute the Doppler spectrum at any incidence at a very small computational cost (less than 10% of the total cost). Unlike Cheong et al. (2008), this approach is not based on sampling and is thus deterministic, but the computational cost is much smaller.



Using the specified hydrometeor terminal velocity relations, it is possible to not only compute the average radial velocity, but also the Doppler spectrum: the power weighted distribution of scatterer radial velocities within the radar resolution volume.

This is done by first computing the resolved velocity classes of the Doppler spectrum $v_{r,\text{bins}}[i]$, for every bin $i$,

based on the specified radar FFT window length $N_{\text{FFT}}$ and Nyquist velocity $v_{\text{Nyq}}$.

$$v_{\text{rad,bins}}[i] = (i - \frac{N_{\text{FFT}}}{2})\frac{v_{\text{nyq}}}{N_{\text{FFT}}} \quad \forall i = -\frac{N}{2}, ..., \frac{N}{2} \tag{31}$$

where $v_{\text{Nyq}}$ is the Nyquist velocity, in m s$-1$, given by

$$v_{\text{Nyq}} = 100\frac{PRF \cdot \lambda}{2} \tag{32}$$

where $\lambda$ is the radar wavelength in cm.

For every hydrometeor $j$ and every velocity bin $i$, given the three-dimensional wind components $(U, V, W)$, one can estimate the hydrometeor terminal velocity $v_t$ that would be needed to yield the radial velocity $v_{\text{rad,edges}}[i]$:

$$v_t^{(j)}(r,\phi',\theta')[i] = W(r,\phi',\theta') + \frac{U(r,\phi',\theta')\sin\phi' + V(r,\phi',\theta')\cos\phi'}{\tan\theta'} - \frac{v_{\text{rad,bins}}[i]}{\sin\theta'} \tag{33}$$

Once this is done, the corresponding diameters $D^{(j)}[i]$ can be retrieved by inverting the diameter-velocity power-laws (see Table 1). Finally, for a given radar gate defined by coordinates $(r_o,\phi_o,\theta_o)$ the Doppler spectrum $S$ in linear

$Z_e$ units (mm$^6$ m$^{-3}$), for a given velocity bin $i$ is

$$S(r_o,\phi_o,\theta_o)[i] = I\left[\sum_{j=1}^{H}\int_{D^{(j)}[i+1]}^{D^{(j)}[i]} \sigma_h^{b,(j)}(D)N^{(j)}(D) \; dD\right] \tag{34}$$

Any statistical moment can then be computed from this spectrum. The average radial velocity, for example is simply the first moment of the Doppler spectrum:

$$v_{\text{rad}}(r_o,\phi_o,\theta_o) = \frac{\sum_{i=0}^{N} v_{\text{rad,bins}}[i]S(r,\phi,\theta)[i]}{\sum_{i=0}^{N} S(r_o,\phi_o,\theta_o)[i]} \tag{35}$$

## 3.10 Turbulence and antenna motion correction

The standard deviation of the Doppler spectrum, often referred to as the spectral width, is a function of both radar system parameters and meteorological parameters that describe the distribution of hydrometeor density and velocity within the sampling volume (Doviak and Zrnić, 2006). Assuming independence of the spectral broadening



mechanisms, the square of the velocity spectrum width $\sigma_v^2$ (i.e. standard deviation of the spectrum) can be considered as the sum of all contributions (Doviak and Zrnić, 2006).

$$\sigma_v^2 = \sigma_s^2 + \sigma_\alpha^2 + \sigma_d^2 + \sigma_o^2 + \sigma_t^2 \tag{36}$$

where $\sigma_s^2$ is due to the wind shear, $\sigma_\alpha^2$ to the rotation of the radar antenna, $\sigma_d^2$ to variations in hydrometeor
terminal velocities, $\sigma_o^2$ to changes in orientations or vibration of hydrometeors and $\sigma_t^2$ to turbulence.

In the forward radar operator, $\sigma_s^2$ is already taken into account by the integration scheme, $\sigma_d^2$ by the use of the diameter-velocity relations and $\sigma_o^2$ by the integration of the scattering properties over distributions of canting angles. Thus, the spectrum computed in Equation 3.9 needs to be corrected only for turbulence and antenna motion. Doviak and Zrnić (2006) gives the following estimation for $\sigma_\alpha$.

$$\sigma_\alpha = \left( \frac{\omega \lambda \cos \theta_o}{2\pi \Delta_{3\mathrm{dB}}} \right) \sqrt{\ln 2} \tag{37}$$

where $\omega$ is the angular velocity (in rad s$^{-1}$). Note that $\sigma_\alpha$ is equal to zero at vertical incidence, which is the most common configuration for Doppler spectrum retrievals.

For $\sigma_t$, Doviak and Zrnić (2006) gives the following estimation, originally derived by Labitt (1981), which is based on the hypothesis of isotropic and homogeneous turbulence, with all contributions to turbulence coming from the
inertial subrange.

$$\sigma_{\mathrm{t}} = \begin{cases} \left[ \dfrac{\epsilon_t r_o \left( 1.35B \right)^{3/2}}{0.72} \right]^{1/3} & \text{if } \sigma_r \ll r\sigma_\theta \\[2em] \left[ \dfrac{\epsilon_t \sigma_r \left( 1.35B \right)^{3/2}}{\left[ \frac{11}{15} + \frac{4}{15} (r^2 \sigma_{\theta_o}^2 \sigma_r^{-2} \right]^{-3/2}} \right]^{1/3} & \text{else} \end{cases} \tag{38}$$

where $B$ is a constant between 1.53 and 1.68[3] and $\epsilon_t$ is the eddy dissipation rate (EDR) expressed in units of m$^2$s$^{-3}$. $\epsilon_t$ is the rate at which turbulent kinetic energy is converted into thermal internal energy. It is a model variable, simulated by the turbulence parameterization and can be obtained as any other variable used in the radar
operator, by downscaling to the radar gates. Finally $\sigma_r$ and
$sigma_\theta$ depend on the radar specifications: $\sigma_r = 0.35 c\tau/2$ ($\tau$ is the pulse duration in s) and $\sigma_\theta = \Delta_{3\mathrm{dB}}/4\sqrt{\ln(2)}$.

This makes it possible to estimate both $\sigma_o$ and $\sigma_t$ using the specified radar system parameters and simulated turbulence variables. If one assumes the spectral broadening caused by the antenna motion and turbulence to be Gaussian with zero mean (e.g., Babb et al., 2000; Kang, 2008), the corrected spectrum can be obtained by convolution
with the corresponding Gaussian kernel.

---

[3]A constant value of 1.6 is used in the radar operator.



$$S^{\mathrm{corr}}[i] = \sum_{j=0}^{N_{\mathrm{FFT}}} S[i-j] \frac{1}{\sqrt{(2\pi\sigma_{t+\alpha})^2}} \exp\left[-\frac{v_{r,\mathrm{bins}}[i]}{2\sigma_{t+\alpha}^2}\right] \tag{39}$$

where $\sigma_{t+\alpha} = \sigma_t + \sigma_\alpha$

### 3.11 Attenuation correction in the Doppler spectrum

In reality, attenuation will cause a decrease in observed radar reflectivities at all velocity bins within the spectrum.
To take into account this effect, the total path integrated attenuation at a given radar gate is distributed uniformly
throughout the spectrum with the constraint, however, that attenuated reflectivities should never become negative.
The total path-integrated attenuation in linear $Z_e$ units is given by (c.f. Equations C2):

$$PIA(r_0) = Z_h^{\mathrm{att}}(r_0) \left[\int_{r=0}^{r_0} 10^{0.1 \cdot 2 \cdot k_H(r,\theta_o,\phi_o)} dr\right] \tag{40}$$

where $k_H$ is the specific attenuation in dB km$^{-1}$ as defined in Appendix C. Note the factor 2, which occurs because
attenuation affects both the emitted and received signal.

To distribute $PIA(r_0)$ within the Doppler spectrum, the attenuation at every bin is considered to be equal to a
constant value $K$, expressed in linear $Z_e$ units, except for bins where $K > S[i]$. Hence the attenuated reflectivities
at every bin $i$ are equal to:

$$S^{\mathrm{att}}[i] = S[i] - \min(S[i], K) \tag{41}$$

$\gamma$ is the maximum attenuation at every velocity bin and can be related to the $PIA$ by summing the attenuation
at all bins.

$$PIA(r_0) = \sum_{i=0}^{N_{\mathrm{FFT}}} \min(S[i], K) \tag{42}$$

$K$ can easily be estimated by finding the root of Equation 42 with a numerical solver (Newton-Raphson for
example). The related computational cost is negligible (less than 1% of the total computation time).

### 3.12 Simulation of satellite swaths

The radar operator was adapted to be able to simulate swaths from spaceborne radar systems such as the GPM
dual-frequency radar (Iguchi et al., 2003) at both Ku and Ka bands. The main modifications to the standard routine
concern the beam tracing, which is estimated from the GPM data (in HDF5 format) by using the WGS84 coordinates





at the ground and the radar position in Earth-centered Earth-fixed coordinates to retrieve the coordinates of every radar gate. Currently, the atmospheric refraction is neglected which leads to an average positioning error of 55 m, the error being minimal at the center of the swath (where the incidence angle is nearly vertical) and maximal at the edges of the swath. The integration scheme remains unchanged and a fixed beamwidth of 0.5 ° is used according to

GPM specifications. An important advantage of simulating satellite radar measurements over simply comparing the precipitation intensities at the ground, is that it allows a three-dimensional evaluation of the model data.

### 3.13    Computation time

Though being mostly written in Python, the forward radar operator was optimized for speed as all computations are parallelized and its most time consuming routines are implemented in C. In addition, the scattering properties

of individual hydrometeors are pre-computed and stored in lookup tables. Table 4 gives some indication of the computation times encountered for different types of simulated scans. The RHI scan consists of 150 different elevations in the main direction of the precipitation system, with a maximal range of 20 km and a radial resolution of 75 m. The melting layer is simulated with the quadrature oversampling scheme described in Section 3.7.4. The RHI scan was also computed with the full Doppler scheme (Section 3.9). The PPI scan consists of 360 different azimuth angles at

1° elevation at C-band, with a maximal range of 150 km and a radial resolution of 500 m. All scans were performed in a stratiform rain situation (8 April 2014 for ground radars and 4 April 2014 for GPM), with a wide precipitation coverage. The advanced refraction scheme by Zeng et al. (2014) was used for all scans except the GPM swath. To integrate over the antenna density pattern 3 quadrature points in the horizontal and 5 in the vertical were used for all scans (with an oversampling factor of 10 at the ML edges).

The computation times are usually reasonable even on a standard desktop computer, except when simulating the melting layer effect on a PPI scan at low elevation. However, it can be seen that the forward radar operator scales very well with increasing number of computation power and nodes, since the computation time decreases more or less linearly with increasing computer performance.

|         | RHI, with ML and spectrum | PPI, no ML | PPI, with ML | GPM Ku band, no ML |
|---------|---------------------------|------------|--------------|--------------------|
| Desktop | 2.1 min                   | 5.3 min.   | 11.1 min.    | 8.9 min.           |
| Server  | 1 min.                    | 2.1 min.   | 6.16 min.    | 5.3 min.           |

**Table 4.** Observed computation times for three types of scans and two computers. The desktop has an 8 cores i7-4770S CPU with 3.1 GHz (30.5 GFlops/s) and 32 GB of RAM, the server has a 12 cores i7-3930K with 3.20GHz (59 GFlops/s) and 32 GB of RAM





## 4   Evaluation of the operator

In this section, a comparison of simulated radar fields with radar observations is performed. It is important to realize that discrepancies between measured and simulated radar variables can be caused both by:

1. The inherent inexactitude of the model which manifests itself by differences in magnitude as well as temporal and spatial shifts in the simulated state of the atmosphere, compared with the real state of the atmosphere.

2. Limitations of the forward radar operator, e.g. imperfect assumptions on hydrometeor shapes, density and permittivity, inaccuracies due to numerical integration, non-consideration of multiple scattering effects.

When validating the radar operator, only the second factor is of interest but as the discrepancies are often dominated by the first factor, validation becomes a difficult task.

Hence, for evaluation purposes, it is important to run the model in its best configuration, in order to limit as much as possible its inaccuracy. This is is why the model was run in analysis mode, with a 12 hours spin-up time, using analysis runs of the coarser COSMO-7 (7 km resolution) as input and boundary condition. Note that even though COSMO has recently become operational at a resolution of 1km over Switzerland, the simulations performed in this work were still done at a 2km resolution. Note that the present evaluation was done with the standard one-moment scheme, for sake of simplicity, but Appendix B gives some additional indications and results for the two-moments scheme.

Evaluation of the radar operator was first done by visual inspection on a time step basis and was followed by a more quantitative evaluation over the course of the whole precipitation events.

### 4.1   Qualitative comparisons

#### 4.1.1   PPI scans at C-band

Figures 8 and 9 show two examples of simulated and observed PPI scans from the La Dôle radar in western Switzerland at 1° elevation during one mostly convective event (13 August 2015) and one mostly stratiform event (8 April 2014). The displayed radar reflectivites are raw uncorrected ones, and the attenuation effect is taken into account for simulated reflectivities. It can be seen that in both cases, the model is able to locate the center of the precipitation event quite accurately but tends to overestimate its extent, especially in the convective case. Generally, the simulated $Z_{\mathrm{H}}$, $Z_{\mathrm{DR}}$ and $K_{\mathrm{dp}}$ are of the same order of magnitude as the observed ones, with the exception of the stratiform case, where the simulated $K_{\mathrm{dp}}$ is underestimated on the edges of the precipitating system. The simulated radial velocities seem very realistic and agree well with observations both in terms of amplitude and spatial structure.





### 4.1.2 RHI with melting layer at X-band

Figure 10 shows one example of simulated and observed RHI scan in a stratiform situation (22 March 2014) with a clearly visible melting layer at low altitude. It can be seen that the forward radar operator is indeed able to simulate a realistic polarimetric signature within the melting layer with a clearly visible bright-band in $Z_\mathrm{H}$, an increase in $Z_\mathrm{DR}$ followed by a sharp decrease in the solid phase above and higher values of $K_\mathrm{dp}$. The extent of the melting layer seems also to be quite accurate when compared with radar measurements. Note that, in this case, the model slightly overestimates the signature in $Z_\mathrm{DR}$ and $Z_\mathrm{H}$ below the melting layer, but this is related to the fact that COSMO tends to overestimate the rain intensity during this particular event. In terms of radial velocities, again the model simulates some very realistic patterns that agree well with the observations, with two shear transitions at around 1 and 3.5 km altitude followed by a strong increase in velocities at higher altitudes.

### 4.1.3 GPM swath

Figure 11 shows an example of simulated and measured GPM swath at Ku band at different altitudes. Again the forward radar operator produces a realistic horizontal and vertical structure as well as plausible values of reflectivities, given the fact that in this case the simulated average rain rate is lower than the GPM estimated average rain rate (0.38 mm s$^{-1}$ vs 0.46 mm s$^{-1}$).

### 4.2 Doppler variables

Evaluation of the simulated average radial velocities was performed by comparison of simulated velocities with observations from the MXPol X-band radar deployed in Payerne in Western Switzerland in Spring 2014 in the context of the PARADISO measurement campaign.

A total of 720 RHI scans (from 0 to 180° elevation) were simulated over six days of mostly stratiform precipitation (c.f. Table 3). Figure 12 shows a comparison of the distributions of radial velocities between the simulation and the radar observations. The scatter-plot in Figure 13 shows the excellent overall agreement when considering all events and scans. Simulations match very well observations, both in terms of distributions and in terms of one-to-one relations, which shows that the radar operator is indeed able to simulate accurate radial velocities. Since wind observations from the radiosoundings performed in Payerne are assimilated into the model, one can expect it to perform well in this regard. These results indeed confirm these expectations.

During the PARADISO campaign, MXPol was also retrieving the Doppler spectrum at vertical incidence, which allows to compare simulated spectra with real measurements. Figure 14 shows the daily averaged simulated and measured Doppler spectra during the same six days of precipitation. Generally, the simulated spectrum is able to reproduce the transition from high velocities near the ground (in liquid precipitation) to smaller velocities in altitude (solid precipitation). The height of this transition, which corresponds roughly to the isotherm 0°, as well as the simulated velocities above and below the isotherm 0° agree quite well with the observations. Thanks to the




melting layer scheme, the operator is able to produce a quite realistic transition between solid and liquid phase. Indeed, when the melting scheme is disabled, the simulated Doppler spectra show a very abrupt and unrealistic transition in velocities. In terms of reflectivity, the bright-band effect is clearly visible on the simulated spectra and its magnitude relative to the reflectivites below and above the melting layer agrees well with observations. In absolute terms however, some events show a good agreement (22 March 2014, 7 May 2014), while in others, the simulated reflectivites tend to be ovestimated over the whole spectrum (8 April 2014, 14 May 2014, 1st May 2014). We think however that these discrepancies are mostly caused by the larger precipitation intensities simulated by the model during these days. Precipitation measurements with a rain gauge collocated with the radar tend to confirm this hypothesis. For the two events with the strongest discrepancies (1st May and 14 May), the gauge measured in total 1.9 and 1.2 mm of precipitation, whereas the model simulated 16.9 and 2.1 mm of precipitation in the closest grid cell.

### 4.3 Polarimetric variables

Evaluation of polarimetric variables ($Z_{\mathrm{H}}$, $Z_{\mathrm{DR}}$ and $K_{\mathrm{dp}}$) is difficult, because their agreement with radar observations depends heavily on the temporal and spatial accuracy of simulated precipitation fields. However, when averaging over a sufficiently large number of samples, the radar operator should at least be able to simulate realistic distributions of polarimetric variables, as well as realistic relations between these polarimetric variable. Augros et al. (2016) for example, validated their operator, inter alia, by comparing simulated and observed membership functions between the polarimetric functions.

In order to test the quality of the simulated polarimetric variables, five events corresponding to different synoptic situations with widespread precipitation over Switzerland were selected (Table 3). The simulated polarimetric variables were compared with observations from three operational C-band radars (La Dôle, Albis and Monte Lema).

The duration of all events ranges between 12 and 24 hours with a resolution in time of 5 minutes (which corresponds to the temporal resolution of the available radar data). A total of 1017 PPI scans were simulated at 1° elevation with a maximum range of 100 km (in order to limit the effect of beam-broadening). Both observed and simulated radar data were censored with an $SNR_{\mathrm{thr}}$ value of 8 dB (Equation 16).

The *shape* parameter of the gamma DSD used in COSMO for rain has a strong influence on the outcome of the radar operator. Indeed, the skewness of the gamma distribution is inversely proportional to $\mu^{\mathrm{rain}}$, so DSDs with small values of $\mu^{\mathrm{rain}}$ will have longer right tails. This is of particular importance when simulating polarimetric variables that are related to statistical moments of a high order, such as $Z_{\mathrm{DR}}$. Two values of $\mu^{\mathrm{rain}}$ have been tested, $\mu^{\mathrm{rain}} = 0.5$, which is the default value in the model and $\mu^{\mathrm{rain}} = 2$ which corresponds to the upper range of recommended values in the model.

The comparison between simulated and observed radar variables was performed separately in the liquid and solid phases. Indeed, the uncertainty in the liquid phase is expected to be lower than in the ice phase because the scattering properties of raindrops are more reliable than in snowfall. The simulated model temperatures were taken





as a criterion to separate the phases; the liquid phase corresponds to $T > 5°$ and the solid phase to $T < -5°$ as in Augros et al. (2016). Areas with temperatures in between have been ignored in order to limit the contribution of wet snow which is not directly simulated by COSMO. It was observed that increasing the temperature margin between liquid and solid phases did not change significantly the main results and conclusions. Decreasing it, however, would

affect quite significantly the observed radar signatures due to the inclusion of measurements from the melting layer, which have a much stronger polarimetric signature than dry snow.

Figure 15 shows the corresponding histograms of observed and simulated polarimetric variables and precipitation intensities at the ground in the liquid phase, for $\mu^{\mathrm{rain}} = 2$. The histograms for $\mu^{\mathrm{rain}} = 0.5$ (not displayed) show only minor differences. The simulated distributions agree well with the observed ones in terms of broad features, which

confirms the fact that the operator is able to simulate realistic radar observables at least in liquid phase. One can observe that the radar operator is not able to simulate negative $Z_{\mathrm{DR}}$, which can be explained by the assumptions about the drop shapes and orientations, which make it almost impossible for a drop to have a vertical dimension larger than its horizontal dimension. In addition, the radar operator seems to produce slightly smaller values of $Z_{\mathrm{H}}$ than observed, but this can be attributed to the fact that COSMO tends to simulate smaller precipitation intensities

than the ones estimated from the radar reflectivities (bottom-right of Figure 15). Indeed, the discrepancies in $Z_{\mathrm{H}}$ agree well with the discrepancies in precipitation intensities.

Figure 16 shows the observed (from MeteoSwiss radars) and the simulated $Z_{\mathrm{H}} - Z_{\mathrm{DR}}$ and $Z_{\mathrm{H}} - K_{\mathrm{dp}}$ relations averaged over all radars and all events in the liquid and solid phases. It appears that the radar operator is able to simulate realistic relations between polarimetric variables at least in the liquid phase. In terms of $Z_{\mathrm{DR}}$, a value of

$\mu^{\mathrm{rain}} = 2$ seems more appropriate than a value of 0.5, which tends to overestimate the differential reflectivity for a given horizontal reflectivity. For $K_{\mathrm{dp}}$ the trend is reversed. A possible explanation is that $Z_{\mathrm{DR}}$ is independent of the concentration and highly dependent on the length of the DSD tail, i.e. small differences in the numbers of large and oblate drops can cause large differences in differential reflectivity. $K_{\mathrm{dp}}$ however, depends on both the total concentration and the tail of the DSD, and is quite sensitive to the mode of the DSD. However, one must also keep

in mind that the "observed" $K_{\mathrm{dp}}$ values are in fact estimated from noisy $\Psi_{\mathrm{dp}}$ measurements and as such are likely to be underestimated (Grazioli et al., 2014). This dependency of simulated polarimetric variables on small changes in the DSD shape illustrates quite well the difficulty to parameterize the DSDs to match both the lower order moments used in weather prediction (number and mass concentration) and the higher order moments, to which the radar observables are related.

In the solid phase, the radar operator tends to underestimate $Z_{\mathrm{DR}}$ and $K_{\mathrm{dp}}$, which is a trend also observed by Augros et al. (2016). This is likely due to the combination of the imperfect parameterization of snow PSD in the model, the crude assumptions about the permittivity of snow and graupel (mixture model derived from the COSMO density parameterizations), and the estimation of the scattering properties (T-matrix is likely not correct for ice-phase hydrometeors).





## 4.4    Comparison of the COSMO rain DSDs with ground measurements

In order to further investigate these surprisingly large discrepancies in the distributions of polarimetric variables between the different COSMO rain DSD parameterizations, a comparison with ground measurements from three Parsivel disdrometer was performed. The same events used for the Doppler evaluation were used: six events over

Payerne in Switzerland dominated by stratiform rainfall. The COSMO DSDs were obtained at the lowest model level, on the grid cell comprising all three Parsivels.

Figure 17 shows a comparison of the average measured rain DSD and the COSMO parameterized DSDs over the six days of precipitation. It is obvious that the COSMOS DSDs with $\mu^{\mathrm{rain}} = 0.5$ tends to produce too many small drops when compared with the Parsivel data. However one must keep in mind that due to the instrument's

limitations, the Parsivel, as most disdrometers, has difficulty to measure very small drops and might underestimate their numbers (Thurai et al., 2017). However, one can still observe with certitude that the mode of the COSMO parameterized DSDs is located too much on the left, especially for $\mu^{\mathrm{rain}} = 0.5$. When fitting a gamma DSD on the measured data, the optimal value of $\mu^{\mathrm{rain}}$ is around 3.4, which indicates that the match with the real radar observations could possibly be even better by increasing even more the value of $\mu^{\mathrm{rain}}$.

## 4.5    GPM swaths

In order to evaluate the simulation of GPM swaths, the distributions of simulated and observed reflectivities at both Ku and Ka band were compared for 100 GPM overpasses over Switzerland, corresponding to the overpasses with the largest precipitation fluxes (c.f. Section 2.4).

Figure 18 shows the overall distributions of reflectivity at both frequency bands as well as the distributions of

estimated GPM precipitation intensities and COSMO simulated intensities. Note that all reflectivities below 14 dBZ have been discarded as this corresponds roughly to the radar sensitivities at Ka and Ku band (Toyoshima et al., 2015). Although the distributions are very consistent, some minor discrepancies are present, mostly for low reflectivities (at Ka band only) and high reflectivities which appear more frequently in the simulations than in the measurements from the GPM-DPR. Again, this is consistent with the differences in simulated precipitation intensities (in panel **c**).

COSMO tends to produce a larger number of precipitation intensities $\geq 30$ mm hr$^{-1}$ as well as a larger number of precipitation intensities below 0.15 mm hr$^{-1}$ which corresponds roughly to 14 dBZ. In addition, comparison of GPM measurements with ground radar observations confirms that GPM tends to underestimate larger reflectivities Speirs et al. (2017). Overall, the simulated distributions of reflectivity at both frequency bands are realistic and agree quite well with the observations for both microphysical scheme. Note that when neglecting ice crystals the match is much

poorer (see Section 4.6).





## 4.6 Effect of ice crystals

In order to evaluate the addition of ice crystals to the forward operator, a two-fold analysis was performed. First, the simulated polarimetric variables obtained with and without considering ice crystals were compared with real observations by MXPol during three pure snowfall events in the Swiss Alps in Davos (Table 3). Since no liquid

precipitation or melting layer was present during these events, the attenuation effect is expected to be negligible. Note that the analysis focused on the one-moment scheme but the effect on the two-moment scheme is expected to be quite similar. Figure 19 shows a comparison of the distributions of polarimetric variables in the solid phase averaged over all three events for the one-moment microphysical scheme. On $Z_{\mathrm{H}}$, the effect of adding ice crystals is characterized by an additional mode around 8 dBZ, which is not present on radar observations. This mode is caused

by the large homogeneity in the simulated ice crystals, which, according to the microphysical parameterization, are all assumed to be hexagonal plates. In reality, ice crystals can have a large variability of shapes (e.g., Magono and Lee, 1966; Bailey and Hallett, 2009), and their backscattering coefficients can be quite different (Liu, 2008), which would result in a much more spread out reflectivity signature of ice crystals. On $Z_{\mathrm{DR}}$, one can see that, when neglecting ice crystals, one completely removes the right tail of the distribution (values above 0.2 dBZ) that is clearly visible on the

observed values. When considering ice crystals, which have a quite strong signature in differential reflectivity, this right tail gets accurately reproduced and matches well with the observations. However, even when adding ice crystals, the radar operator is not able to reproduce the negative $Z_{\mathrm{DR}}$ values that are quite frequent in the observations. On $K_{\mathrm{dp}}$, a similar effect can be observed, though not as clear. Still, the addition of ice crystals creates an additional mode in the distribution of simulated values which slightly better matches with the observed one (longer tail and

good agreement of the additional mode with the mode of the observed distribution). Just as with $Z_{\mathrm{DR}}$, the radar operator is not really able to simulate negative values of $K_{\mathrm{dp}}$, which are also frequent in the observations. These discrepancies could however also be due in part to uncertainties in the radar observations, coming from possible miscalibration (for $Z_{\mathrm{DR}}$) and inaccuracies in the retrieved $K_{\mathrm{dp}}$ values. Still, overall at X-band, the addition of ice crystals leads to a much better representation of $Z_{\mathrm{DR}}$ in solid precipitation, a slightly better representation of $K_{\mathrm{dp}}$

and no significant improvement in $Z_{\mathrm{H}}$.

Due to their smaller sizes, the effect of ice crystals on $Z_{\mathrm{H}}$ should increase with the frequency. To investigate this effect, a second comparison was performed on the simulation of GPM swaths, with and without ice crystals. The resulting distributions of $Z_{\mathrm{H}}$ at Ku and Ka band were compared with means of QQ-plots of observed versus simulated quantiles. Figure 20 shows these QQ-plots at Ka band for both the one-moment and the two-moments

scheme. The red line is the 1:1 which implies a perfect match with the observed quantiles. The results at Ku band are not displayed as they are visually very similar to the results at Ka band. For the one-moment scheme, a much better agreement with observations is observed for small quantiles (up to 20 dBZ) when adding ice crystals. Without ice crystals, small quantiles tend to be underestimated. Large simulated quantiles tend to be overestimated when compared with GPM observations. For very large quantiles, this overestimation is slightly stronger when adding ice





crystals but this might be a sampling effect as large quantiles are very sensitive to outliers. For the two-moments scheme, adding ice crystals does not seem to significantly improve the agreement with observed quantiles.

As a conclusion, adding ice crystals improves the quality of the simulated $Z_{DR}$ and $K_{dp}$ in pure solid precipitation at X-band and when simulating horizontal reflectivities at K band.

## 5   Conclusions

In this work we propose a new polarimetric radar forward operator for the COSMO NWP model which is able to simulate measurements of reflectivity at horizontal polarization, differential reflectivity and specific differential phase shift on propagation for ground based or spaceborne (e.g. GPM) radar scans, while taking into account most physical effects affecting the propagation of the radar beam (atmospheric refractivity, beam-broadening, partial beam-blocking

and attenuation). Integration over the antenna pattern is done with a simple Gauss-Hermite quadrature scheme. This scheme was compared with more advanced schemes that also take into account antenna side lobes, but was shown to offer on average the best trade-off, due to its better representation of the main lobe and lower computational cost. The operator was extended with a new Doppler scheme, which allows to efficiently estimate the full Doppler spectrum, by taking into account all factors affecting the spectral width (antenna rotation, turbulence, wind shear

and attenuation), as well as a melting layer scheme able to reproduce the very specific polarimetric signature of melting hydrometeors, even though the COSMO model does not explicitly simulate them. Finally, the operator was adapted both to the operational one-moment microphysical scheme of COSMO and to its more advanced two-moment scheme. Performance tests showed that the operator is sufficiently fast and efficient to be run on a simple desktop computer.

The scattering properties of individual hydrometeors are pre-computed with the T-matrix method and stored into lookup tables for various frequencies. The permittivities for the complex hydrometeors (snowflakes, hail and graupel) are obtained with a mixture model by using the mass-diameter relations of COSMO to estimate their densities. The other required parameters for the T-matrix method (canting angle distributions and aspect-ratios) are obtained from the literature (for rain, hail and ice crystals) and from measurements performed in the Swiss Alps with a multi-angle

snowflake camera (MASC), for snow and graupel. A large number of MASC pictures were used to estimate realistic parameterizations of the distributions of aspect-ratio and canting angle of graupels and aggregates, leading to a good agreement with measured quantiles. Integration of the hydrometeors scattering properties over these distributions was shown to increase the polarimetric signature of solid hydrometeors, which tends to be often underestimated in radar operators.

The operator was evaluated by a comparison of the simulated fields of radar observables with observations from the operational Swiss radar network, from a high resolution X-band research radar and from GPM swaths. Visual comparisons between simulated and measured polarimetric variables showed that the operator is indeed able to simulate realistic looking fields of radar observables both in terms of spatial structure and intensity and to simulate




a realistic melting layer both in terms of thickness and polarimetric signature. Comparisons of the radial velocities measured by the X-band radar and simulated by the radar operator, in the vicinity of the Payerne radiosounding site showed an excellent agreement with a high determination coefficient. The operator was also able to simulate realistic Doppler spectra at vertical incidence, with realistic fall velocities and reflectivites below and above the
melting layer, as well as within the melting layer, thanks to the melting scheme. A comparison of the distributions of polarimetric variables as well as the relations between these variables with measurements from the Swiss operational C-band radar network was performed. In the liquid phase, the radar operator is generally able to simulate realistic distributions of polarimetric variables and realistic relations between them. A comparison with measurements from Parsivel disdrometers revealed that the agreement between simulated and observed polarimetric variables depends
strongly on the shape parameter used in the drop size distribution of raindrops.

In the solid phase, however, the polarimetric variables tend to be underestimated when using the T-matrix method to simulate hydrometeor scattering properties, even with the local MASC parameterization. Finally the effect of considering or not ice crystals in the simulation was investigated and it was observed that at X-band the agreement with observed differential reflectivity and differential phase shift improves significantly, whereas at GPM frequencies,
the simulated distributions of reflectivity are more realistic, especially for smaller reflectivities.

Ultimately, this operator provides a convenient way to relate outputs of a NWP model (state of the atmosphere, precipitation) to polarimetric radar measurements. The evaluation of the operator has shown that this tool is a promising way to test the validity of some of the hypothesis of the microphysical parameterization of COSMO. Future work will focus on a detailed sensitivity analysis of the main parameters and assumptions of the radar
operator will be performed, taking again a large dataset of radar observations as reference. In the liquid phase, the analysis should focus on the geometry of raindrops as well as the parameterization of the DSD. In the ice phase, the potential benefit of using more sophisticated methods to estimate the scattering properties of solid hydrometeors will be investigated.

*Code availability.* The radar operator code is available at https://github.com/wolfidan/cosmo_pol

**Appendix A: Trilinear downscaling**

Downscaling is computationally faster if the radar gate coordinates are first converted from the World Geodetic System 1984 (WGS) lat/lon coordinates to the local pole-rotated model coordinates, where the model variables are defined on a regular grid. To this end, the spherical WGS coordinates of the radar gate ($\psi^{\mathrm{WGS}} = \mathrm{lon}$, $\lambda^{\mathrm{WGS}} = \mathrm{lat}$) are first projected to Earth-centered,earth-fixed (ECEF) coordinates $(x, y, z)$ and then rotated to the pole-rotated
system using two rotations matrices, one for the longitudinal rotation of the pole $\Delta_{\lambda^{\mathrm{WGS}}}$, and one for the latitudinal rotation of the pole $\Delta_{\psi^{\mathrm{WGS}}}$, to yield $(x^{\mathrm{rot}}, y^{\mathrm{rot}}, z^{\mathrm{rot}})$.





$$
\begin{pmatrix} x_m \\ y_m \\ z_m \end{pmatrix} = \begin{pmatrix} \cos\Delta_{\lambda^{\text{WGS}}} & \sin\Delta_{\lambda^{\text{WGS}}} & 0 \\ -\cos\Delta_{\lambda^{\text{WGS}}} & \cos\Delta_{\lambda^{\text{WGS}}} & 0 \\ 0 & 0 & 1 \end{pmatrix} \begin{pmatrix} \cos\Delta_{\psi^{\text{WGS}}} & 0 & \sin\Delta_{\psi^{\text{WGS}}} \\ 0 & 1 & 0 \\ -\sin\Delta_{\psi^{\text{WGS}}} & 0 & 1 \end{pmatrix} \begin{pmatrix} x \\ y \\ z \end{pmatrix}
\tag{A1}
$$

Finally, the Cartesian coordinates $(x_m, y_m, z_m)$ in the model pole-rotated system, are projected back to spherical coordinates to yield $(\psi^{\text{rot}}, \lambda^{\text{rot}})$, the spherical coordinates of radar gates in the model pole-rotated system.

For every radar gate, the eight neighbor model nodes can efficiently be identified by direct mapping of the $(\psi^{\text{rot}}, \lambda^{\text{rot}})$ coordinates (which as stated are on a regular grid) and by binary search through all vertical model levels. Once the neighbors have been identified (Figure A1), downscaling is done by first linearly interpolating all neighbors with identical $(\psi^{\text{rot}}, \lambda^{\text{rot}})$ to the height $z$ of the radar gate: $(A_u, A_l) \rightarrow A^\star$, $(B_u, B_l) \rightarrow B^\star$, $(C_u, C_l) \rightarrow A^\star$, $(D_u, D_l) \rightarrow D^\star$ . The resulting points $(A_\star, B_\star, C_\star, D_\star)$ are then bilinearly interpolated to the horizontal location of

the radar gate.

## Appendix B: Specificities of the two-moments scheme

In the two-moment scheme all prescribed PSDs are initially defined as a function of particle mass.

$$
N_m(x) = N_{0,m} x^{\mu_m} \exp(-\Lambda_m x^{\nu_m})
\tag{B1}
$$

where the subscript $m$ denotes that the quantity is mass-based and $N_m(x)$ is in units of $\text{kg}^{-1}\text{m}^{-3}$.

However in the context of this radar operator, it is much more convenient to work with diameter-based PSDs. This conversion can be done by using the prescribed mass-diameter relations which are part of the microphysical scheme: $D(x) = a_m x^{b_m} \Rightarrow x = \frac{D}{a_m}^{\frac{1}{b_m}}$ and by considering that $N_m(D) = N_d(x) \cdot \frac{dD}{dx} = a_m(b_m - 1)x^{b_m - 1}N_d(x)$, where the subscript $d$ denotes that the quantity is diameter-based and $N_d(x)$ is in units of $\text{mm}^{-1}\text{m}^{-3}$. Replacing this in Equation B1 yields:

$$
N_d(x) = N_{0,d} D^{\mu_d} \exp(-\Lambda_d D^{\nu_d})
\tag{B2}
$$

with

$$
\begin{aligned}
N_{0,d} &= \frac{N_{0,m}}{b_m}\left(\frac{1}{a_m}\right)^{\frac{\mu_m+1}{b_m}} \\
\mu_d &= \frac{\mu_m+1}{b_m} - 1 \\
\Lambda_d &= \frac{\Lambda_m}{a_m^{\nu_m/b_m}} \\
\nu_d &= \frac{\nu_m}{b_m}
\end{aligned}
\tag{B3}
$$





By equating $\mathcal{M}_0$ with the number concentration $Q_N$ and $a_d\mathcal{M}_{b_d}$ with the mass concentration $Q_M$, where $a_d = a_m^{-1/b_m}$ and $b_d = 1/b_m$, one is able to retrieve the $N_{0,d}$ and $\Lambda_d$ from the prognostic parameters of the PSDs.

$$N_{0,d} = \frac{\nu_d Q_N}{\Gamma\left(\frac{\mu_d+1}{\nu_d}\right)}\Lambda_d^{\frac{\mu_d+1}{\nu_d}} \quad \text{and} \quad \Lambda_d = \left[\frac{1}{a_d}\frac{\Gamma\left(\frac{\mu_d+1}{\nu_d}\right)}{\Gamma\left(\frac{\mu_d+b_d+1}{\nu_d}\right)}\overline{x}\right]^{-\nu_d/b_d} \tag{B4}$$

where $\overline{x} = Q_M/Q_N$ is the average particle mass.

5    Note that besides these differences in PSD retrieval, the two-moment scheme also yields slightly different hydrometeor scattering properties, since the mass-diameter relations differ from the one-moment scheme.

## Appendix C: Polarimetric equations

Equations C1 give the polarimetric equations integrated over ensembles of hydrometeors and over the antenna power density, for a given set of spherical coordinates $x_0 = (r_0, \theta_0, \phi_0)$, where $r_0$ is the range, $\theta_0$ is the elevation angle $\theta_0$

10   and $\phi_0$ is the azimuth angle. The backscattering matrix $C^b$, forward scattering vector $S^f$, and backscattering cross-sections $\sigma^b$ for a given hydrometeor $(j)$, are defined as in Equations 7, 8 and 9. $\lambda$ is the wavelength in cm. $I$ is the quadrature antenna integration operator defined in Equation 6.


$$Z_h(x_0) = \frac{\lambda^4}{\pi^5 |K_w|^2} I \left[ \sum_{j=0}^{H} \int_{D_{\min}^{(j)}}^{D_{\max}^{(j)}} N^{(j)}(D, x_0) \sigma_h^{b,(j)}(D, x_0) dD \right] (x_0) \qquad \left[ \mathrm{mm^6 m^{-3}} \right]$$

$$Z_v(x_0) = \frac{\lambda^4}{\pi^5 |K_w|^2} I \left[ \sum_{j=0}^{H} \int_{D_{\min}^{(j)}}^{D_{\max}^{(j)}} N^{(j)}(D, x_0) \sigma_v^{b,(j)}(D, x_0) dD \right] (x_0) \qquad \left[ \mathrm{mm^6 m^{-3}} \right]$$

$$K_{\mathrm{dp}}(x_0) = \frac{0.18}{\pi} \lambda I \left[ \sum_{j=0}^{H} \int_{D_{\min}^{(j)}}^{D_{\max}^{(j)}} N^{(j)}(D, x_0) \Re \left( S_1^{f,(j)}(D, x_0) - S_2^{f,(j)}(D, x_0) \right) dD \right] (x_0) \qquad \left[ {}^\circ \ \mathrm{km^{-1}} \right]$$

$$\delta_{\mathrm{hv}}(x_0) = \frac{180}{\pi} \lambda I \left[ \arg \left( \sum_{j=0}^{H} \int_{D_{\min}^{(j)}}^{D_{\max}^{(j)}} N^{(j)}(D, x_0) C_{2,1}^{b,(j)}(D, x_0) dD \right) \right] (x_0) \qquad \left[ {}^\circ \right]$$

$$k_{\mathrm{H}}(x_0) = 8.686 \lambda I \left[ \sum_{j=0}^{H} \int_{D_{\min}^{(j)}}^{D_{\max}^{(j)}} N^{(j)}(D, x_0) \Im \left( S_1^{f,(j)}(D, x_0) \right) dD \right] (x_0) \qquad \left[ \mathrm{dB \ km^{-1}} \right]$$

$$k_{\mathrm{V}}(x_0) = 8.686 \lambda I \left[ \sum_{j=0}^{H} \int_{D_{\min}^{(j)}}^{D_{\max}^{(j)}} N^{(j)}(D, x_0) \Im \left( S_2^{f,(j)}(D, x_0) \right) dD \right] (x_0) \qquad \left[ \mathrm{dB \ km^{-1}} \right]$$

(C1)

where $Z_h$ and $Z_v$ are the linear reflectivity factors at horizontal and vertical polarizations, $K_{\mathrm{dp}}$, is the specific differential phase shift upon propagation, $\delta_{\mathrm{hv}}$ is the total differential phase shift upon backscattering, and $k_H$ and $kA_V$ are the specific attenuation coefficients in logarithmic scale.

The phase shift upon backscattering $\delta_{\mathrm{hv}}$ is not taken into account in the final $K_{\mathrm{dp}}$, because the radar $K_{\mathrm{dp}}$ retrieval method that is being used (Schneebeli et al., 2013) is able to remove the contribution of $\delta_{\mathrm{hv}}$. Besides $K_{\mathrm{dp}}$, the total phase shift $\Psi_{\mathrm{dp}}$ is also simulated, which combines the phase shift due to backscattering and propagation. Additionally, to compute the final reflectivity factors $Z_{\mathrm{H}}$ and $Z_{\mathrm{V}}$, the linear reflectivity factors $Z_h$ and $Z_v$ are converted to logarithmic scale and the specific attenuations are taken into account. This yields the final polarimetric products simulated by the radar operator (Equations C2).



$$Z_{\mathrm{H}}^{\mathrm{att}}(x_0) = 10 \cdot \log\left[Z_h(x_0)\right] + 2 \int_{r=0}^{r_0} k_H(r,\theta_o,\phi_o)\, dr \qquad [\mathrm{dBZ}]$$

$$Z_{V}^{\mathrm{att}}(x_0) = 10 \cdot \log\left[Z_v(x_0)\right] + 2 \int_{r=0}^{r_0} k_V(r,\theta_o,\phi_o)\, dr \qquad [\mathrm{dBZ}]$$

$$Z_{\mathrm{DR}}^{\mathrm{att}}(x_0) = Z_{\mathrm{H}}^{\mathrm{att}} - Z_{V}^{\mathrm{att}} \qquad [\mathrm{dB}]$$

$$\Psi_{\mathrm{dp}}(x_0) = 2 \int_{r=0}^{r_0} K_{\mathrm{dp}}(r,\theta_0,\phi_0) + \delta_{hv}(x_0) \qquad [°] \tag{C2}$$

5    *Acknowledgements.* The authors would like to thank MeteoSwiss for providing the data from the Swiss operational radar network. The authors are also thankful to J. Grazioli for the processing of the raw MXPol radar data and to T. Raupach for the processing of Parsivel data.





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

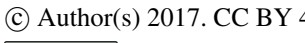



**Figure 8.** Example of simulated and observed (with the Swiss La Dôle C-band radar) PPI at 1° elevation during the 13 August 2015 convective event (Table 3). The left side panel corresponds to the simulated radar observables and the right side to the observed ones. The displayed variables are, from top to bottom, the horizontal reflectivity factor (in dBZ), the differential reflectivity (in dB), the specific differential phase shift upon propagation (in °km$^{-1}$, and the radial velocity (in m s$^{-1}$).





**Figure 9.** Same as Figure 8 but for the stratiform event on the 8 April 2014 (Table 3).



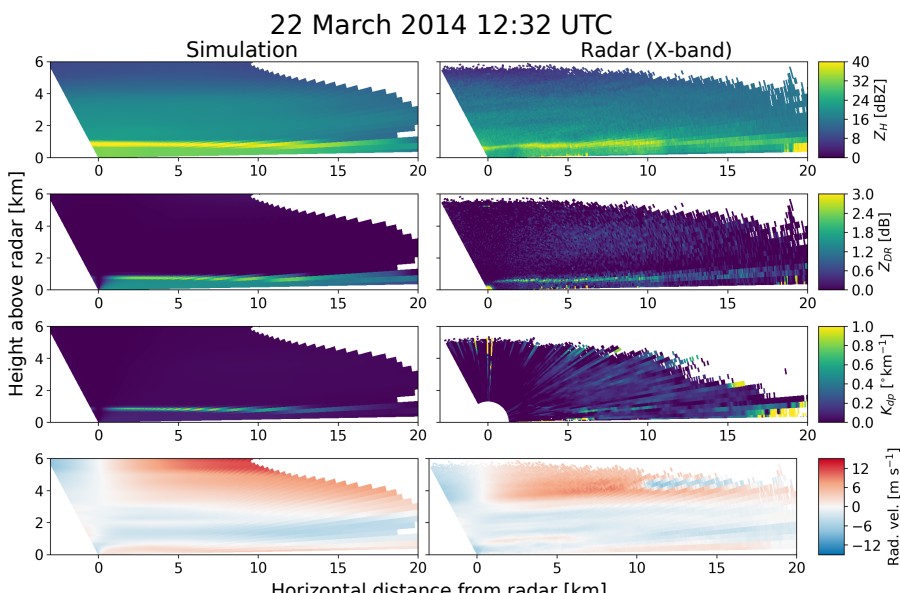

**Figure 10.** Example of RHI showing the observed and simulated melting layer during the PARADISO campaign in Spring 2014 (Table 3). The left panel corresponds to the simulated radar observables, the right panel to the observed values at X-band. Note that there is an area with velocity folding (blue area in the middle of a larger red area) around 5 km altitude and 10-15 km horizontal distance on the radar RHI scan.



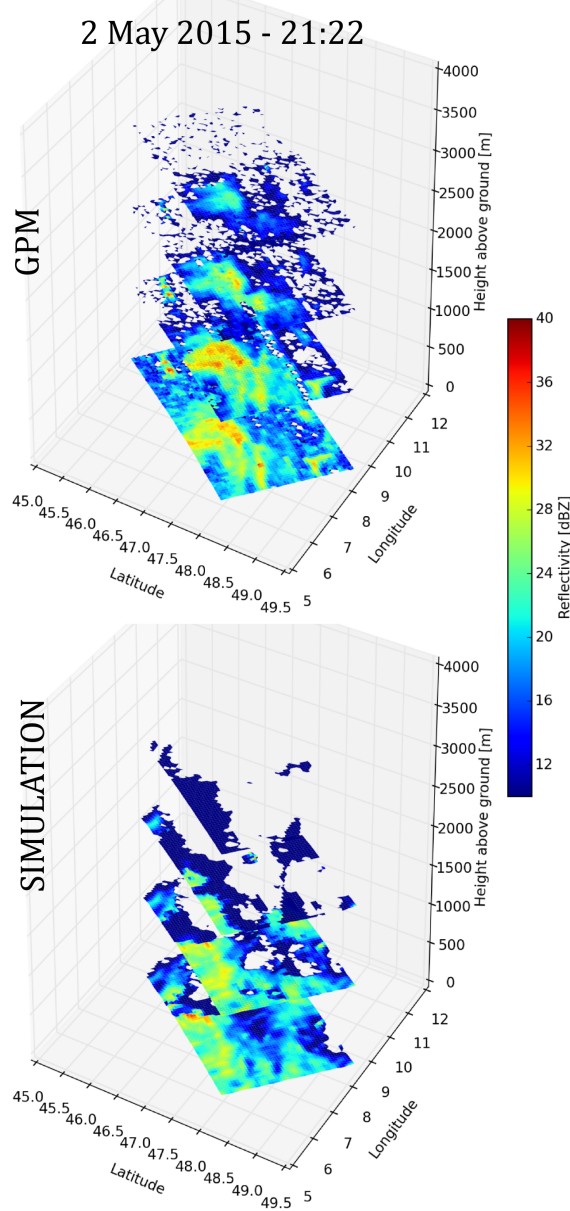

**Figure 11.** Example of comparison at several altitude levels between GPM radar observations at Ka band (top) and the corresponding radar operator simulation from the COSMO model (bottom) for one GPM overpass.





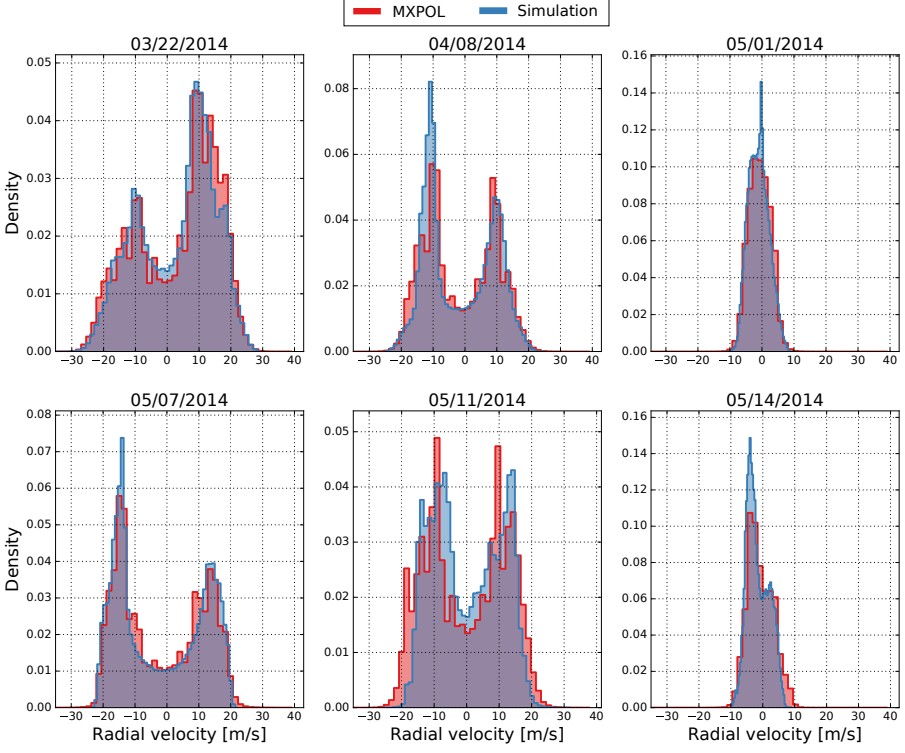

**Figure 12.** Distributions of simulated (blue) and observed (red) radial velocities at X-band during six days of precipitation in Western Switzerland.

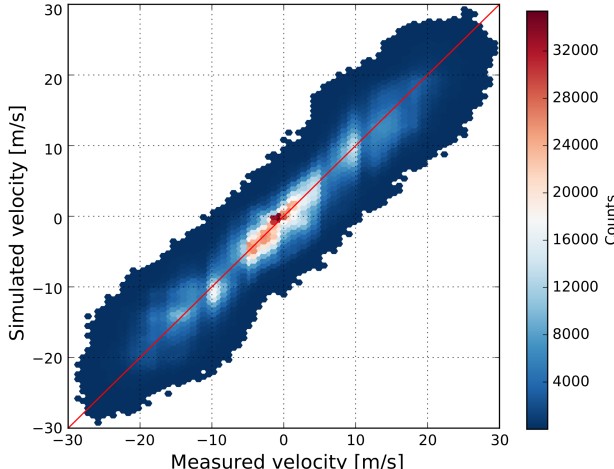

**Figure 13.** Scatter-plot of the measured and simulated radial velocities (for all events). The red line shows the 1:1 relation. The coefficient of determination ($R^2$) is 0.9.



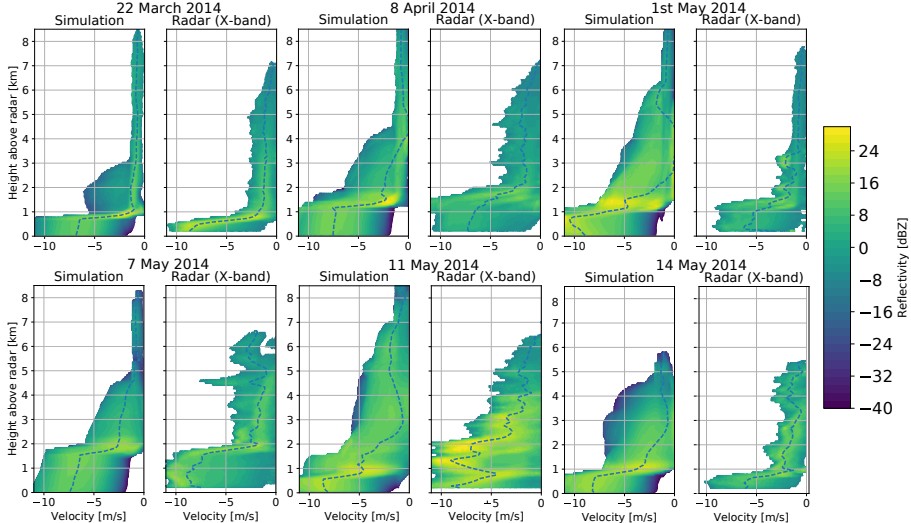

**Figure 14.** Simulated and measured daily averaged Doppler spectrum at X-band at vertical incidence during six days of precipitation in Western Switzerland. The dashed line represents the radial velocity calculated from the spectrum (Equation 35)

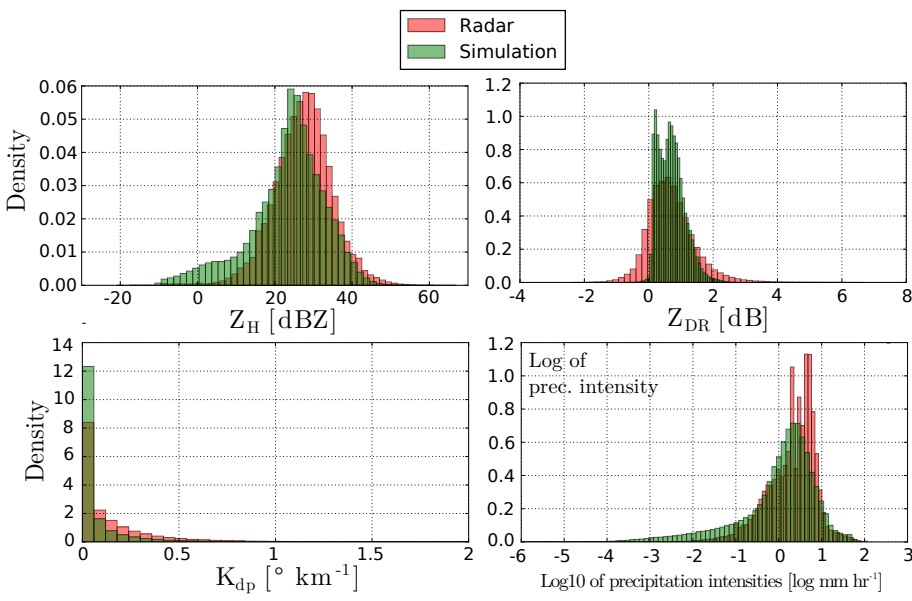

**Figure 15.** Observed (red) and simulated (green) distributions of polarimetric variables ($Z_H$, $Z_{DR}$ and $K_{dp}$) as well as the precipitation intensities on the ground (in log scale) for the one-moment scheme with $\mu^{rain} = 2$ in the liquid phase.




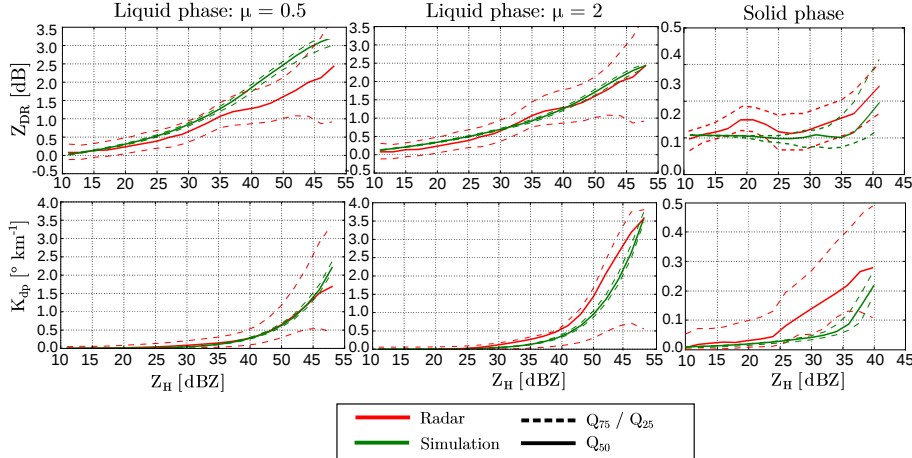

**Figure 16.** Observed (red) and simulated (green) $Z_H - Z_{DR}$ and $Z_H - K_{dp}$ relationships for the COSMO one-moment scheme in liquid and solid phases. These membership functions are computed by dividing all simulated values in bins of reflectivity of 1 dBZ of width, and computing the quantiles of the dependent variable (on the $y$-axis) within every bin.

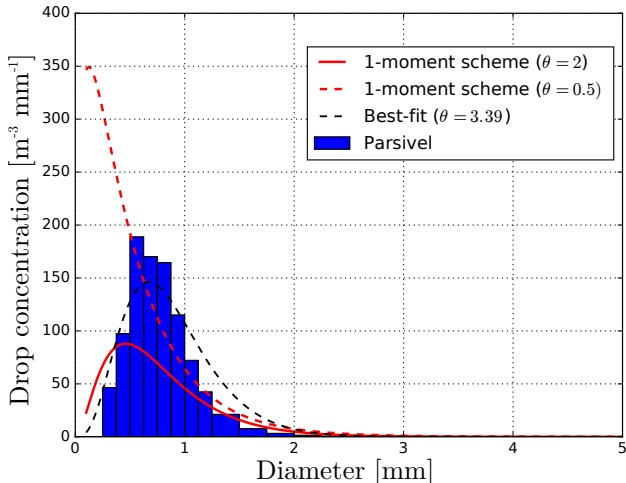

**Figure 17.** Average measured (blue bins) and parameterized rain DSDs at the ground in Payerne over six stratiform precipitation events. The dashed black line corresponds to the best fit of a gamma DSD on the measurements





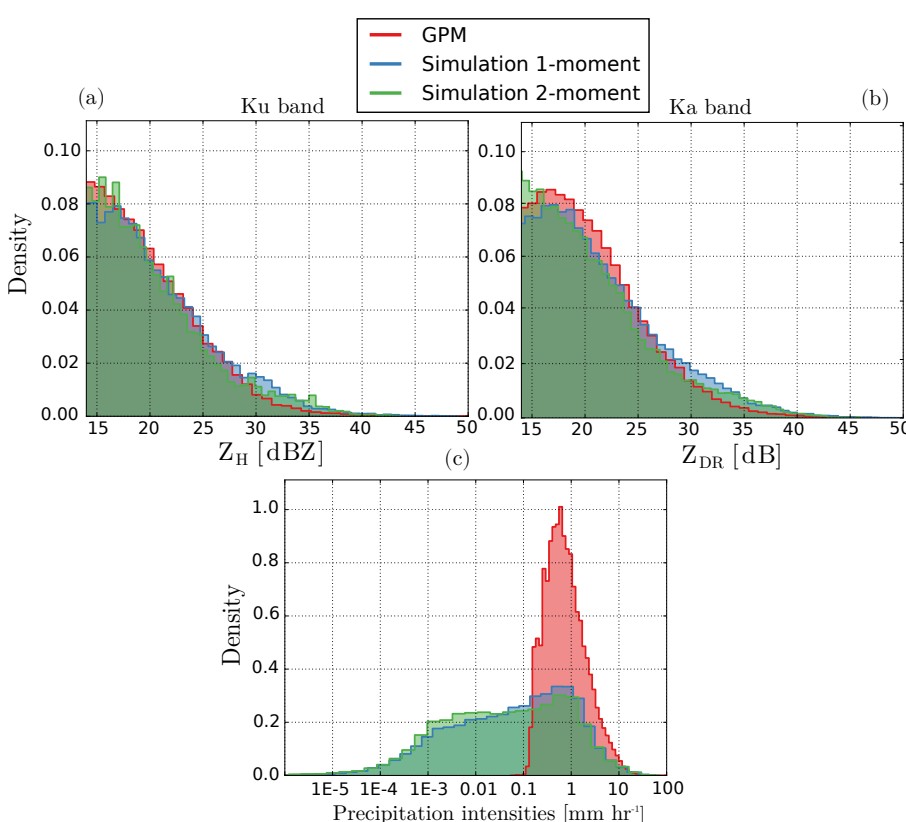

**Figure 18.** Observed (red) and simulated (blue = one-moment, green = two-moments) reflectivities at Ku band (a) and Ka band (b), as well as the precipitation intensities (in log-scale) at the ground (c) estimated by GPM and simulated by COSMO



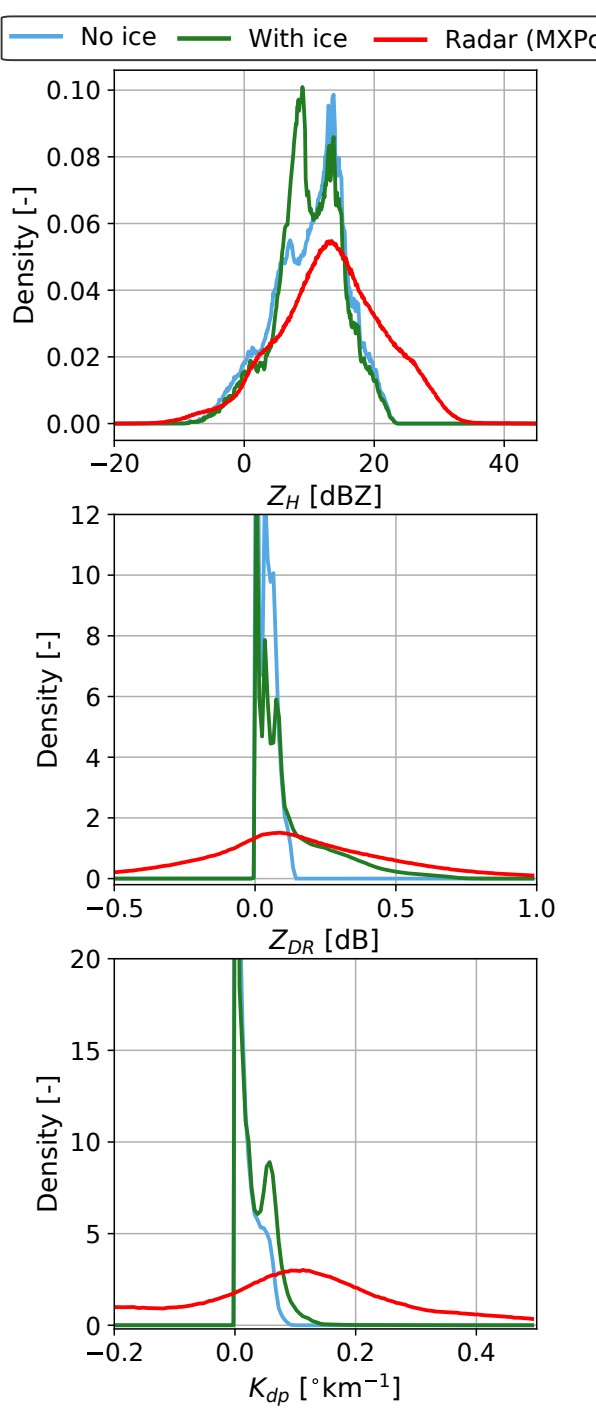

**Figure 19.** Observed and simulated (with and without ice crystals) distributions of polarimetric variables during three pure snowfall events for the one-moment microphysical scheme.



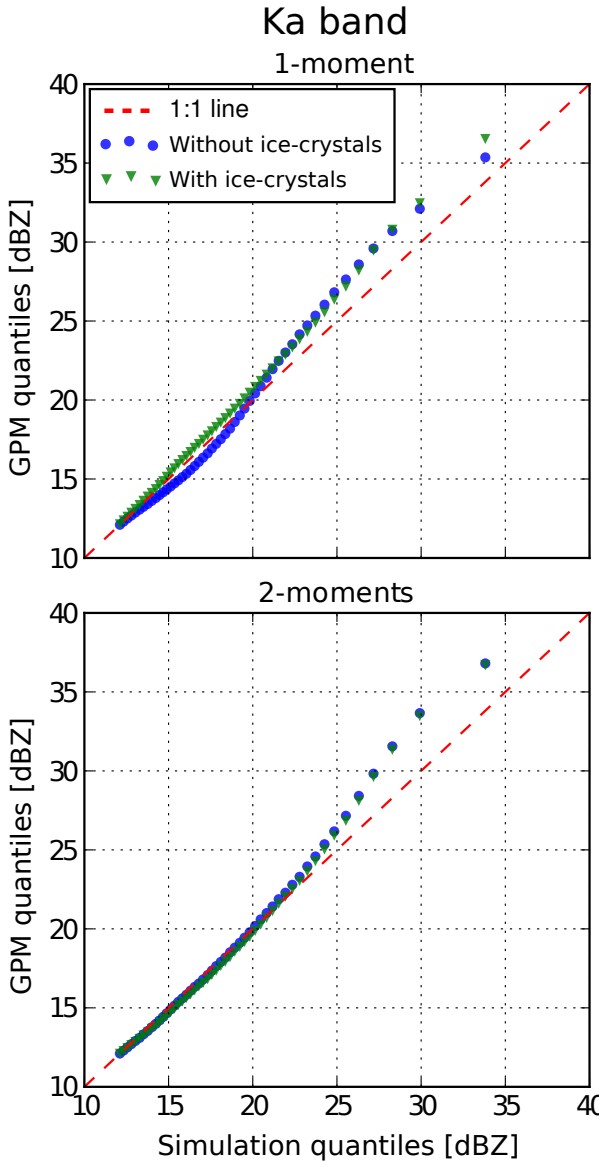

**Figure 20.** QQ-plots of the quantiles of simulated $Z_H$ values versus the quantiles of observed GPM $Z_H$ values at Ka band. The red line corresponds to the 1:1 line indicating a perfect match with observed quantiles.



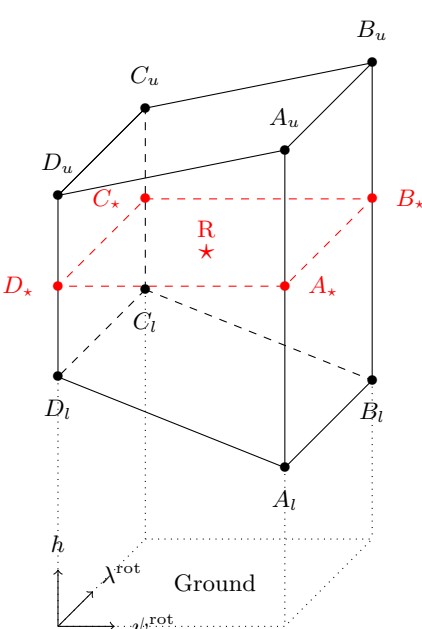

**Figure A1.** Location of the eight neighbours of a radar gate $R$. The position of the radar gate is shown by a red star.