# Peer review of "From model to radar variables: a new forward polarimetric radar operator for COSMO"

_Atmospheric Measurement Techniques, 2017_

## Referee Comment (RC1) · Anonymous Referee #1 · 18 Jan 2018

January 18, 2018

**1   General Comments**

This paper deals with the development of a new polarimetric forward operator for the COSMO-model (ZH, ZDR, KDP, AH, ADR, radial wind, Doppler spectrum). Although the forward operator employs state-of-the-art computation methods to simulate beam propagation- and beam broadening effects and to compute the polarimetric moments from model output (T-matrix for oblate spheroids, 2-component Maxwell-Garnett effective medium approximation for melting particles), it clearly has its merits and useful new developments:

- A particular emphasis is put to efficient and yet accurate numerical methods (quadrature schemes, lookup tables) which is important with respect to its practical applicability.

- Todays state-of-the-art NWP models do not contain any prognostic information about shape asymetry and tumbling behaviour of the hydrometeors (canting angle distribution), but these are important input parameters for the simulation of polarimetric moments. To this end, new parameterizations for graupel- and snow particles of the probability density functions of axis ratio and canting angle distribution as function of partice size have been developed. These new fits are based

on a large particle sample from in-situ observations of a multi-angle snow flake camera (MASC) and might be useful for the community.

- Special quadrature scheme for beam broadening effects at the edges of the melting layer.

- Simulation of the entire Doppler spectrum, not only mean value (radial wind) and standard deviation (spectral width).

This operator is then used to compare measured and simulated polarimetric moments of the Swiss C-Band radar network, an X-Band radar and a space-born Ka-Band cloud radar for different case studies and weather situations. The source code of the operator is freely available on the internet.

While the above merits make the paper well worth publishing, the presentation needs polishing and there seems to be an error in the attenuation simulation for the Doppler spectrum, see below. This error, however, should be possible to correct with a reasonable effort, and therefore I recommend to **accept this paper after major revisions.**

**2 Specific comments**

**Page 14, line 11 ff:**

The exact definition of the scattering matrix elements $s$ which relate the incident and scattered $\vec{E}$ field as function of direction (which angles?) remains somewhat unclear, which is not uncommon in the literature. However, I would find it useful to see the exact equation and a sketch defining the scattering angles. Also, which sign convention for the imaginary part of the refractive index of the scatterers is applied?

These choices influence the exact equations for the radar observables in Appendix C. unclear

**Page 20, line 1:**
Maxwell-Garnett is only one of many known Effective Medium Approximations, and Eq. (13) is the special case of a 2-component mixture ($n$-component mixture see Bohren and Huffman (1983)) where small ice spheres are suspended in air ("matrix"). You could mention some alternative formulations from the literature (see Blahak (2016) for a summary) but stating that, if none of the components is a strong dielectric, all these formulas approximately agree to first order (Bohren and Huffman (1983)). This will become more important later in your section 3.7.2, Permittivity.

**Page 21, line 9:**
Your Eq. (10) is wrong, because the argument of $\log$ should be dimensionless. Did you mean something like

$$Z_H(r) = S(r_0) + G + SNR_{thr} + 20 \cdot \log_{10}\left(\frac{r}{r_0}\right) \qquad (1)$$

where $S(r_0)$ is the sensitivity at a certain reference range $r_0$. Please review this Equation. Is this just a "typo" or are your results affected?

**Page 22, line 19:**
Please define $D$: melted diameter or actual diameter of a melting particle?

**Page 23, line 15 ff:**
The description of your applied EMA is too short and omits necessary detail: Please

give the exact formula of $\epsilon_{eff}$ that you applied for partially melted particles. Describe the role of the air inclusions. Note that there is an $n-$component version of Maxwell-Garnett given in Bohren and Huffman (1983), as well as a variant that assumes spheroidal inclusions instead of spherical inclusions in the matrix medium. Note also that there are other EMA's "on the market", derived under different assumptions on the internal melting morphology and it is not clear which one is "best". This might also depend on the specific radar observable under consideration and is really hard to determine.

Definition of $\rho_{total}$?

Also, please illustrate in a new figure the typical dependence of the mass fraction of water $f_{water} = m_{water}/m_{ice}$ for single particles as function of $D$ and $f_{wet}^{m}$, as derived from your Eq. (24) together with (21). This will shed more light on your implicit assumptions about the distribution of melt water among the particle sizes for given average degree of melting.

**Page 24, line 1:**

In contrast to your Eq. (28), in the original literature Szyrmer and Zawadzki (1999) the equation reads

$$N^r(D_r)v_t^r(D_r) = N^m(D)v_t^m(D) \qquad (2)$$

with $D_r$ equivalent melted diameter of the particle and $D$ its "true" diameter, claiming that this will describe a one-to-one correspondence, i.e., one snow flake leads to one raindrop during the melting process (no shedding/aggregation) and in a steady state the spectral precipitation rate through the melting layer is conserved, While this is not entirely correct (the original formula neglects the functional determinant $dD(D_r)/dD_r$ of the size distribution transformation) your (28) is also not correct in the sense of a one-to-one correspondence.

The correct transformation to achieve this would be

$$N^r(D_r)v_t^r(D_r)dD_r = N^m(D)v_t^m(D)dD \quad \Longrightarrow \quad N^m(D) = N^r(D_r(D))\frac{v_t^r(D_r(D))}{v_t^m(D)}\frac{dD_r}{dD} \quad (3)$$

$$with \quad D_r(D) = \left(\frac{\rho_{total}(D)}{\rho_{water}}\right)^{1/3} D \quad (4)$$

where $\rho_{total}(D)$ is the bulk density of the melting particle having diameter $D$.

While your (28) as a parameterization is entirely valid, there is no one-to-one-correspondence and therefore implicitly shedding/aggregation processes are parameterized across the melting layer in a possibly unrealistic way.

I see two possible ways forward: (a) change your computation of $N^m(D)$ using the "correct" transformation for the one-to-one-correspondence, or (b) keep your parameterization but discuss the implicitly contained shedding/aggregation parameterization somehow.

(a) would be a solution in line with other literature, but some of your data/plots may have to be recomputed,

(b) would be possible by, e.g., producing an exemplary plot based on a specific rain DSD $N^r(D_r)$, which compares $N^m(D)-$ spectra derived by your Eq. (28) and (29) and by the above "correct" one-to-one-correspondence-transformation for different values of $f_{wet}^m$ at constant precipitation flux.

In your Eq. (29), the distribution in the denominator should be $N^m(D)$, not $N^r(D)$, right?

Because $N^m(D)$ through the melting layer is extrapolated from the rain DSD at the bottom, the transition to the dry snow PSD just above the melting layer is not continuous. How large is the "jump"?

**Page 28, line 1:**

The numeric representation of the convolution with a Gaussian kernel in Eq. (39) is wrong. To correct, do either:

- replace the wrong index $i$ in $v_{r,bins}[i]$ by $j$, take the square of this velocity and let the sum over $j$ run over a symmetric interval ($-N_{FFT}/2$ to $+N_{FFT}/2$ I guess),

- or change $S[i-j]$ to $S[j]$ and $v_{r,bins}[i]$ to $(v_{r,bins}[i] - v_{r,bins}[j])^2$.

Also, you have to divide by the sum of the Gaussian weights!

**Page 28, line 3 ff:**

This section should perhaps be better named "attenuation computation" instead of "correction", because the latter is usually used to denote the inverse procedure applied to observations.

Also, the attenuation computation given in Eqs. (40) to (42) is wrong. According to Lambert-Beers law, attenuation in the space of linear reflectivities (such as your spectral reflectivity $S$) is given by

$$Z_h^{att} \; = \; Z_h \, 10^{0.1 \int_0^{r_0} k_H(r)\,dr} \; = \; K\,Z_h \qquad (5)$$

$k_H$ is already 2-way according to your definition in Appendix C (Factor $20/\ln 10 = 8.686$). The attenuation factor $K$ simply applies to all channels of $S$, so that

$$S^{att}[i] \;=\; K\,S[i] \tag{6}$$

Please correct also the text of this section accordingly and recompute the data of your figure 14.

**Page 34, line 1:**

While I agree with the findings of the DSD-comparison in this special case, the well-known general difficulties of such comparisons (vastly different sampling volumes, shapes of normalized spectra strongly depend on rain rate) should be discussed a bit more and why their influence is presumably small in this case.

Also, whether or not to use this "improved" shape parameter value in the forward operator instead of the microphysics-consistent value depends on the application (model verification vs. data assimilation).

Applying it in the model microphysics may be a good idea, but without re-tuning other parameters in the model, one might end up with a degradation of the surface precipitation, because one of the compensating errors has been taken away.

**3   Technical Corrections**

**Page 4, line 16, 24:**
Since COSMO 5.1, ice sedimentation is also taken into account in the 1-moment schemes.

**Page 4, line 19:**

Add two more references for the 2-moment scheme, because the addition of the separate hail class came after Seifert (2006): Blahak (2008), Noppel et al. (2010)

**Page 5, line 6 (Table 1):**

$N_0$ Rain: missing "free" after "2529". Also check the value 2529 (which units???), because the $N0$-$\mu$-relation of Ulbrich (1983) is applied, with a base value of $8000 \, \mathrm{m^{-3} mm^{-1}}$ for $\mu = 0$ and increasing with increasing $\mu$.

Specify the units of $N_0$ in the table caption.

**Page 10, line 3:**

$\frac{dn}{dh} = const$, not $cst$

**Page 17, line 8 ff:**

Change mathematical presentation of your formulas (10) and (11). To reflect that in your *ansatz* the parameters of the Normal- and generalized gamma distribution depend on diameter $D$, you don't have to use the awkward superscripts. In the second formula, I think $a_r$ has to be replaced by $1/a_r$, if I look at your Figure 5 and if I'm not mistaken:

$$o: \qquad g_o(o, D) \;=\; \mathcal{N}\left(0, \sigma_o(D)\right) \qquad (7)$$

$$1/a_r: \qquad g_{1/a_r}(1/a_r, D) \;=\; \frac{(1/a_r - 1)^{\Lambda_{a_r}(D) - 1} \; \exp\left(-\frac{1/a_r - 1}{M(D)}\right)}{M(D)^{\Lambda_{a_r}(D)} \, \Gamma(\Lambda_{a_r}(D))} \qquad (8)$$

where $g_o$ and $g_{1/a_r}$ are the distributions of $o$ and $1/a_r$, respectively. You can eliminate the offset $l$ from the formula and text. Just set it to 1.

Is it correct that $g_{a_r}(a_r, D) = g_{1/a_r}(1/a_r, D) / a_r^2$ ? If yes, you can mention this in the text, because $g_{a_r}(a_r, D)$ is the one you most likely used for computing the radar observables, right?

**Page 17, line 11 ff:**

Delete the sentence starting with "The superscript [D] . . . ". In the next sentence, correct ". . . constant factor $l = 1, \ldots$ " $\Longrightarrow$ "constant shift of 1, . . . ".
Replace also the next sentence "The relationship . . . " by
"These parameters depend on the diameter $D$. Technically, $\Lambda_{a_r}$, $M$ and $\sigma_o$ first have been fitted separately for each single diameter bin of MASC, then their dependence on $D$ has been fitted by power laws for each parameter,"
At this point, you can insert the power laws from Figure 5 as equations in the text, they deserve it! When you do so, please indicate all units.
Then continue with "Note that these power laws allow to estimate the parameters for any arbitrary maximum diameter. This also allows integration over the canting angle . . . "

**Page 20, line 23:**

Homogenize notation of the probability density functions $p(\beta)$, $p(a_r)$ with Eq. (10) and (11)

**Page 27, line 21:**

Missing backslash in front of $sigma_\theta$.

**Page 37, line 20:**

Sentence is garbled, delete "will be performed".

**Page 37, line 21:**

"$(x^{rot}, y^{rot}, z^{rot})$" the same as "$(x_m, y_m, z_m)$"?

**Page 41, line 1 and 2:**
The factor 2 has to be removed from the equations because $k_H$ and $k_V$ are already two-way attenuation coefficients. And the "$+$" sign should be "$-$".

**References**

Blahak, U.: Towards a Better Representation of High Density Ice Particles in a State-of-the-Art Two-Moment Bulk Microphysical Scheme, Extended Abstract, International Conference on Clouds and Precipitation, Cancun, 7.7. – 11.7.2008, http://cabernet.atmosfcu.unam.mx/ICCP-2008/abstracts/Program_on_line/Poster_07/Blahak_extended_1.pdf, 2008.

Blahak, U.: RADAR_MIE_LM and RADAR_MIELIB — Calculation of Radar Reflectivity from Model Output, Tech. Rep. 28, Consortium for Small Scale Modeling (COSMO), http://www.cosmo-model.org/content/model/documentation/techReports/docs/techReport28.pdf, 2016.

Bohren, C. F. and Huffman, D. R.: Absorption and Scattering of Light by Small Particles, John Wiley and Sons, Inc., 1983.

Noppel, H., Blahak, U., Seifert, A., and Beheng, K. D.: Simulations of a Hailstorm and the Impact of CCN Using an Advanced Two-Moment Cloud Microphysical Scheme, Atmos. Res., 96, 286–301, https://doi.org/doi:10.1016/j.atmosres.2009.09.008, 2010.

Szyrmer, W. and Zawadzki, I.: Modelling of the Melting Layer. Part I: Dynamics and Microphysics, J. Atmos. Sci., 56, 3573–3592, 1999.

Ulbrich, C. W.: Natural Variations in the Analytical Form of the Raindrop Size Distribution, J. Clim. Appl. Meteor., 22, 1764–1775, 1983.

---

## Referee Comment (RC2) · Anonymous Referee #2 · 26 Jan 2018

**1   General comments**

The manuscript describes a new polarimetric radar simulator for COSMO. Although polarimetric radar simulators have already been presented in the literature, the authors strived to include several novelties such as Doppler spectrum simulations from bulk quantities and a representation of solid hydrometeors scattering properties derived from MASC observations. Also, the validation is done with a substantial data set of radar observations. The originality of the scientific material makes the manuscript worth publishing in a journal like Atmospheric Measurement Techniques.

[Figure]

The formal presentation of the manuscript is overall good. However, a careful proof-reading will be required before the article is published (only part of the language and math errors is reported here below). All in all, I recommend the manuscript for publication once the following specific comments have been addressed satisfyingly.

**2 Specific comments**

1. p3l25-27. Please split the line in two.

2. p4l14. 'Rutledge' is probably meant here instead of 'Rudledge'.

3. p5. Table 1 contains several typos: 'f' instead of 'free', minus sign instead of empty sign and vice versa, some missing information. Please check it carefully.

4. p7. In the caption of Figure 1, five radars are mentioned whereas there are only three used in the study.

5. p10l25-31. Why is it better to interpolate uncorrelated variables?

6. p10l29. The terms 'number concentration' and 'mass concentration' are both used in the text. Please specify whether you talk about the number or mass whenever the term 'concentration' is used. Alternatively, use another term, like 'contents' to refer to 'mass concentration'.

7. p11l16. Has $Q_N^{(j)}$ been already defined?

8. p11l19-20. I believe that the omission of the contribution of ice crystals in previous radar forward operators is somewhat overestimated by the authors. In particular, ice crystals are actually taken into account by Augros et al. (2016).

[Figure]

9. p12l1-4. I do not understand how the PSDs of ice crystals are retrieved. May the authors provide more details? In particular, I am confused with the different moments that are used.

10. p13. The math symbols 'ln' and 'log' are both used in the study. Do they have different meanings? If not, please use only one notation to avoid confusion.

11. p13. In Equation 6, $z'_j$ and $z'_k$ are not used consistently.

12. p16l3-4. The authors write that the T-matrix method is 'also used for solid hydrometeors (snow, graupel and hail)'. If it was also used for ice crystals, it should be added to the list of solid hydrometeors in parenthesis.

13. p17. Please check Equation 11 which contains some typos: unexpected use of 'd', use of '1' instead of 'l', etc.

14. p18l6 and p19l2. Please check the meaning of 'whereas'. I think 'while' applies better in these contexts.

15. p19. How come ZDR is always above 1 dB in Figure 6?

16. p22. I do not understand Equations 19 and 20. Why introduce $f^{ms}_{wet}$ and $f^{mg}_{wet}$ if they are both equal to $Q^r/(Q^r + Q^s + Q^g)$?

17. p24. Please check Equation 29. I suspect m is actually $m^m$. Also, are terminal velocities missing?

18. p24l13-19. I do not understand how propagation effects (attenuation, in particular) are taken into account when the number of quadrature points is increased in the melting layer only.

19. p28. Please check Equation 39. A parenthesis is not balanced, the function is not Gaussian, etc.

20. p28. Please check Equation 40 which is wrong, given the definition of $k_H$ in Appendix C.

21. p28l15. What is gamma?

22. p32l29-31. Is $\mu^{rain}$ changed in the radar forward operator only, or in the COSMO simulations as well?

23. p34l27. I do not understand why it is argued that GPM tends to underestimate larger reflectivities to explain why larger reflectivities are present more frequently in the simulations. Attenuation is taken into account in the simulations, isn't it? Please elaborate.

24. p46l9-11. The reference is incomplete.

---

## Author Comment (AC1) · 29 Mar 2018

**Answer to reviewers**

Daniel Wolfensberger and Alexis Berne

March 29, 2018

We thank both reviewers for their constructive and relevant comments and suggestions. We have taken into account all the suggested points and we think that the revised version of the paper has gained considerably in clarity and accuracy. Our answers to the reviewers' comments (which are in italic) are shown in black regular font, the new additions or modifications in the revised paper are shown in blue font.

**Anonymous Reviewer #1**

**Specific comments**

1. *Page 14, line 11 ff: The exact definition of the scattering matrix elements s which relate the incident and scattered $\vec{E}$ field as function of direction (which angles?) remains somewhat unclear, which is not uncommon in the literature. However, I would find it useful to see the exact equation and a sketch defining the scattering angles. Also, which sign convention for the imaginary part of the refractive index of the scatterers is applied?*

   We have added some explanation and a sketch about the scattering matrix formalism at the very beginning of Section 3.5.

   The mathematical formulation of the radar observables involves the *scattering matrix* $\mathbf{S}$, which relates the scattered electric field $\mathbf{E^s}$ to the incident electric field $\mathbf{E^i}$ (Bringi and Chandrasekar, 2001) for a given scattering angle.

   $$\begin{bmatrix} E_h^s \\ E_v^s \end{bmatrix} = \frac{e^{-ik_0 r}}{r} \mathbf{S}_{\mathrm{FSA}} \begin{bmatrix} E_h^i \\ E_v^i \end{bmatrix} \tag{1}$$

   where $k_0$ is the wave number of free space ($k_0 = 2\pi/\lambda$).

   The scattering matrix $\mathbf{S}_{\mathrm{FSA}}$ is a $2\times2$ matrix of complex numbers in units of $\mathrm{m^{-1}}$ (e.g., Bringi and Chandrasekar, 2001; Doviak and Zrni, 2006; Mishchenko et al., 2002).

   $$\mathbf{S}_{\mathrm{FSA}} = \begin{bmatrix} s_{hh} & s_{hv} \\ s_{vh} & s_{vv} \end{bmatrix}_{\mathrm{FSA}} \tag{2}$$

   The FSA subscript indicates the forward scattering alignment convention, in which the positive $z$-axis is in the same direction as the travel of the wave (for both the incident and scattered wave). A sketch illustrating the reference unit vectors for the scattered wave in the FSA convention is given in Figure 5.

[Figure]

Figure 5: The direction of the far-field scattered wave is given by the spherical angles $\theta_s$ and $\phi_s$, or by the unit vector $\hat{\psi}_s$. In the FSA convention, the horizontal and vertical unit vectors are defined as $\hat{h}_s = \hat{\phi}_s$ and $\hat{v}_s = \hat{\phi}_s$. The unit vectors for the spherical coordinate system form the triplet $(\hat{\psi}_s, \hat{\theta}_s, \hat{\phi}_s)$, which in the FSA convention becomes $(\hat{\psi}_s, \hat{v}_s, \hat{h}_s)$, with $\hat{\psi}_s = \hat{v}_s \times \hat{h}_s$. This figure was adapted from Bringi and Chandrasekar (2001).

We have also added an explanation on the permittivity at the beginning of Section 3.6.

In the following, the term (complex) *permittivity* will be used for the relative dielectric constant of a given material. It is defined by:

$$\epsilon = \epsilon' + i\epsilon'' \tag{3}$$

where $\epsilon'$ is the real part, related to the phase velocity of the propagated wave, and $\epsilon''$ is the imaginary part, related to the absorption of the incident wave.

2. *Maxwell-Garnett is only one of many known Effective Medium Approximations, and Eq. (13) is the special case of a 2-component mixture (n-component mixture see Bohren and Huffman (1983)) where small ice spheres are suspended in air (matrix). You could mention some alternative formulations from the literature (see Blahak (2016) for a summary) but stating that, if none of the components is a strong dielectric, all these formulas approximately agree to first order (Bohren and Huffman (1983)). This will become more important later in your section 3.7.2, Permittivity*

Thanks, we have added some discussion regarding this point in the paragraph following Eq. 13:

Note that other EMAs exist, such as the Bruggemann (1935) and Oguchi (1983) approximations. If none of the components is a strong dielectric, all these EMAs approximately agree to first order (Bohren and Huffman, 1983). The interested reader is referred to Blahak (2016), for an intercomparison of these EMA in the context of simulated reflectivity fields.

Please see also the answer to point 5 for more general modifications to this section.

3. *Your Eq. (10) is wrong, because the argument of log should be dimensionless. Did you mean something like*

$$Z_H(r_g) < S(r_0) + G + SNR_{thr} + 20 \cdot log_{10}\left(\frac{r_g}{r_0}\right) \tag{4}$$

*where $S(r_0)$ is the sensitivity at a certain reference range $r_0$. Please review this Equation. Is this just a "typo" or are your results affected?*

We agree that the distance in the log should be normalized to yield a dimensionless argument. This is why we modified this Equation as recommended, by introducing a reference distance $r_0$.

The received power at the radar antenna decreases with the square of the range, which leads to a decrease of signal-to-noise ratio (SNR) with the distance. To take into account this effect, all simulated radar variables at range $r_g$ are censored if:

$$Z_{\mathrm{H}}(r_g) < S + G + \mathrm{SNR}_{\mathrm{thr}} + 20 \cdot \log_{10}\left(\frac{r_g}{r_0}\right) \tag{5}$$

where $G$ is the overall radar gain in dBm, $S$ is the radar antenna sensitivity in dBm, $Z_{\mathrm{H}}$ is the horizontal reflectivity factor in dBZ, and $\mathrm{SNR}_{\mathrm{thr}}$ corresponds to the desired signal-to-noise threshold in dB (typically 8 dB in the following). $r_0$ is a distance used to normalize the argument of the logarithm. If all units are consistent and SI based then $r_0 = 1$ m.

Concerning $S$, it is related to the sensitivity of the receiver, so to a certain extent $S$ is defined at a reference distance of 0, this is why we define it as a constant.

4. *Please define D: melted diameter or actual diameter of a melting particle?*

It is the the the maximum dimension of a melting particle. We have added this information in the text following the equation:

The considered diameter $D$ is the actual maximum dimension of a melting particle, and not the melted diameter.

5. *The description of your applied EMA is too short and omits necessary detail: Please give the exact formula of $\epsilon_{eff}$ that you applied for partially melted particles. Describe the role of the air inclusions. Note that there is an ncomponent version of Maxwell-Garnett given in Bohren and Huffman (1983), as well as a variant that assumes spheroidal inclusions instead of spherical inclusions in the matrix medium. Note also that there are other EMAs on the market, derived under different assumptions on the internal melting morphology and it is not clear which one is best. This might also depend on the specific radar observable under consideration and is really hard to determine. Definition of $\rho_{total}$ ? Also, please illustrate in a new figure the typical dependence of the mass fraction of water $f_{water} = m_{water}/m_{ice}$ for single particles as function of D and $f_{wet}^m$, as derived from your Eq. (24) together with (21). This will shed more light on your implicit assumptions about the distribution of melt water among the particle sizes for given average degree of melting.*

We thank Reviewer 1 for this comment. It is quite true that this part had to be improved. We have rewritten the beginning of Section 3.6.2 (paragraph: Snow, graupel, hail and ice crystals) by starting from the general Maxwell-Garnett EMA before introducing the simpler form used for dry solid hydrometeors:

The permittivity of composite materials, such as snow, which consists of a mixture of air and ice, can be estimated with a so-called Effective Medium Approximation (EMA). A well known EMA is the Maxwell-Garnett approximation (Bohren and Huffman, 1983), in which the effective medium consists of a matrix medium with permittivity $\epsilon^{\mathrm{mat}}$ and inclusions with permittivity $\epsilon^{\mathrm{inc}}$:

$$\epsilon_{\mathrm{eff}} = \epsilon^{\mathrm{mat}}\left(\frac{1 + 2 f_{\mathrm{vol}}^{\mathrm{inc}} \frac{\epsilon^{\mathrm{inc}} - \epsilon^{\mathrm{mat}}}{\epsilon^{\mathrm{inc}} + 2\epsilon^{\mathrm{mat}}}}{1 - f_{\mathrm{vol}}^{\mathrm{inc}} \frac{\epsilon^{\mathrm{inc}} - \epsilon^{\mathrm{mat}}}{\epsilon^{\mathrm{inc}} + 2\epsilon^{\mathrm{mat}}}}\right) \tag{21}$$

where $\epsilon_{\mathrm{eff}}$ is the effective permittivity of the composite material, and $f_{\mathrm{vol}}^{\mathrm{inc}}$ is the volume fraction of the inclusions.

Note that other EMAs exist, such as the Bruggemann (1935) and Oguchi (1983) approximations. If none of the components is a strong dielectric, all these EMAs approximately agree to first order (Bohren and Huffman, 1983). The interested reader is referred to Blahak (2016), for an intercomparison of these EMA in the context of simulated reflectivity fields.

Dry solid hydrometeors consist of inclusions of ice in a matrix of air. In this case $\epsilon_{\mathrm{mat}} \approx 1$, which leads to a simplified form of the mixing formula (e.g., Ryzhkov et al. 2011).

$$\epsilon^{(j)} = \frac{1 + 2 f_{\mathrm{vol}}^{\mathrm{ice}} \frac{\epsilon^{\mathrm{ice}} - 1}{\epsilon^{\mathrm{ice}} + 2}}{1 - f_{\mathrm{vol}}^{\mathrm{ice}} \frac{\epsilon^{\mathrm{ice}} - 1}{\epsilon^{\mathrm{ice}} + 2}} \tag{22}$$

The rest of the section is unmodified. We have also added some information about how we compute the permittivity of melting hydrometeors in Section 3.7.2. Note that $\rho^{\text{total}}$, was somewhat of a typo, as it should have been $\rho^{\text{water}}$ instead. We have fixed this in Equation 30 and we have added the mathematical expression for $\rho^m$, the density of the melting hydrometeor.

In Equation 18, we have previously introduced the general two-component Maxwell-Garnett EMA. However, melting hydrometeors are a mixture of three components: water, ice, and air. To compute their permittivity, the general two-component formulation is used recursively, first to derive the permittivity of dry snow (as was done previously for dry snow, graupel, hail and ice crystals), and then the permittivity of the dry snow and water mixture.

The necessary volume fractions of all components $f_{\text{vol}}$ can again be estimated with the mass-diameter model:

$$f_{\text{vol}}^{\text{water}} = f_{\text{wet}}^m \frac{\rho^m}{\rho^{\text{water}}} \tag{34}$$

$$f_{\text{vol}}^{\text{ice}} = \frac{\rho^m - f_{\text{vol}}^{\text{water}} \rho^{\text{water}}}{\rho^{\text{ice}}} \tag{35}$$

$$f_{\text{vol}}^{\text{air}} = 1 - f_{\text{vol}}^{\text{water}} - f_{\text{vol}}^{\text{ice}} \tag{36}$$

$$\tag{37}$$

where $\rho^m = \frac{m^m(D)}{\pi/6 D^3}$ is the density of the melting hydrometeor.

In a first step, Equation 21 is used with $f_{\text{vol}}^{\text{inc}} = \frac{f_{\text{vol}}^{\text{ice}}}{f_{\text{vol}}^{\text{ice}} + f_{\text{vol}}^{\text{air}}}$, $\epsilon^{\text{mat}} \approx 1$, $\epsilon^{\text{inc}} = \epsilon^{\text{ice}}$, to yield $\epsilon^d$, the permittivity of the dry part of the melting hydrometeor. For the second step, however, the estimated permittivity of the melting hydrometeor will depend on whether water is treated as the matrix and snow as the inclusions or the opposite, giving two different possible outcomes. To overcome this issue, a formulation proposed by Meneghini and Liao (1996) is used, where the final permittivity is a weighted sum of both permittivities and where the weights are function of the wet fraction. This method is also used by Ryzhkov et al. (2011). Precisely, Equation 18 is used first with $f_{\text{vol}}^{\text{inc}} = f_{\text{vol}}^{\text{water}}$ and $\epsilon^{\text{mat}} = \epsilon^d$, $\epsilon^{\text{inc}} = \epsilon^{\text{water}}$, to yield $\epsilon^{\text{m,(1)}}$, and at second with $f_{\text{vol}}^{\text{inc}} = f_{\text{vol}}^{\text{air}} + f_{\text{vol}}^{\text{ice}}$ and $\epsilon^{\text{mat}} = \epsilon^{\text{water}}$, $\epsilon^{\text{inc}} = \epsilon^d$, to yield $\epsilon^{\text{m,(2)}}$. The final $\epsilon^m$ is a weighted sum of $\epsilon^{\text{m,(1)}}$ and $\epsilon^{\text{m,(2)}}$:

$$\epsilon^{\text{m}} = \frac{1}{2} \left[ (1+\tau)\epsilon^{\text{m,(1)}} + (1-\tau)\epsilon^{\text{m,(2)}} \right] \tag{38}$$

where parameter $\tau$ is a function of $f_{\text{wet}}^m$:

$$\tau = \text{Erf}\left( 2\frac{1 - f_{\text{wet}}^m}{f_{\text{wet}}^m} - 1 \right) \quad \text{if } f_{\text{wet}}^m > 0.01, \tag{39}$$

We think that this revised version better describes our approach. The dependency of $f_{\text{vol}}^{\text{water}}$ on the diameter $D$ and the wet fraction $f_{\text{wet}}^m$ is illustrated in Figure 6. It basically shows that there is a roughly polynomial increase in the volume fraction with the wet fraction for a fixed diameter, which can be expected. In terms of diameter dependence, the relation is more complex. It can seem surprising, that, for a fixed wet fraction, $f_{\text{vol}}^{\text{water}}$ increases with the diameter for graupel, but decreases for snow. It is easy though to figure out though that this is caused by the mass-diameter relations of dry snow and graupel: dry snow has a power of 2 in its power-law, whereas graupel has a power of 3.1. This implies that dry graupel becomes denser with the size, whereas dry snow becomes less dense (because $3.1 > 3 > 2$). Since melting hydrometeors depend on the density of water (which is constant) and their dry counterparts, the same trend can be found for melting hydrometeors.

6. *In contrast to your Eq. (28), in the original literature Szyrmer and Zawadzki (1999) the equation reads*

$$N^r(D_r)v_t(D_r) = N^m(D)v_t^m(D)$$

*[...]. I see two possible ways forward: (a) change your computation of $N_m(D)$ using the correct transformation for the one-to-one-correspondence, or (b) keep your parameterization but discuss the implicitly contained shedding/aggregation parameterization somehow. [..]. Because $N_m(D)$ through the melting layer is extrapolated from the rain DSD at the bottom, the transition to the dry snow PSD just above the melting layer is not continuous. How large is the jump?*

[Figure]

Figure 6: Dependency of melting graupel and snow $f_{\text{vol}}^{\text{water}}$ on the diameter (top) and wet fraction (bottom).

Thanks to Reviewer 1 for pointing this out. Indeed there has been some misinterpretation of Zawadzki's paper. Hence, we have decided to improve this part in the following way: we have first corrected the approach based on Szyrmer and Zawadzki (1999), by talking into account Reviewer 1's corrections. By closer visual inspection, we realized that, indeed, there is a jump between the dry snow PSD and the PSD of melting snow when $f_{\text{wet}} = 0$ and it is quite large. It can lead to an unrealistic sharp drop of $Z_{\text{H}}$ of several dBZ over one or two radar gates, above the melting layer. We have then added another approach based on a simple empirical weighting between the DSD of raindrops and the PSD of dry solid hydrometeors (snow/graupel). We have then compared these two approaches, and it seems the second ones performs better in comparison with radar data, and it also allows for a seamless transition between the PSD of melting hydrometeors and the PSD of dry hydrometeors. In the end, the second approach is favored.

Since this new part is quite long, we do not copy it here, but we would like to refer the reviewers to the new Section 3.7.3 in the revised paper.

7. *The numeric representation of the convolution with a Gaussian kernel in Eq. (39) is wrong. To correct, do either: [...]. Also, you have to divide by the sum of the Gaussian weights!*

We thank Reviewer 1 for pointing this error out. Fortunately, this is just an error in the text, as in the code, we used the *convolve* function from the well-known Python *scipy* library. We fixed the equation according to your second proposition:

$$S^{\text{corr}}[i] = \sum_{j=0}^{N_{\text{FFT}}} S[i-j] \frac{1}{\sigma_{t+\alpha}\sqrt{2\pi}} \exp\left[-\frac{(v_{\text{rad,bins}}[j])^2}{2\sigma_{t+\alpha}^2}\right] \tag{52}$$

where $\sigma_{t+\alpha} = \sigma_t + \sigma_\alpha$

However, we think that there is no need to divide by the sum of the Gaussian kernel because this sum should be one, indeed, the term $\frac{1}{\sigma_{t+\alpha}\sqrt{2\pi}}$ yields a normalized Gaussian kernel.

8. *This section should perhaps be better named attenuation computation instead of correction, because the latter is usually used to denote the inverse procedure applied to observations. Also, the attenuation computation given in Eqs. (40) to (42) is wrong. According to Lambert-Beers law, attenuation in the space of linear reflectivities (such as your reflectivity S) is given by [...]. Please correct also the text of this section accordingly and recompute the data of your figure 14.*

Reviewer 1 is right and we thank him for this correction. We somehow got quite confused, and ended up solving a uselessly difficult problem...This derivation as well as the title have been changed in the text. Note that $k_h$ (one-way attenuation in linear units) is used now (as can be seen in the new appendix C2):

In reality, attenuation will cause a decrease in observed radar reflectivities at all velocity bins within the spectrum. To take into account this effect, the path integrated attenuation in linear units at a given radar gate ($k_h$ in Equations C2) is distributed uniformly throughout the spectrum.

$$S(r_g, \phi_g, \theta_g)^{\mathrm{att}}[i] = S(r_g, \phi_g, \theta_g)[i] \cdot \exp\left(-2 \int\limits_{r=0}^{r_g} k_h(r, \theta_g, \phi_g) \; dr\right) \tag{53}$$

Please note that the appendix $C$ has changed, and a single-way linear attenuation $k_h$ is used instead, which explains why there is still a factor of 2 in the equation above. This modification leads to a slightly different Figure 14 (please see the revised paper), which does not change the conclusions that are being drawn. Note that Figure 14 also shows some differences in SNR censoring of the radar data. This is because, while recomputing the plot, we realized that we were not being consistent with the SNR threshold of 8 dB, that was mentioned in the paper (a smaller value was used). Now this is fixed as well.

9. *Page 34, line 1: While I agree with the findings of the DSD-comparison in this special case, the well-known general difficulties of such comparisons (vastly different sampling volumes, shapes of normalized spectra strongly depend on rain rate) should be discussed a bit more and why their influence is presumably small in this case. Also, whether or not to use this improved shape parameter value in the forward operator instead of the microphysics-consistent value depends on the application (model verification vs. data assimilation). Applying it in the model microphysics may be a good idea, but without re-tuning other parameters in the model, one might end up with a degradation of the surface precipitation, because one of the compensating errors has been taken away.*

Thanks to Reviewer 1 for providing this complement of information. We have added the recommended remarks in text, at the end of the paragraph:

However, one must keep in mind the numerous difficulties in the comparison of these DSDs. First of all, the sampling volumes are vastly different (around 80 millions of cubic meters for the COSMO grid cell, around 10000 cubic meters for the three Parsivels integrated over a time interval of 5 minutes and averaged over 520 of these time intervals. Secondly, the shape of the DSDs depend strongly on the simulated precipitation intensity which is not always agreeing with observations (rain gauges). Regarding the first point, giving the large homogeneity of the studied precipitation events (widespread stratiform rain), the representativity issue comparison still has some relevance. Concerning the second point, since precipitation intensity is a moment of the DSD, one can expect a better agreement with Parsivel observations with more realistic COSMO microphysics, especially for larger particles.

As conclusion, changing the shape parameter in the COSMO microphysics is a delicate task, as without re-tuning other parameters in the model, it might lead, *in fine*, to a degradation of the surface precipitation. Using it solely off-line in the context of the forward radar operator might be a better choice, as it can help to reduce the bias in simulated polarimetric variables.

**Technical corrections**

1. *Since COSMO 5.1, ice sedimentation is also taken into account in the 1-moment schemes.*

Thanks to Reviewer 1 for providing this correction, we have added the following details in the text:

In terms of terminal velocities, in the version of COSMO that is being used (5.04), neither ice crystals nor cloud droplets are sedimenting. In more recent versions (starting from 5.1) however, ice crystals have a bulk non-diameter dependent terminal velocity, that depends on their mass concentration.

2. *Page 4, line 19: Add two more references for the 2-moment scheme, because the addition of the separate hail class came after Seifert (2006): Blahak (2008), Noppel et al. (2010)*

We have added these references in the corresponding sentence.

A more advanced two-moment scheme with a sixth hydrometeor category, hail, was developed for COSMO by Seifert and Beheng (2006) and extended by Blahak (2008) and Noppel et al. (2010).

3. *Page 5, line 6 (Table 1): $N_0$ Rain: missing free after 2529. Also check the value 2529 (which units???), because the $N_0$-$\mu$-relation of Ulbrich (1983) is applied, with a base value of 8000 $m^3 mm^1$ for $\mu = 0$ and increasing with increasing $\mu$. Specify the units of $N_0$ in the table caption.*

   We fixed the term "free" after "2529", and performed several modifications (see point 3 of Reviewer 2). We also checked the value 2529, and this value is clearly false. Fortunately, it was just a typo. We checked the code, and the value used there is 1253 $m^{-3} mm^{-1-\mu}$. To get this value, we used the formulas defined in the following document: `http://www.cosmo-model.org/content/model/releases/histories/cosmo_4.21.htm`. They specify that

   $$N_0 = \text{rain\_n0\_factor} \cdot N_{00} \exp(3.2\mu)$$

   Note that in the document it is written lambda, but we expect it to be mu, it doesn't make sense otherwise (this document is a bit confusing). Also $N_{00} = 8e6$ m$^{-4}$. With $\mu = 0.5$ and rain\_n0\_factor $= 1.0$, which is the value we used and it is the default used by MeteoSwiss, this gives $N_0 = 39624259.39$, which must be in units of m$^{-4-\mu}$. However we prefer to have it units of m$^{-3}$mm$^{-1-\mu}$, because we work with diameters in mm, so we divide by $1000^{1+\mu}$ and this gives 1253.03 m$^{-3}$mm$^{1-\mu}$. We added the units in the description of the table.

4. *Page 10, line 3: $\frac{dn}{dh} = const$, not cst*

   This has been fixed, thanks.

5. *Change mathematical presentation of your formulas (10) and (11). To reflect that in your ansatz the parameters of the Normal- and generalized gamma distribution depend on diameter D, you dont have to use the awkward superscripts. In the second formula, I think $a_r$ has to be replaced by $1/a_r$, if I look at your Figure 5 and if Im not mistaken: [...]. You can eliminate the offset l from the formula and text. Just set it to 1.*

   The equations have been corrected and edited according to the Reviewer 1's recommendations:

   $$o : g_o(o, D) = \mathcal{N}(0, \sigma_o(D)) \tag{15}$$

   $$\frac{1}{a_r} : g_{1/a_r}(1/a_r, D) = \frac{(\frac{1}{a_r} - 1)^{\Lambda_{a_r}(D)-1} \exp\left(-\frac{\frac{1}{a_r}-1}{M(D)}\right)}{M(D)^{\Lambda_{a_r}(D)} \Gamma(\Lambda_{a_r}(D))} \tag{16}$$

   where $\Lambda_{a_r}$ and $M$ are the *shape* and *scale* parameters of the gamma aspect-ratio probability density function and $\sigma_o$ is the *standard deviation* of the Gaussian canting angle distribution. These parameters depend on the diameter $D$. Technically $\Lambda$, $M$ and $\sigma_o$ have been fitted separately for each single diameter bin of MASC, then their dependence on $D$ has been fitted by power-laws for each parameter, which also allows further integration over the canting angle and aspect-ratio distributions for all particle sizes. Note also that the gamma distribution is rescaled with a constant shift of 1, to account for the fact that the smallest possible inverse of aspect-ratio is 1 and not 0.

   $$\begin{aligned} \sigma_o(D) &= 58.07 \ D^{-0.11} &[°] \\ \Lambda_{a_r}(D) &= 6.33 \ D^{-0.4} &[-] \\ M(D) &= 0.06 \ D^{-0.71} &[-] \end{aligned} \tag{17}$$

   Note that using the properties of the inverse distribution, the distribution of aspect-ratios can easily be obtained from the distributions of their inverses:

   $$g_{a_r}(a_r, D) = \frac{1}{a_r^2} g_{1/a_r}(1/a_r, D) \tag{18}$$

6. *Delete the sentence starting with The superscript [D] . . . . In the next sentence, correct. . . constant factor $l = 1, . . . \Rightarrow$ constant shift of 1, . . . . Replace also the next sentence The relationship . . . by These parameters depend on the diameter D. Technically, $\Lambda_{a_r}$, M and $\sigma_o$ first have been fitted separately for each single diameter bin of MASC, then their dependence on D has been fitted by power laws for each parameter, At this point, you can insert the power laws from Figure 5 as equations in the text, they deserve it! When you do so, please indicate all units. Then continue with Note that these power laws allow to estimate the parameters for any arbitrary maximum diameter. This also allows integration over the canting angle...*

   Thanks to Reviewer 1 for this comment, we have adapted this part of the paper and added the numerical expression for the best-fits, please see the previous point.

7. *Page 20, line 23: Homogenize notation of the probability density functions $p(\beta)$, $p(a_r)$ with Eq. (10) and (11)*

   Thanks, we have fixed these equations and their description:

   $$C^{b,(j)}(D) = \frac{1}{2\pi} \int\limits_{0}^{2\pi} \int\limits_{-\pi/2}^{\pi/2} \int\limits_{0}^{1} c^{b,(j)}(D, a_r, \alpha, o) \ \cos(o) \ g_o(o, D) \ g_{a_r}(a_r, D) \ d\alpha \ do \ da_r \tag{23}$$

   And for rain and hail, where $a_r$ is constant for a given diameter:

   $$C^{b,(j)}(D) = \frac{1}{2\pi} \int\limits_{0}^{2\pi} \int\limits_{-\pi/2}^{\pi/2} c^{b,(j)}(D, \alpha, o) \ \cos(o) \ g_o(o, D) \ d\alpha \ do \tag{24}$$

   where $c^{b,(j)}(D, \alpha, o)$ are the scattering properties for a fixed diameter, canting angle $o$ and yaw Euler angle (azimuthal orientation) $\alpha$. $g_o(o)$ and $g_{a_r}$ are the probabilities of $o$ and $a_r$ for a given diameter $D$ as obtained from Equations 15 and 18. Note that the final scattering properties are averaged over all azimuthal angles $\alpha$, which are all considered to be equiprobable. The $\cos(o)$ in the equation is the *surface element* which arises from the fact that the integration over $\alpha$ and $o$ is a surface integration in spherical coordinates. The procedure for $S^f$ is exactly the same.

8. *Page 27, line 21: Missing backslash in front of sigmaθ.*

   Fixed, thanks.

9. *Sentence is garbled, delete will be performed.*

   We have corrected the sentence according to the suggestion.

10. *Page 37, line 21: "$(x^{rot}, y^{rot}, z^{rot})$" the same as $(x_m, y_m, z_m)$ ?*

    Yes, thanks to Reviewer 1. for pointing this out. We have changed all superscripts $^{rot}$ to the subscript $_m$, in order to be consistent.

11. *Page 41, line 1 and 2: The factor 2 has to be removed from the equations because $k_H$ and $k_V$ are already two-way attenuation coefficients. And the + sign should be .*

    Please note that this part of the appendix has been rewritten to be more mathematically correct. The attenuation is now considered in linear $Z$ units, so it becomes a multiplication and not an addition.

---

## Author Comment (AC2) · 29 Mar 2018

**Answer to reviewers**

Daniel Wolfensberger and Alexis Berne

March 29, 2018

We thank both reviewers for their constructive and relevant comments and suggestions. We have taken into account all the suggested points and we think that the revised version of the paper has gained considerably in clarity and accuracy. Our answers to the reviewers' comments (which are in italic) are shown in black regular font, the new additions or modifications in the revised paper are shown in blue font.

**Anonymous Reviewer #2**

**Specific comments**

1. *p3l25-27. Please split the line in two.*

   Thanks to Reviewer 2 for pointing this out. This has been fixed:

   The article is structured as follows. In Section 2, a description of the COSMO NWP model as well as the radar data used for the evaluation of the operator is given. In Section 3, the different steps of the polarimetric radar operator are extensively described and its assumptions are discussed in details.

2. *p4l14. Rutledge is probably meant here instead of Rudledge.*

   Yes, indeed, there was a typo in the bibliography file. This is now fixed.

3. *p5. Table 1 contains several typos: f instead of free, minus sign instead of empty sign and vice versa, some missing information. Please check it carefully*

   Yes, indeed, we are thankful to Reviewer 2 for noticing that. We have checked the table and made several corrections. The occurence of empty signs were removed and replaced with dash signs, every time the hydrometeor was considered in the microphysical scheme, but the specific parameter was not used in the parameterization. The $\emptyset$ is only used if the hydrometeor type is not considered by the microphysical scheme (hail in the one-moment scheme). Please check the new Table 1 in the revised version.

4. *p7. In the caption of Figure 1, five radars are mentioned whereas there are only three used in the study.*

   Yes, indeed the sentence was a bit clumsy. In fact there are currently five operational polarimetric C-band radars in Switzerland, which are all displayed in the Figure 1. However, two of them were installed only quite recently and as some of the studied events are already quite old (up to 2010), only the three "older" radars were used, which explains the caption. We have reformulated the sentence a bit to make this less confusing:

   An overview of the specifications of all radars used in this study is given in Table 2. The location of the Swiss operational radars used in the evaluation of the radar operator (Section 4.3) and their maximum considered range (100 km) are shown in Figure 1.

   And in the caption of Figure 1:

   Location of the Swiss operational radars. The three radars used in the context of this study are surrounded by black circles which indicate the maximum range of radar data (100 km) used for the evaluation of the radar operator (Section 4.3). Note that as they were installed only quite recently, no data from the Weissfluhgipfel and Plaine Morte radars were used in this study.

5. *p10l25-31. Why is it better to interpolate uncorrelated variables?*

Thanks to Reviewer 2. for raising this point. After reflexion, we realized that our explanation could be confusing. What we wanted to say is that by computing radar observables after downscaling model variables (concentration and temperature), one is able to ensure the conservation of the mathematical relations between radar observables (which are clearly correlated and dependent) at the radar gate scale. Doing the opposite (i.e. computing radar observables at the model grid scale) and downscaling them later to the radar grid does not guarantee this conservation and leads to an artificial linearization of these relations, caused by the linear interpolation method.

To illustrate this we designed a small idealized setup: a Gaussian field of rain mass concentrations (g of rain per m$^3$ of air) of size 128 x 128 pixels is generated randomly. Let's call this field $Q$. Two approaches are then compared.

(a) The resolution of the field $Q$ is decimated by reducing by two its resolution along both dimensions. The factor of decimation is then increased from 2 to 4, 8, 16 and 32. For a factor of 32, this implies that the resulting decimated field will be of size 4 x 4. The decimated field is then downscaled with bilinear interpolation to match the original resolution of 128 x 128. From this downscaled field of rain concentration, for every decimation factor, the radar observables $Z_H$ and $Z_{DR}$ are computed using the COSMO DSD and lookup tables used in the radar operator (at X-band).

(b) $Z_H$ and $Z_{DR}$ are computed only once on the original undecimated field $Q$. The computed $Z_H$ and $Z_{DR}$ fields are then decimated and downscaled in a similar fashion as in approach 1.

Figure 1 shows the resulting $Z_H$ - $Z_{DR}$ relationships for both approaches, for a random generation of $Q$. However the conclusions that can be drawn are general and apply to any randomly generated field (with differences in the magnitude of the observed trends). It is evident that approach 1 (downscaling before computing the radar variables) seems preferable to approach 2 (computing radar variables and then downscaling). Indeed, the $Z_H$ - $Z_{DR}$ relationship of the original undecimated $Q$ field is preserved no matter the decimation factor that is used. In contrary, when downscaling $Z_H$ and $Z_{DR}$, the $Z_H$ - $Z_{DR}$ becomes more and more distorted, the larger the decimation factor. In fact it becomes more and more linear, which can be explained by the bilinear interpolation that is used.

We hope that this example has made our point more clear. Note also that the explanation has been modified in the revised article.

Secondly, computing radar observables after downscaling allows to preserve the mathematical relation between them. Indeed, radar variables are far from being independent. For example, in the liquid phase $Z_H$ is closely co-fluctuating with $Z_{DR}$, in the form of a power-law that tends to stagnate at large reflectivities. Some tests were performed on random Gaussian fields of rain mass concentration. The results indicate that when computing the radar observables first and then downscaling them, this theoretical relation becomes more and more linear when the final downscaled resolution increases, which is quite unrealistic. In contrary, when computing the radar variables after downscaling the rain concentration field, the theoretical relationship is always preserved, regardless of the downscaling that is used.

6. *p10l29. The terms number concentration and mass concentration are both used in the text. Please specify whether you talk about the number or mass whenever the term concentration is used. Alternatively, use another term, like contents to refer to mass concentration.*

Thanks. To solve this issue, we have removed all occurrences of the term "concentration" as a single word. It is now always explicitly referred to as "mass concentration" or "number concentration".

7. *p11l16. Has $Q_N^{(j)}$ been already defined?*

No, indeed Reviewer 2 is right, we forgot to define it! We have now corrected this sentence in the text:

[...] at every radar gate using the model variable $Q_M^{(j)}$, and, for the two-moments scheme, the prognostic number concentration $Q_N^{(j)}$ ($\mathcal{M}_0$) as well.

[Figure]

Figure 1: Example of random $Q$ field (top) and $Z_H$ - $Z_{DR}$ relationships obtained by computing radar observables after downscaling (bottom-left) and before downscaling (bottom-right). The different lines correspond to the relationships obtained with different decimation factors.

8. *p11l19-20. I believe that the omission of the contribution of ice crystals in previous radar forward operators is somewhat overestimated by the authors. In particular, ice crystals are actually taken into account by Augros et al. (2016).*

   Yes, Reviewer 2 is right, sorry about that. We removed this sentence from the paper.

9. *p12l1-4. I do not understand how the PSDs of ice crystals are retrieved. May the authors provide more details? In particular, I am confused with the different moments that are used.*

   Yes, this part was really too short. We have added some details in the paper that we hope make it much more clear:

   Instead, a realistic PSD is retrieved with the double-moment normalization method of Lee et al. (2004). This formulation of the PSD requires to know two moments of the PSD as well as an appropriate normalized PSD function. Field et al. (2005) proposes best-fit relations between the moments of ice crystals PSDs as well as fits of generating functions for different pair of moments. Precisely, assuming moments 2 ($\mathcal{M}_2$) and 3 ($\mathcal{M}_3$) of the size distributions are known, Field et al. (2005) suggest to parameterize the PSD in the following way:

$$N^{\text{ice}}(D) = \mathcal{M}_2^4 \cdot \mathcal{M}_3^{-3} \phi_{23}(x), \qquad \text{with} \qquad x = D\left(\frac{\mathcal{M}_2}{\mathcal{M}_3}\right) \tag{4}$$

   with

$$\phi_{23}(x) = 490.6 \exp(-20.78x) + 17.46x^{0.6357} \exp(-3.290x) \tag{5}$$

Unfortunately, in the one-moment scheme of COSMO, only one single moment is known, which corresponds to $\mathcal{M}_3$, since the value of the $b$ parameter in the mass-diameter power-law for ice crystals is equal to 3 (see Table 1). Fortunately Field et al. (2005), also provide best-fit relations relating $\mathcal{M}_2$ to other moments of the PSD. According to these relationships, $\mathcal{M}_3$ can be estimated from $\mathcal{M}_2$ with:

$$\mathcal{M}_3 \approx a(3, T_c)\mathcal{M}_2^{b(3,T_c)} \tag{6}$$

where $a(3, T_c)$ and $b(3, T_c)$ are polynomial functions of the in-cloud temperature (in ° C) and the moment order (3 in this case).

Taking advantage of these results, it is possible to retrieve a PSD for ice crystals in the radar operator by (1) using the COSMO temperature to retrieve an estimate for $a(3, T_c)$ and $b(3, T_c)$, (2) inverting Equation 6 to get an estimate of $\mathcal{M}_2$, and (3) use Equations 4 and 5 to estimate the PSD of of ice crystals.

10. *p13. The math symbols ln and log are both used in the study. Do they have different meanings? If not, please use only one notation to avoid confusion.*

Thank, we have replaced all occurences of "ln" by "log", to indicate the natural (Naperian) logarithm. If the common logarithm (base 10) is used it is written explicitly as $\log_{10}$. We hope this will clear it out.

11. *p13. In Equation 6, $z_j^{'}$ and $z_k^{'}$ are not used consistently*

Unfortunately, we were not totally sure about what the Reviewer 2 refers to as an inconsistency. We have rephrased the description of Equation 6 to try to be more precise.

$$I[y](r_o, \theta_o, \phi_o) \approx \sum_{j=1,k=1}^{J,K} w_j' w_k' \; y(r_o, \; \theta_0 + z_j', \; \phi_0 + z_k') \cos(\theta_0 + z_k') \tag{9}$$

where $w_i' = \sigma w_i$, $w_j' = \sigma w_j$ and $z_i' = \sigma z_i$, $z_j' = \sigma z_j$ with $\sigma = \frac{\Delta_{3dB}}{2\sqrt{2\log 2}}$, where $\Delta_{3dB}$ is the 3 dB beamwidth of the antenna in degrees. $w_i$ and $z_i$ are respectively the weights and the roots of the Hermite polynomial of order $J$ (for azimuthal integration) and $w_i$ and $z_i$ are the weights and roots of the Hermite polynomial of order $K$ (for elevational integration).

12. *p16l3-4. The authors write that the T-matrix method is also used for solid hydrometeors (snow, graupel and hail). If it was also used for ice crystals, it should be added to the list of solid hydrometeors in parenthesis.*

Indeed, the T-matrix method was used for ice crystals as well. We have added this info in the parenthesis according to your suggestion.

This method was also used for the solid hydrometeors (snow, graupel, hail and ice crystals), at the expense of some adjustments, that will be described later on.

13. *p17. Please check Equation 11 which contains some typos: unexpected use of d, use of 1 instead of l, etc.*

In accordance with the remarks of Reviewer 1 and 2, this equation has been significantly changed.

$$o : g_o(o, D) = \mathcal{N}(0, \sigma_o(D)) \tag{15}$$

$$\frac{1}{a_r} : g_{1/a_r}(1/a_r, D) = \frac{(\frac{1}{a_r} - 1)^{\Lambda_{a_r}(D)-1} \exp\left(-\frac{\frac{1}{a_r}-1}{M(D)}\right)}{M(D)^{\Lambda_{a_r}(D)}\Gamma(\Lambda_{a_r}(D))} b \tag{16}$$

14. *p18l6 and p19l2. Please check the meaning of whereas. I think while applies better in these contexts.*

We have replaced both occurences of "whereas" by "while"

15. *p19. How come ZDR is always above 1 dB in Figure 6?*

Thanks to Reviewer 2 for raising this important point. The plot label was indeed wrong because ZDR was in linear units (dimensionless $Z_h/Z_v$) instead of dB, so this explains the strange values. This has been fixed in the revised version, and ZDR is now plotted in dB as indicated in the label.

16. *p22. I do not understand Equations 19 and 20. Why introduce $f_{wet}^{ms}$ and $f_{wet}^{mg}$ if they are both equal to $Q^r/(Q^r + Q^s + Q^g)$?*

We are not sure to understand the Reviewer's point here. In the text, it is written that:

$$f_{\text{wet}}^{ms} = \frac{Q^r Q^s}{Q^s (Q^s + Q^g) + Q^r Q^s} \tag{28}$$

$$f_{\text{wet}}^{mg} = \frac{Q^r Q^g}{Q^g (Q^s + Q^g) + Q^r Q^g} \tag{29}$$

So they are not equal to $Q^r/(Q^r + Q^s + Q^g)$ and are not equal to each other either. Could you please clarify this point if still needed?

17. *p24. Please check Equation 29. I suspect m is actually $m^m$. Also, are terminal velocities missing?*

Indeed, Reviewer 2. is right. Thanks for pointing this out. We have now fixed the equation:

$$\kappa = \frac{Q^m}{\int\limits_{D_{\min}}^{D_{\max}} m^m(D) N^m(D) dD} \tag{41}$$

Terminal velocities are indeed missing from this equation, which depends only on the concentration (in kg m$^{-3}$) of the melting hydrometeor (we force the concentration of the adjusted PSD to match the concentrations of melting hydrometeors derived from Equations 23 and 24). There is no precipitation intensity involved, so no terminal velocity is needed.

18. *p24l13-19. I do not understand how propagation effects (attenuation, in particular) are taken into account when the number of quadrature points is increased in the melting layer only.*

Indeed there were some missing indications in the text. Some trades-off are required to be able to use such a simple oversampling scheme. In fact, in the melting layer integration scheme, the order of attenuation correction and integration are reversed, i.e. attenuation correction is done only at the end, after all variables have been integrated over the antenna diagram. This allows to get an estimation of $k_h$, even when not all sub-beams are used at a certain range (by integrating it only over the available sub-beams). Of course, this is a somewhat strong simplification but it is the only way to perform a local oversampling, which is the only computationally feasible way to simulate the melting layer effect. We have added some additional information in the revised version:

Unfortunately, some trades-off are required to run such a simple oversampling scheme. Because the number of quadrature points is not constant at every radar gate (as not all sub-beams cover the whole radar beam trajectory), the order of attenuation computation and integration have to be reversed, i.e. attenuation computation is done only at the very end, once all radar variables (including $k_h$ and $k_v$) have been integrated over the antenna diagram. This is a somewhat unrealistic simplification but it is the only way to perform a local oversampling, which is the only computationally feasible way to simulate the melting layer effect with volumetric integration. The effect of this approximation was investigated for the strong convective event of the 13 August 2015 (with $J = 5, K = 7$ and an oversampling factor of 10). The results indicate an overestimation of the final $Z_H$ by an average of 0.58 dBZ, with respect to the normal integration scheme. This bias is caused by the underestimation of the attenuation effect. For $Z_{DR}$ however, the bias is negligible (0.03 dB), which is likely due to the fact that this simplification affects $Z_H$ and $Z_v$ to a similar extent.

19. *p28. Please check Equation 39. A parenthesis is not balanced, the function is not Gaussian, etc.*

We thank Reviewer 2 for this correction, this equation was indeed wrong. It has been fixed in the text, both in terms of parenthesis, index of convolution and missing square in the Gaussian (it was right in the code, since we were using the appropriate function from the *numpy* and *scipy* python libraries).

$$S^{\mathrm{corr}}[i] = \sum_{j=0}^{N_{\mathrm{FFT}}} S[i-j] \frac{1}{\sigma_{t+\alpha}\sqrt{2\pi}} \exp\left[-\frac{(v_{\mathrm{rad,bins}}[j])^2}{2\sigma_{t+\alpha}^2}\right] \tag{52}$$

where $\sigma_{t+\alpha} = \sigma_t + \sigma_\alpha$

20. *p28. Please check Equation 40 which is wrong, given the definition of $k_H$ in Appendix C.*

    Thanks for pointing this out. This has been fixed in the paper. Please read all the response to Specific comment 8 of Reviewer 1, which is directly related. The new section is:

    In reality, attenuation will cause a decrease in observed radar reflectivities at all velocity bins within the spectrum. To take into account this effect, the path integrated attenuation in linear units at a given radar gate ($k_h$ in Equations C2) is distributed uniformly throughout the spectrum.

$$S(r_g, \phi_g, \theta_g)^{\mathrm{att}}[i] = S(r_g, \phi_g, \theta_g)[i] \cdot \exp\left(-2\int_{r=0}^{r_g} k_h(r, \theta_g, \phi_g)\, dr\right) \tag{53}$$

21. *p28l15. What is gamma?*

    The attenuation computation in the Doppler spectrum was wrong (see Specific comment 8 of Reviewer 1). This was fixed in the revised version, so there is no $\gamma$ anymore in this part of the paper.

22. *p32l29-31. Is $\mu_{rain}$ changed in the radar forward operator only, or in the COSMO simulations as well?*

    It has been changed in the COSMO simulations as well. We have added a sentence to make it more clear.

    Note that the COSMO model has been run twice, once with $\mu^{\mathrm{rain}} = 0.5$ and once with $\mu^{\mathrm{rain}} = 2$.

23. *. p34l27. I do not understand why it is argued that GPM tends to underestimate larger reflectivities to explain why larger reflectivities are present more frequently in the simulations. Attenuation is taken into account in the simulations, isnt it? Please elaborate.*

    Yes attenuation is taken into account, though for GPM, which is a spaceborne radar, its effect is quite small. We were just stating that previous comparison by Speirs et al. (2017) have shown that GPM tends to be negatively biased in complex terrain in terms of estimated precipitation intensities at the ground when compared with rain gauge and the operational C-band QPE. So, the results we observe in Figure 20 (there are less observations of high reflectivities than simulated values), are not surprising.

    We have slightly rephrased the corresponding sentence:

    Note that similar observations in terms of underestimation of surface rainfall intensities by GPM with respect to the Swiss operational rain gauge and radar precipitation products have been reported by Speirs et al. (2017).

24. *p46l9-11. The reference is incomplete.*

    We have fixed this reference.

---

## Referee Report (RR1)

**2nd Review on "From model to radar variables: a new forward polarimetric radar operator for COSMO" from Wolfensberger et. al**

Anonymous reviewer

**1 General Comments**

The authors have done a good job in improving the manuscript and I am generally satisfied as to how my comments have been addressed. Only two minor issues are listed below, which I regard as technical corrections and which are easy to correct.

**2 Technical Correction**

**Page 33, line 1:**

The numeric representation of the convolution of $S$ with the Gaussian distribution function of turbulence (let's call it $p$) in Eq. (52) is still not correct. It should be a numerical representation of the convolution integral

$$S(v) = \int_{-\infty}^{\infty} S(v')\,p(v - v')\,dv' \tag{1}$$

with

$$\int_{-\infty}^{\infty} p(v - v')\,dv' = 1 \tag{2}$$

Eq. (1) can therefore be written as

$$S(v) = \frac{\int_{-\infty}^{\infty} S(v')\,p(v - v')\,dv'}{\int_{-\infty}^{\infty} p(v - v')\,dv'} \tag{3}$$

Two points:

1) the sum of weights (the discrete $p(v_i - v_j)$ values) in Eq. (52) is not 1 but their integral, provided the range of $v_j$-values adequately covers the Gaussian tails.

2) the sum in Eq. (52) does not cover the whole integration range $[-\infty, \infty]$, so the numerical fullfillment of 1) depends on the mean and standard deviation of $p$ in relation to the range of $v_j$-values.

Taking these points into account, I would propose the following bias-free numerical formula:

$$S_i = \frac{\sum\limits_{j=0}^{N_{fft}} S_j \, p(v_i - v_j) \, \Delta v}{\sum\limits_{j=0}^{N_{fft}} p(v_i - v_j) \, \Delta v} = \frac{\sum\limits_{j=0}^{N_{fft}} S_j \, p(v_i - v_j)}{\sum\limits_{j=0}^{N_{fft}} p(v_i - v_j)} \tag{4}$$

assuming equidistant spacing $\Delta v$ of the $v_j$ values. This is what I meant by "You have to divide by the sum of the Gaussian weights".

Because the authors used a library function to compute the convolution (which I would assume to be working correctly), only the formula in the text has to be changed, not the results.

**Page 4, line 16-20:**

Your COSMO version 5.04 does contain ice sedimentation in the microphysics. The COSMO version numbers are unfortunately not consistently used in the COSMO community. Actually your version 5.04 is the same as 5.4, just an incoherent version number labelling across developers. Therefore, your 5.04 is newer than what I called 5.1 in my previous review and includes ice sedimentation. Sorry, I should have called it 5.01 to maintain consistency.

---

## Author Response (AR2)

**Answer to reviewers - second revision**

Daniel Wolfensberger and Alexis Berne

May 11, 2018

**Anonymous reviewer #1**

We thank reviewer 1 for taking the time to revise again our work and for suggesting some additional corrections

**Technical corrections**

1. *Page 33, line 1: numerical representation of the Gaussian convolution*

   Thank you for these precisions. We have adjusted the equation according to your suggestions.

   $$S^{\mathrm{corr}}[i] = \frac{\sum\limits_{j=0}^{N_{\mathrm{FFT}}} S[j]G(v_{\mathrm{rad,bins}}[i] - v_{\mathrm{rad,bins}}[j])}{\sum\limits_{j=0}^{N_{\mathrm{FFT}}} G(v_{\mathrm{rad,bins}}[i] - v_{\mathrm{rad,bins}}[j])} \tag{1}$$

   where $G$ is the Gaussian kernel defined by:

   $$G(x) = \frac{1}{\sigma_{t+\alpha}\sqrt{2\pi}} \exp\left[-\frac{x^2}{2\sigma_{t+\alpha}^2}\right] \tag{2}$$

   where $\sigma_{t+\alpha} = \sigma_t + \sigma_\alpha$

2. *Page 4, line 16-20: ice sedimentation in the COSMO version that is being used*

   Thank you for this information. We have corrected the text accordingly.

   In the version of COSMO that is being used (5.04), ice crystals have a bulk non-diameter dependent terminal velocity, that depends on their mass concentration.

   And in the next paragraph:

   In COSMO, with the exception of ice crystals and rain in the two-moments scheme, mass-diameter relations as well as velocity-diameter relations are assumed to be power-laws. For rain in the two-moments scheme, a slightly more refined formula by Rogers et al. (1993) is used. For ice crystals, the two-moment scheme, in contrast with the one-moment scheme uses a spectral (diameter-dependent) representation of ice crystal terminal velocities.

**Anonymous reviewer #2**

We thank reviewer 2 for taking the time to revise again our work and for suggesting some additional corrections

**Technical corrections**

1. *p4l16-19. 'In terms of terminal velocities, in the version of COSMO that is being used (5.04), neither ice crystals nor cloud droplets are sedimentating.' is repeated twice.*

   We have rephrased this sentence in accordance to the second remark of reviewer 1. So there is no repetition anymore.

2. *p10l34-p11l5. Please either remove these explanations or merge them with the newly introduced ones (in blue p10l25-32).*

   We're sorry about this regrettable formatting issue. We indeed forgot to remove the older part when we added the new one (in blue). This is now fixed.

3. *p11l6. 'Trilinear downscaling' should be mentioned before. Otherwise, the previous explanations are wrong in the general case: for example, nearest-neighbour interpolation preserves the mathematical relations between radar observables. By the way, the term 'interpolation' seems more appropriate to me than 'downscaling' in this context.*

   Yes we agree. We have replaced all mentions of downscaling by interpolation and we have modified the mentioned paragraph to introduce the fact that trilinear interpolation is being used and insist that the explanations are valid for linear interpolation only.

   Once the coordinates of all radar gates have been defined, the model variables must be interpolated to the location of the radar gates. This is done with trilinear interpolation (linear interpolation in three dimensions). The advantage of interpolating model variables before estimating radar observables, instead of doing the opposite, is twofold. At first, it is much more computationally efficient, because computing radar observables requires numerical integration over a particle size distribution at every bin, which is costly. Secondly, computing radar observables after linear interpolation allows to preserve the mathematical relation between them. Indeed, radar variables are far from being independent. For example, in the liquid phase $Z_\mathrm{H}$ is closely co-fluctuating with $Z_\mathrm{DR}$, in the form of a power-law that tends to stagnate at large reflectivities. Some tests were performed on random Gaussian fields of rain mass concentration. The results indicate that when computing the radar observables first and then interpolating them, this theoretical relation becomes more and more linear when the the interpolation resolution increases, which is quite unrealistic. In contrary, when computing the radar variables after interpolating the rain concentration field, the theoretical relationship is always preserved, regardless of the interpolation technique that is being used.

4. *p13eq8. The 3-dB beamwidth should be squared.*

   Thank you for pointing this out. Indeed we forgot the square. We have fixed it in the equation.

$$I\left[y\right](r_o, \theta_o, \phi_o) = \int\limits_{\theta_o - \pi/2}^{\theta_o + \pi/2} \int\limits_{\phi_o - \pi}^{\phi_o + \pi} y(r_0, \theta, \phi) \, \exp\left(-8\log 2 \left[\frac{\theta_0 - \theta}{\Delta_\mathrm{3dB}}\right]^2 - 8\log 2 \left[\frac{\phi_0 - \phi}{\Delta_\mathrm{3dB}}\right]^2\right) \cos\theta d\theta d\phi \qquad (3)$$

5. *p13eq9. Inconsistency in equation*

   Yes, thank you for insisting on this point. There were indeed many inconsistencies in this equation and in the following explanation. First of all, the equation should have a double sum, and second $z_j$ and $z_k$ were used in the equation but $z_i$ and $z_j$ were used in the following explanation. We have rewritten the equation and the explanation to make it consistent and accurate.

   Note that the index $j$ now stands for elevational integration and the index $k$ for elevational integration, to be consistent with equation 8., where the first integral corresponds to the elevation angle, and with order of the arguments of $y$. Placing the elevational integration sum first also allows to factorize the cosinus term, simplifying the equation.

$$I\left[y\right](r_o, \theta_o, \phi_o) \approx \sum_{j=1}^{J} w'_j \cos\left(\theta_0 + z'_j\right) \sum_{k=1}^{K} w'_k \; y(r_0, \; \theta_0 + z'_j, \; \phi_0 + z'_k) \tag{4}$$

where $w'_j = \sigma w_j$, $w'_k = \sigma w_k$ and $z'_j = \sigma z_j$, $z'_k = \sigma z_k$ with $\sigma = \frac{\Delta_{3\text{dB}}}{2\sqrt{2\log 2}}$, where $\Delta_{3\text{dB}}$ is the 3 dB beamwidth of the antenna in degrees. $w_j$ and $z_j$ are respectively the weights and the roots of the Hermite polynomial of order $K$ (for elevational integration) and $w_k$ and $z_k$ are the weights and roots of the Hermite polynomial of order $K$ (for azimuthal integration).

6. *p24 eqs28 and 29. I thought that the fraction in eq 28 could be simplified by dropping $Q^s$ in the numerator and denominator, and that the fraction in eq 29 could be simplified by dropping $Q^g$ in the numerator and denominator. After doing so, isn't it obvious that $f^{ms}_{wet} = f^{mg}_{wet}$?*

Yes, thanks, of course they are equal. Sorry for not realizing this earlier...We have fixed it in the text:

The wet fraction within melting hydrometeors can be estimated by the fraction of mass coming from rainwater over the total mass. This results in equal wet fraction for wet snow and wet graupel:

$$f^{ms}_{\text{wet}} = f^{mg}_{\text{wet}} = \frac{Q^r}{Q^s + Q^g + Q^r} \tag{5}$$

7. *p45eqC2. The upper limits of the integrals are misplaced.*

Yes thank you for pointing this out. This is a small issue that appeared when using a script (*latexdiff*) to highlight in blue the modifications added to the text during the first revision. This typo is not present in the latest version of the manuscript.

8. *p46-52. Please check the references carefully. There are still typos here and there, such as 'reectivity' and missing conference name in Furukawa et al. (2016).*

Thanks for pointing this out, we have corrected the typos, added the conference name for this particular citation and have homogenized the references, for example by adding all the DOIs that were missing.

9. *p58fig20. I am not sure I understand the y axes. Which density is meant here? What is the reference? In panel c, density does not seem to be normalized (ie, the area under the curve is not equal to 1) since the density of GPM observations can exceed 1. I am actually wondering whether the densities of observations and COSMO in panel c are comparable. GPM can certainly not detect low values such as 1E-3 mm/h (by the way 'hour' should be abbreviated with 'h' in the international systme of units). In panels a and b, the x axis starts at 14 dBZ, which corresponds to the sensitivity of GPM observations (by the way, in panel c, the x-axis legend is wrong). In contrast, in panel c, the sensitivity of GPM observations does not seem to be taken into account. The same remarks about the meaning of 'density' apply to p56fig17 and p59fig21.*

The density here indicates the frequency density, i.e. the y-axis of a normalized histogram for which the area (integral) is equal to one. So the units would be the proportion of samples per unit of variable of interest $x$, so 1 over over the units of $x$, in analogy with the density of a probability density function. We have added an explanation the first time the term is employed (second paragraph of Section 4.2).

Note that in the scope of this work, the term *density* indicates the frequency density, in analogy with a probability density function. It represents the proportion of samples within every bin divided by the width of the bin, such that the integral of the empirical distribution is equal to one. It is thus in units of $x^{-1}$, where $x$ is the unit of the considered variable (in this particular case $x = \text{m s}^{-1}$).

We have also fixed the notation for hours by replacing hr by h on all the plots.

Finally regarding the strange values of density in the precipitation distribution, you are indeed right, this is confusing. The reason for this is that the histogram has actually been computed in $\log_{10}$ scale, so the values are between -5 and 2, and the bin widths are thus smaller than 1, which explains why you can have values of density larger than 1. Thanks to your pertinent remark, we noticed that for this particular type of plot, it is not correct to replace the logarithmic x axis tick labels by their linear equivalents (e.g. 1 = 10, 2 = 100). We have thus reverted to the tick labels in logarithmic scale and have updated the x-axis label accordingly (insisting on the fact that the histogram is computed on the logarithm of precip. intensities).